# ANGPTL4 exacerbates pancreatitis by augmenting acinar cell injury through upregulation of C5a

Kyung Hee Jung[1,†], Mi Kwon Son[1,†], Hong Hua Yan[1], Zhenghuan Fang[1], Juyoung Kim[1], Soo Jung Kim[1], Jung Hee Park[1], Ji Eun Lee[1], Young-Chan Yoon[1], Myeong Seong Seo[1], Beom Seok Han[1], Soyeon Ko[1], Young Ju Suh[1], Joo Han Lim[1], Don-Haeng Lee[1], Ziqiang Teo[2], Jonathan Wei Kiat Wee[2], Nguan Soon Tan[2,3,*] & Soon-Sun Hong[1,**]

## Abstract

Pancreatitis is the inflammation of the pancreas. However, little is known about the genes associated with pancreatitis severity. Our microarray analysis of pancreatic tissues from mild and severe acute pancreatitis mice models identified angiopoietin-like 4 (ANGPTL4) as one of the most significantly upregulated genes. Clinically, ANGPTL4 expression was also increased in the serum and pancreatic tissues of pancreatitis patients. The deficiency in ANGPTL4 in mice, either by gene deletion or neutralizing antibody, mitigated pancreatitis-associated pathological outcomes. Conversely, exogenous ANGPTL4 exacerbated pancreatic injury with elevated cytokine levels and apoptotic cell death. High ANGPTL4 enhanced macrophage activation and infiltration into the pancreas, which increased complement component 5a (C5a) level through PI3K/AKT signaling. The activation of the C5a receptor led to hypercytokinemia that accelerated acinar cell damage and furthered pancreatitis. Indeed, C5a neutralizing antibody decreased inflammatory response in LPS-activated macrophages and alleviated pancreatitis severity. In agreement, there was a significant positive correlation between C5a and ANGPTL4 levels in pancreatitis patients. Taken together, our study suggests that targeting ANGPTL4 is a potential strategy for the treatment of pancreatitis.

**Keywords** acute pancreatitis; ANGPTL4; C5a; macrophage
**Subject Categories** Immunology; Molecular Biology of Disease

## Introduction

Acute pancreatitis is an inflammatory disorder of the pancreas that is associated with a mortality rate of 15–25% (Steer, 2002). Acute pancreatitis has long been considered to be a disorder of pancreatic self-digestion, in which premature and intracellular activation of digestive proteases induces tissue injury (Gaisano & Gorelick, 2009). Cytokines and chemokines released from damaged pancreatic cells attract inflammatory cells; the disease severity is determined by the systemic response of inflammatory cells and cytokines/chemokines. A recent study reported that activation of leukocytes such as macrophages occurs before the onset of necrosis (Liou et al, 2013). However, the etiological factors involved in the initiation and exacerbation of acute pancreatitis are still poorly understood, and disease treatment is limited to supportive therapies (Pandol et al, 2007). Thus, it is imperative to understand the pathophysiology of this disorder and develop clinical strategies to attenuate disease progression.

Angiopoietin-like protein 4 (ANGPTL4) consists of a coiled-coil domain at the N-terminus and a fibrinogen-like domain at the C-terminus. ANGPTL4 is expressed in several organs and tissues, including adipose tissue, liver, gut, and blood (Hato et al, 2008). ANGPTL4 is involved in the regulation of lipid and glucose metabolism (Morris, 2018). Several studies have reported that ANGPTL4 knockout (−/−) mice display a 65–90% reduction in fasting TG levels and slightly reduced total cholesterol levels, in addition to decreased circulating VLDL and increased lipoprotein lipase activity (Koster et al, 2005). In particular, these effects are mediated by peroxisome proliferator-activated receptors (PPARs), which stimulate ANGPTL4 expression via a PPAR-response element in the mouse and human *ANGPTL4* genes (Mandard et al, 2004). Recent studies have indicated that ANGPTL4 is also involved in the regulation of inflammatory signaling. Studies have reported that ANGPTL4 protects against the severe proinflammatory effects of saturated fat in lipid metabolism and reduces the infiltration of leukocytes in colitis (Lichtenstein et al, 2010; Phua et al, 2017). In contrast, others have found that ANGPTL4 is increased in inflammation-induced brain and adipose tissues, and lung infection, suggesting that ANGPTL4 is an inflammation

1  Department of Medicine, College of Medicine, Inha University, Incheon, Korea
2  School of Biological Science, College of Science, Nanyang Technological University Singapore, Singapore City, Singapore
3  Lee Kong Chian School of Medicine, Nanyang Technological University Singapore, Singapore City, Singapore
   *Corresponding author. Tel: +65 6904 1295; Fax: +65 6339 2889; E-mail: nstan@ntu.edu.sg
   **Corresponding author. Tel: +82 32 890 3683; Fax: +82 32 890 2462; E-mail: hongs@inha.ac.kr
   †These authors contributed equally to this work

mediator (Rummel *et al*, 2008; Brown *et al*, 2009; Guo *et al*, 2015; Li *et al*, 2015). Although ANGPTL4 plays a role in inflammation, its exact function and mechanism in the inflammatory response is context-dependent and still remains largely undefined, specifically in pancreatitis. This prompted us to investigate the precise role of ANGPTL4 during acute pancreatitis-induced tissue injury and subsequent local and systemic inflammation. Herein, we provide evidence that ANGPTL4 exacerbates pancreatitis and elucidate its underlying mechanisms of action. Our study shows that ANGPTL4 induces macrophage-secreted C5a, which leads to massive release of inflammatory cytokines, resulting in increased pancreatitis severity by injuring acinar cells.

## Results

### ANGPTL4 expression in mild and severe pancreatitis

To identify target genes that are associated with pancreatitis severity, we established mild acute pancreatitis (AP) and severe pancreatitis (SAP) animal models with cerulein and LPS, respectively, and performed microarray analysis of pancreatic tissue that was isolated from these animals. We selected genes with > fourfold increase in both AP and SAP groups, and focused on genes with a gradual increase in expression based on the severity of pancreatitis. Interestingly, the expression of ANGPTL4 was predominantly increased 4.5-fold in AP and 17.1-fold in SAP compared with that of the control group (Appendix Fig S1A and B).

To identify the biological significance of ANGPTL4 in a clinical situation, we first assessed ANGPTL4 expression in pancreatitis patients. A fourfold increase in ANGPTL4 levels was observed in the serum of pancreatitis patients, with increase in amylase, lipase, and CRP (Fig 1A). In addition, the expression of ANGPTL4 and amylase was increased in the pancreatic tissues of pancreatitis patients compared with that of adjacent normal pancreatic tissues (Fig 1B). These results were also confirmed in the serum and tissue from AP and SAP animal models. As expected, we observed that ANGPTL4 expression was increased in pancreatic tissues and serum of AP and SAP models (Fig 1C and D). Furthermore, to identify the expression pattern of ANGPTL4 in pancreatitis, we investigated its expression at 2 h, day 1, and 3 days after

the induction of AP and SAP in mice. While the expression of ANGPTL4 was marginally detectable in normal mouse pancreatic tissues, it was markedly increased during AP and SAP (Appendix Fig S2A and B). ANGPTL4 expression increased concomitantly with the number of apoptotic acinar cells (TUNEL-positive cells) and amylase levels, suggesting that ANGPTL4 is associated with the severity of pancreatitis. Likewise, we observed an increase in the levels of cytokines such as TGF-β, IFN-γ, and IL-1β (Appendix Fig S2C). Until now, ANGPTL4 expression has been reported mainly in adipose tissue, muscle, and the liver (Aryal *et al*, 2018). However, there have been few studies on the expression of ANGPTL4 in the pancreas. Therefore, we investigated whether there was a change in the expression of ANGPTL4 in pancreatitis. ANGPTL4 expression was evaluated in multiple organs from AP and SAP mice, and it progressively increased in the pancreas according to the severity of pancreatitis compared with that of other organs; and in normal mice, ANGPTL4 expression was lower in pancreatic tissues than in adipose, muscle, heart, liver, or kidney tissues (Appendix Fig S1C).

### Pancreatitis induction by ANGPTL4

Because ANGPTL4 was highly expressed in AP and SAP, and affected the induction of the pancreatitis response in acinar cells, we investigated whether ANGPTL4 induced pancreatitis (Appendix Fig S3A–C). ANGPTL4 at dosages of 2 or 4 mg/kg induced severe pathological changes together with increases in amylase and inflammatory cytokines. As shown in Fig 2, the serum level of ANGPTL4 was increased in all groups except the control group. In particular, there was a trend of slightly increased levels in serum in the ANGPTL4-injected groups [AP+ANGPTL4 (2 mg/kg) and ANGPTL4 (4 mg/kg)], compared with those in the other groups. Additionally, we confirmed that the AP+ANGPTL4 group exhibited pathological phenomena that were associated with severe pancreatitis together with high amylase and cytokine levels (Fig 2A and B). The ANGPTL4 alone (4 mg/kg) group showed not only pathological changes associated with severe pancreatitis but also higher cytokine levels than those of the SAP group.

Since vacuoles are known to accumulate in pancreatitis, we also detected acinar cell vacuolation in response to ANGPTL4 treatment using transmission electron microscopy (TEM). As shown in Fig 2C,

**Figure 1. ANGPTL4 expression in pancreatitis.**

A  Concentrations of amylase, lipase, CRP, and ANGPTL4 in the sera of patients (*n* = 90) diagnosed with pancreatitis. The box plots show a typical display consisting of a median value depicted by the line in the center of the box; an interquartile range (IQR; 25th to the 75th percentile) depicted by the box; and the maximum (Q3 + 1.5*IQR) and minimum (Q1-1.5*IQR) values depicted by the whisker. Statistical significance of Mann–Whitney U-tests is indicated (**$P$ < 0.01 and ***$P$ < 0.001). Exact $P$ values are shown in Appendix Table S2. ANGPTL4 and CRP concentration from pancreatitis patients were determined by sandwich ELISA. The correlation between ANGPTL4 and CRP is depicted with Pearson correlation coefficient (R). Each dot represents the levels of ANGPTL4 and CRP from an individual patient (*n* = 90).

B  Pancreatitis and adjacent normal regions (*n* = 6) were stained by immunofluorescence for amylase and ANGPTL4. Scale bar represents 30 μm.

C  H&E staining, ANGPTL4 and amylase expression by immunofluorescence analysis after the induction of mild (AP) and severe acute pancreatitis (SAP) for 2 h (*n* = 15, each group). Scale bar represents 50 μm.

D  Amylase, lipase, and ANGPTL4 were assessed in the serum from AP and SAP animal models, and ANGPTL4 protein expression was determined in the blood and pancreas (*n* = 15, each group).

Data information: Each value represents the mean ± SEM (**$P$ < 0.01 and ***$P$ < 0.001). Values of $P$ were calculated using one-way ANOVA (D) with Tukey's post hoc analysis. Exact $P$ values are shown in Appendix Table S2.
Source data are available online for this figure.

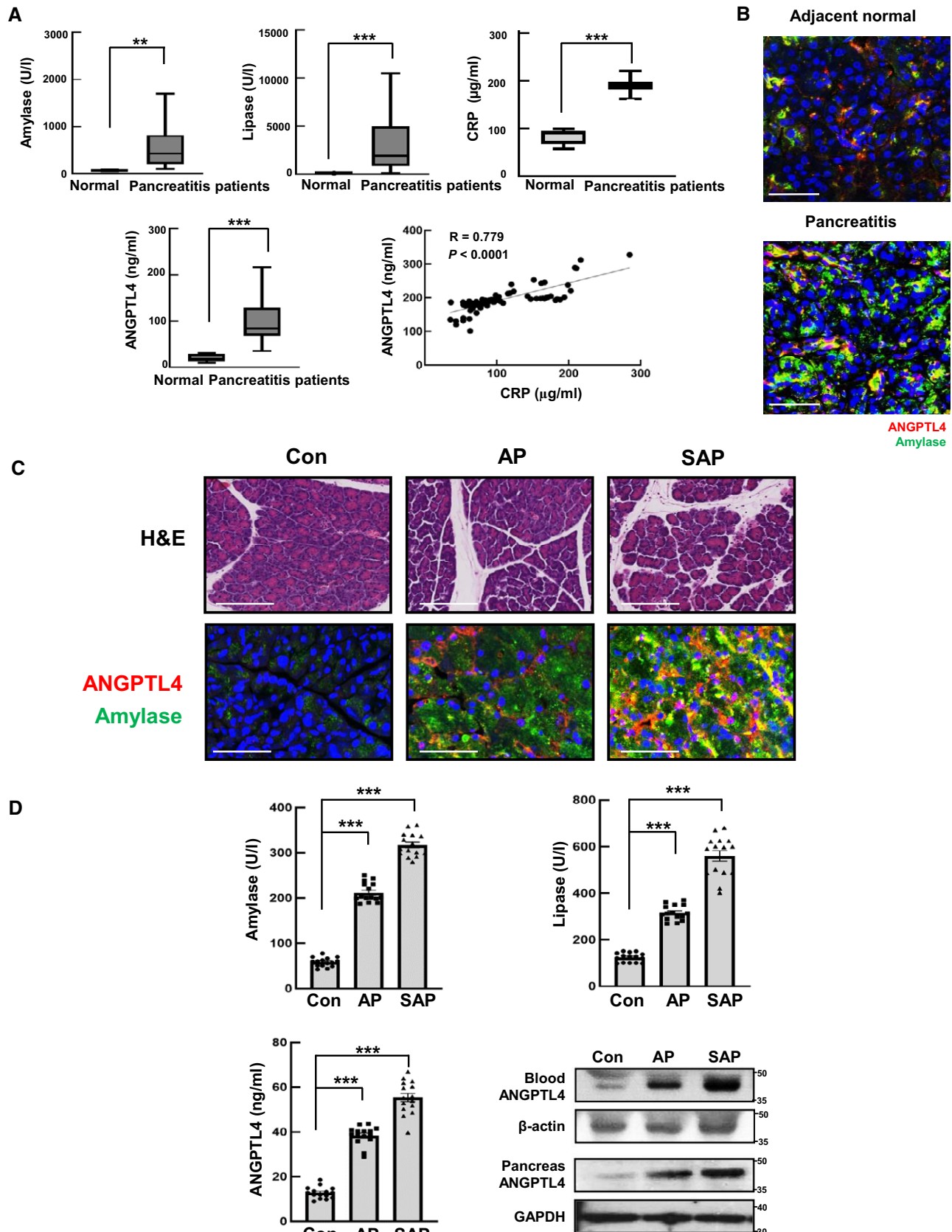

**Figure 1.**

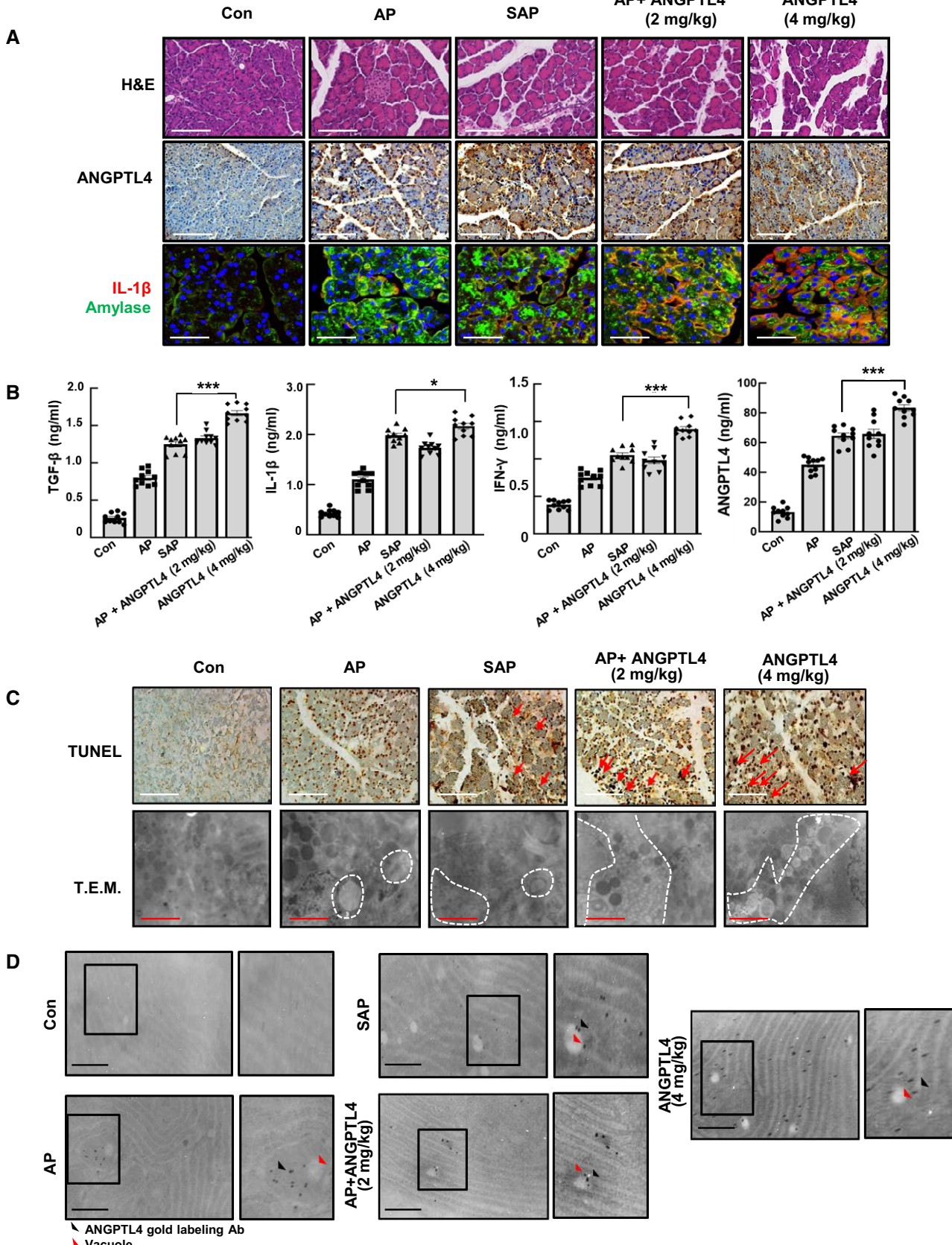

Figure 2.

**Figure 2. Pancreatitis induction by ANGPTL4.**

A   AP was induced by 5 injections (IP) of cerulein at a dosage of 50 µg/kg. SAP was induced by AP and one injection of lipopolysaccharide (LPS, IP) at a dosage of 20 mg/kg. AP+ANGPTL4 (2 mg/kg) was induced by AP and one injection of ANGPTL4 (IP) at a dosage of 2 mg/kg. ANGPTL4 (4 mg/kg) was induced by two injections of ANGPTL4 (IP) at a dosage of 2 mg/kg. Mice were sacrificed 6 h after pancreatitis induction. H&E and immunofluorescent staining for IL-1β (red) and amylase (green) was performed along with immunohistochemistry for ANGPTL4 in pancreas tissues (*n* = 10, each group). Scale bar represents 50 µm.

B   Concentrations of the inflammatory cytokines TGF-β, IL-1β, and IFN-γ as well as ANGPTL4 in the sera (*n* = 10, each group).

C   TUNEL staining for apoptosis and vacuoles formation (TEM). Red arrow, apoptotic cells; white dotted line, vacuoles region.

D   T.E.M. ANGPTL4 immunogold labeling was performed in pancreas tissues from the AP, SAP and ANGPTL4-treated models (*n* = 5, each group).

Data information: Each value represents the mean ± SEM. (*P < 0.05 and ***P < 0.001). Values of P were calculated using *t*-test and one-way ANOVA with Tukey's post hoc analysis. Exact P values are shown in Appendix Table S2. Scale bars represent 50 µm (white), 500 nm (red), and 100 nm (black).

numerous vacuoles containing non-degraded or partially degraded materials (including zymogens) were observed in acinar cells from all the groups, except the control group. Of note, higher number of vacuoles was observed in the ANGPTL4-treated groups than in the AP and SAP groups. These results were confirmed by ANGPTL4 immunogold labeling experiments, wherein immunogold staining of ANGPTL4 around the vacuoles was observed not only in the AP and SAP groups, but also in the ANGPTL4-treated groups (Fig 2D).

## Alleviation of pancreatitis in ANGPTL4−/− mice

To address whether ANGPTL4 induces pancreatitis directly or indirectly by interacting with other factors, we used ANGPTL4−/− mice and assessed the role of ANGPTL4 in acute pancreatitis (AP and SAP, Fig 3). According to the histochemical assessment of pancreatic damage, ANGPTL4−/− mice with AP and SAP showed a significant decline in acinar cell death, inflammatory cell infiltration, and edema, compared with those of AP and SAP in wild-type (WT) mice (Fig 3A and B). The serum levels of amylase, lipase, and MPO, the most commonly used biochemical markers for pancreatitis, were also significantly reduced in the ANGPTL4−/− mice (Fig 3B). Moreover, the levels of cytokines and inflammatory mediators such as TNF-α, IL-6, IL-1β, IFN-γ, and TGF-β were also decreased in the ANGPTL4−/− mice (Fig 3C). These results demonstrate that the loss of ANGPTL4 alleviated the severity of pancreatitis by decreasing inflammation and pancreatic acinar cell death in AP and SAP.

## Association of ANGPTL4 with TG or PPAR-γ

ANGPTL4 is known to inactivate LPL, which reduces triglycerides (TG) conversion to free fatty acids, leading to hypertriglyceridemia (Sukonina *et al*, 2006; Mattijssen & Kersten, 2012). Additionally, ANGPTL4 is associated with increased plasma TG (Barchuk *et al*, 2019; Gao *et al*, 2019). To identify whether elevated plasma TG by

ANGPTL4 is a key risk factor in pancreatitis, we evaluated TG levels in pancreatitis patients (*n* = 80) compared with those of normal subjects. As shown in Appendix Fig S1D, there was no difference between AP patients and normal subjects. The TG levels of two groups were distributed in normal range (40–150 mg/dl). Indeed, hypertriglyceridemia results in acute pancreatitis in up to 7% of cases; however, it rarely occurs except when triglycerides levels are > 1,000 mg/dl (Khan *et al*, 2015). Also, we found that there was no difference in TG levels between ANGPTL4−/− and WT mice, showing that ANGPTL4 does not affect TG regulation in AP and SAP models.

Previous studies have reported that ANGPTL4 is regulated by PPAR-γ in lipid metabolism and PPAR-γ plays a direct role in the inflammatory cascade during early events of AP (Rollins *et al*, 2006; Kennedy *et al*, 2008; Laura *et al*, 2017). In addition, activation of PPAR-γ induced the expression and secretion of ANGPTL4 in adipocyte lipid metabolism and tumor angiogenesis (Tian *et al*, 2009; La *et al*, 2017). However, the association of PPAR-γ and ANGPTL4 in pancreatitis has not been investigated. We identified PPAR-γ transcription levels in the pancreas of ANGPTL4−/− and WT mice with pancreatitis. There was little change in the level of PPAR-γ according to the severity of pancreatitis as well as between ANGPTL4−/− and WT mice (Appendix Fig S1E and F). These findings suggest that there is little correlation between ANGPTL4 and PPAR-γ in pancreatitis.

## Interaction between ANGPTL4 and inflammatory cytokines

Because that ANGPTL4 expression was highly increased in pancreatitis, we hypothesized that ANGPTL4 affects pancreatic acinar cell injury. ANGPTL4 as well as cholecystokinin (CCK) or LPS, which are inducers of pancreatitis (Gorelick & Thrower, 2009), was administrated to acinar cells, and we examined whether these factors directly affected the death of pancreatic acinar cells, leading to

**Figure 3. Alleviation of pancreatitis in ANGPTL4−/− (KO) mice.**

A   AP and SAP were induced in wild-type (WT) and ANGPTL4−/− mice (*n* = 15, each group), as described in the material and methods. H&E staining, TUNEL staining for apoptosis, and immunofluorescence staining for ANGPTL4 (red) and amylase (green) were performed in the pancreatic tissues. ANGPTL4 and IL-6 protein expressions were also determined in pancreatic tissues from ANGPTL4 WT and −/− mice by Western blotting. Scale bar represents 100 µm.

B   Levels of ANGPTL4, amylase, lipase, and MPO in serum (*n* = 15, each group).

C   Immunofluorescence staining of TNF-α and IL-6, and the levels of the inflammatory cytokines TGF-β, IFN-γ, and IL-1β in the sera from ANGPTL4 WT and −/− mice (*n* = 10, each group). Scale bar represents 150 µm.

Data information: Each value represents the mean ± SEM. (***P < 0.001). Values of P were calculated using one-way ANOVA with Tukey's post hoc analysis. Exact P values are shown in Appendix Table S2.

Source data are available online for this figure.

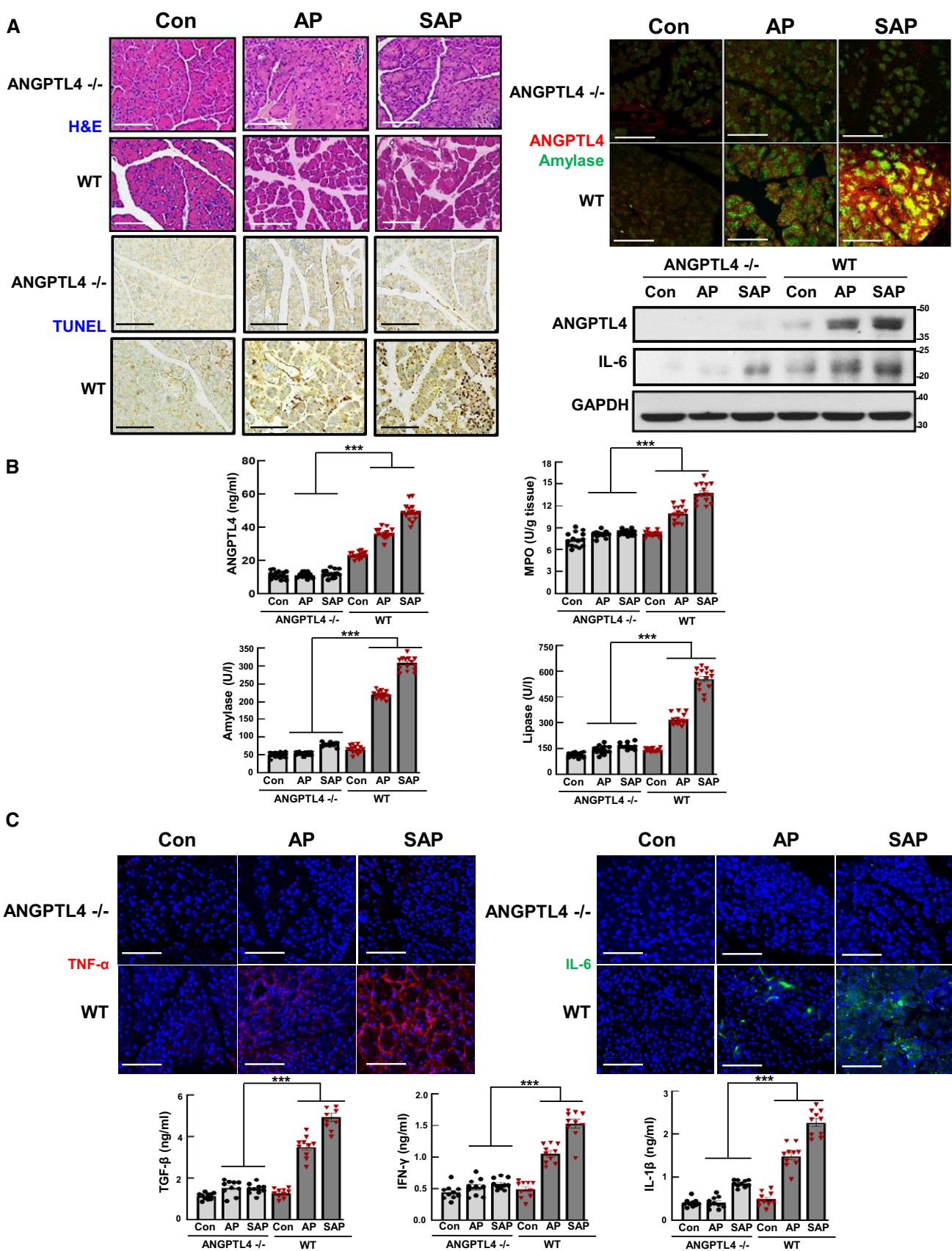

**Figure 3.**

pancreatitis. Acinar cell death in response to ANGPTL4 was less than that of CCK or LPS although ANGPTL4 increased amylase and lipase as well as TGF-β compared with that of CCK and LPS (Appendix Fig S4A–C). Additionally, ANGPTL4 increased various cytokines in acinar cells (Appendix Fig S4D). These results suggest that the induction of cell injury is due to cytokines that are released by ANGPTL4 rather than by directly acinar cell death by ANGPTL4. Additionally, we hypothesized that there would be involvement of inflammatory cells such as macrophages to regulate ANGPTL4-induced cytokine releases.

### Roles of ANGPTL4 in macrophages

The abovementioned inflammatory mediators are initially released by pancreatic acinar cells, resulting in the recruitment of macrophages and neutrophils (Mikami et al, 2002; Sakai et al, 2003). Therefore, the correlation between macrophages and ANGPTL4 in pancreatitis was investigated. We established a macrophage-depleted mouse model and subsequently induced AP and SAP with cerulein and LPS along with ANGPTL4. We observed that macrophage depletion partially blocked the pathological phenomena associated with pancreatitis. Interestingly, the ANGPTL4-treated groups [AP+ANGPTL4 and ANGPTL4 alone (4 mg/kg)] showed a reduced pancreatitis severity in the macrophage-depleted groups compared with that of the macrophage WT groups, with decreased amylase and lipase levels (Fig 4A). Moreover, the expression and secretion of ANGPTL4 were decreased in the macrophage-depleted AP, SAP, and ANGPTL4 (4 mg/kg) groups, compared with those of the macrophage WT groups. In particular, ANGPTL4 expression was considerably lower in the macrophage-depleted ANGPTL4 group than the macrophage-depleted AP and SAP groups (Fig 4B). More importantly, the expression of F4/80, a macrophage marker, was obviously decreased in ANGPTL4$^{-/-}$ mice compared with that of WT mice (Fig 4C). These findings allowed us to hypothesize that ANGPTL4 directly regulated the progression of pancreatitis through macrophages.

To identify the direct interactions between ANGPTL4 and macrophages, we investigated whether ANGPTL4 affected the activation of macrophages. ANGPTL4 induced morphological changes in macrophages including the formation of lamellipodia and filopodia within 6 and 24 h after treatment, and increased the expression of TNF-α, IL-1β, IL-6, TGF-β, and NO in macrophages (Fig 5A). When we assessed the effect of ANGPTL4 on cell migration using a slide dish with different compartments for co-culture and a transwell system, ANGPTL4 potently induced macrophage migration to acinar cells compared with that of CCK, which was similar to the results in the LPS-treated group (Fig 5B). We next investigated whether ANGPTL4 induced acinar cell death via macrophage activation and migration. As shown in Fig 5C, when primary acinar cells were co-cultured with macrophages and then treated with ANGPTL4, the macrophages migrated to the acinar cells after 12 h of treatment and subsequently induced acinar cell death at 48 h. This suggests that acinar cell death is affected by macrophage migration due to ANGPTL4. Furthermore, when acinar cells were treated with conditioned media (CM) from LPS-activated macrophages, high expression of TNF-α and ANGPTL4 as well as increases of TGF-β level was observed (Appendix Fig S5A and B). Therefore, we considered that various cytokines and chemokines released during macrophage

activation by ANGPTL4 influence acinar cell death and further pancreatitis.

### ANGPTL4-mediated upregulation of the C5a-C5aR axis via the PI3K/AKT signaling pathway in macrophages

To assess the involvement of various cytokines and chemokines that are released by macrophages upon ANGPTL4 treatment, we used cytokine profiler arrays and analyzed the macrophage-secreted cytokines and chemokines (Fig 6A). Out of 40 cytokines and chemokines, nine were highly expressed in the ANGPTL4-treated group. These included complement component 5a (C5a), CCL17, IL-4, CXCL1, CXCL4, CXCL11, TNF-α, IL-1β, and IL-6. Of these, C5a showed the most pronounced increase in ANGPTL4-treated macrophages compared with that of the control group. To confirm this result, we investigated whether C5a production was reduced in ANGPTL4$^{-/-}$ macrophages. For this experiment, we prepared bone marrow-derived macrophages from pancreatitis-induced ANGPTL4$^{-/-}$ and WT mice. As shown in Fig 6B, we found that C5a production was increased in macrophages from ANGPTL4 WT mice with pancreatitis (AP and SAP), whereas the level of C5a was reduced in macrophages from ANGPTL4$^{-/-}$ mice, compared with that in macrophages from ANGPTL4 WT mice. To investigate the mechanism by which ANGPTL4 increased C5a, we performed cellular phospho-kinase array analysis in human THP1-derived macrophages. ANGPTL4 increased the phosphorylation of AMPKα, JNK, AKT, and p70S6K in human THP1-derived macrophages (Fig 6C). These results were confirmed by Western blotting analysis. To further explore whether ANGPTL4 affects JNK, AMPK, and PI3K/AKT signaling, THP-1-derived macrophage were treated with each inhibitor (SP600125, compound C, and HS-173, respectively) after ANGPTL4 treatment. Then, we identified the expression of C5a and its related signaling. Among those, we observed that the PI3K inhibitor (HS-173) decreased the expression of C5a by inhibiting PI3K/AKT signaling in human macrophages (Fig 6D).

C5a is a potent chemoattractant that acts on a number of parenchymal cells by binding to C5aR. Accordingly, we detected the expression of C5a and C5aR after acinar cells were treated with ANGPTL4 or co-cultured with macrophages using a transwell assay. As shown in Fig 6E and Appendix Fig S6C, ANGPTL4 increased expression of C5a in macrophages and acinar cells compared with that of the control. More interestingly, C5a and C5aR expression was significantly increased in acinar cells that were co-cultured with ANGPTL4-treated macrophages, resulting in activation of the C5a-C5aR axis. These results were confirmed by immunofluorescence staining (Appendix Fig S6A and B). Moreover, siANGPTL4 considerably decreased the expression of C5a and IL-1β in LPS-induced macrophages. Additionally, when macrophages were treated with ANGPTL4 in transwell inserts (upper), the C5aR antagonist (W-54011) also inhibited cell death (TUNEL) and C5aR expression in acinar cells (bottom) compared with those of the C5a-treated group, indicating that ANGPTL4 contribute to the regulation of the C5a-C5aR axis in macrophages and acinar cells (Fig 6F).

Finally, we tested the direct role of ANGPTL4-overexpressing macrophages in promoting AP severity. Macrophages were injected into the pancreas, and macrophage-mediated AP severity was assessed on days 1, 3, and 5 after injection. The levels of

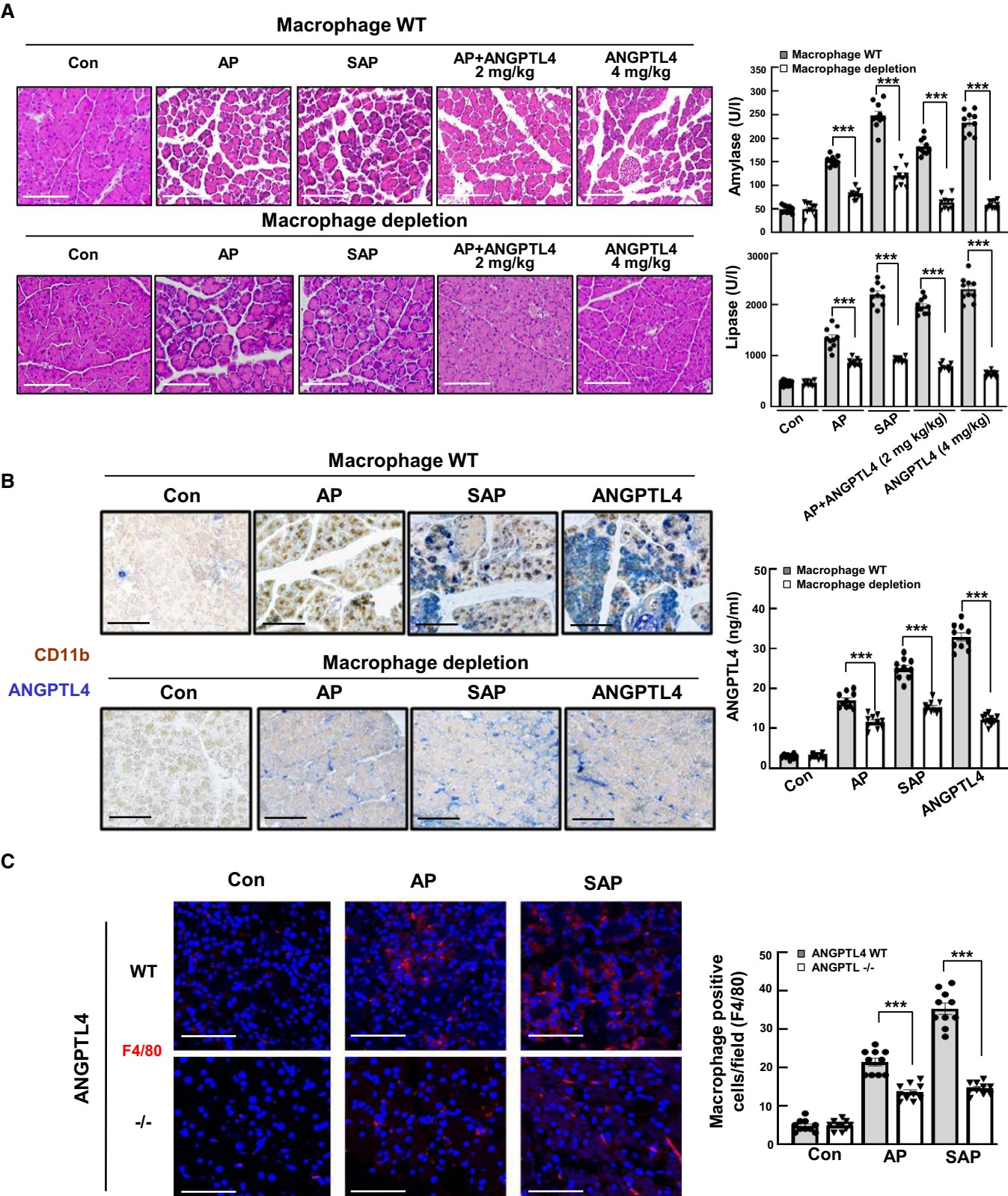

Figure 4.

**Figure 4.  Role of ANGPTL4 in macrophages.**

A  For macrophage depletion models, mice (6 weeks) were injected with GdCl₃ (15 mg/kg). After the induction of macrophage depletion in the mice, pancreatitis was induced by cerulein, LPS, and ANGPTL4 for 24 h (n = 10, each group). H&E staining of pancreatic tissue, and levels of amylase and lipase in the serum. Scale bar represents 100 μm.

B  Immunohistochemistry for the macrophage marker CD11b and ANGPTL4 in pancreas tissues. Levels of ANGPTL4 in the serum were measured in the AP, SAP, and ANGPTL4 (4 mg/kg) groups of macrophage WT or depletion mice h (n = 10, each group). Scale bar represents 100 μm.

C  Immunofluorescent staining of the F4/80 macrophage marker and quantification of macrophage-positive cells in the AP and SAP groups from the ANGPTL4⁻/⁻ and WT mice (n = 10). Scale bar represents 150 μm.

Data information: Each value represents the mean ± SEM. (***P < 0.001). Values of P were calculated using one-way ANOVA with Tukey's post hoc analysis. Exact P values are shown in Appendix Table S2.

amylase and TNF-α were noticeably increased in mice that were injected with ANGPTL4-overexpressed macrophages, compared with those in the WT macrophage-injected group. The expression levels of C5a and C5aR were also highly increased in mice that were injected with ANGPTL4-overexpressing macrophages (Appendix Fig S6D), demonstrating that ANGPTL4 in macrophages also has autocrine effect on the progression of AP through the C5a-C5aR axis.

**Therapeutic effect of ANGPTL4- and C5a-neutralizing antibodies on AP and SAP *in vitro* and *in vivo***

Since the upregulation of C5a by ANGPTL4 is associated with the severity of pancreatitis, we measured the levels of C5a in the sera of patients with pancreatitis and then evaluated the correlation between C5a and ANGPTL4. The level of C5a in patient sera was 2.5-fold higher than that of sera from normal subjects, and there was a significant positive correlation between C5a and ANGPTL4 levels in pancreatitis patients ($R^2$ = 0.72, $P < 0.001$, Fig 7A). Additionally, the expression of C5a and C5aR was higher in tissues from pancreatitis patients than in normal subjects (Fig 7A and Appendix Fig S7A). More importantly, we observed similar changes in ANGPTL4⁻/⁻ mice, which displayed the decrease of C5a levels in the serum, compared with those of WT mice (Appendix Fig S7B). Additionally, the high levels of C5a and C5aR expression that were observed in AP and SAP models were markedly decreased in ANGPTL4⁻/⁻ AP and SAP models, as well as macrophage depletion models (Appendix Fig S7B and C), suggesting that ANGPTL4 induces severe pancreatitis by regulating C5a through a cause-and-effect relationship between ANGPTL4 and C5a. We next tested whether blocking ANGPTL4 and C5a with neutralizing antibodies was sufficient to decrease the inflammatory response in LPS-activated macrophages and acinar cells, and an IgG antibody was used as a control. The ANGPTL4 antibody significantly suppressed ANGPTL4 and C5a levels, whereas the C5a antibody marginally blocked ANGPTL4 levels. Additionally, both antibodies inhibited the levels of TNF-α and IL-6 in the activated macrophages (Fig 7C).

To determine whether ANGPTL4 was responsible for AP and SAP *in vivo*, we injected the ANGPTL4-neutralizing antibody into mice after induction of AP and SAP. Similar to the *in vitro* experimental results, the antibody inhibited not only ANGPTL4 levels, but also C5a levels in AP and SAP animal models. This effect significantly reduced pathological changes, such as decreased edema formation, inflammatory cell infiltration, and necrosis (Fig 7D). Consistently, the serum levels of tissue enzymes (amylase and lipase) and the proinflammatory cytokine TNF-α were clearly reduced by treatment with ANGPTL4-neutralizing antibody. Furthermore, in the SAP models, the ANGPTL4-neutralizing antibody inhibited the expression of F4/80 (a macrophage marker), C5a, and C5aR. Similar to the effects of ANGPTL4-neutralizing antibody, treatment with C5a-neutralizing antibody also exhibited positive and significant effects in the SAP models (Appendix, Fig S8A and B).

## Discussion

AP is a common abdominal disorder that is associated with a high mortality rate in patients with a high disease severity (Mayerle *et al*, 2005). Recent studies have reported that inflammatory cells and their cytokines cause systemic complications in pancreatitis and local damage within the pancreas (Gukovskaya *et al*, 2002; Algul *et al*, 2007; Marrache *et al*, 2008). Additionally, the transmigration of inflammatory cells into the pancreas occurs very rapidly at the onset of the disease before acinar cell injury (Schnekenburger *et al*, 2008; Perides *et al*, 2011). Our study showed that ANGPTL4

**Figure 5.  Effect of ANGPTL4 on macrophages and acinar cells.**

A  Cell morphology and the mRNA levels of inflammatory cytokines, including and TNF-α, IL-6, and IL-1β in mouse primary macrophages treated with LPS and ANGPTL4 (100 ng/ml) using qPCR and RT-PCR, and levels of TGF-β and NO in the LPS- and ANGPTL4-treated primary macrophages (n = 5). Scale bar represents 100 μm.

B  Migration assays were performed with mouse primary acinar cells and macrophages using IBIDI removable 2-well silicone culture inserts that were placed in a cell culture μ-Dish and transwell at 24 h (n = 5). Scale bar represents 50 μm.

C  Mouse primary acinar cells and macrophages were labeled with fluorescence dyes (macrophages, red and acinar cells, green). Co-cultured cells treated with 100 ng/ml of ANGPTL4 or LPS were analyzed by microscopy at 12 and 48 h. Scale bars represent 80 μm (red) and 40 μm (white).

Data information: Data are represented as the mean ± SEM. for three separate experiments. (**P < 0.01 and ***P < 0.001). Values of P were calculated using t-test and one-way ANOVA with Tukey's post hoc analysis. Exact P values are shown in Appendix Table S2.

Source data are available online for this figure.

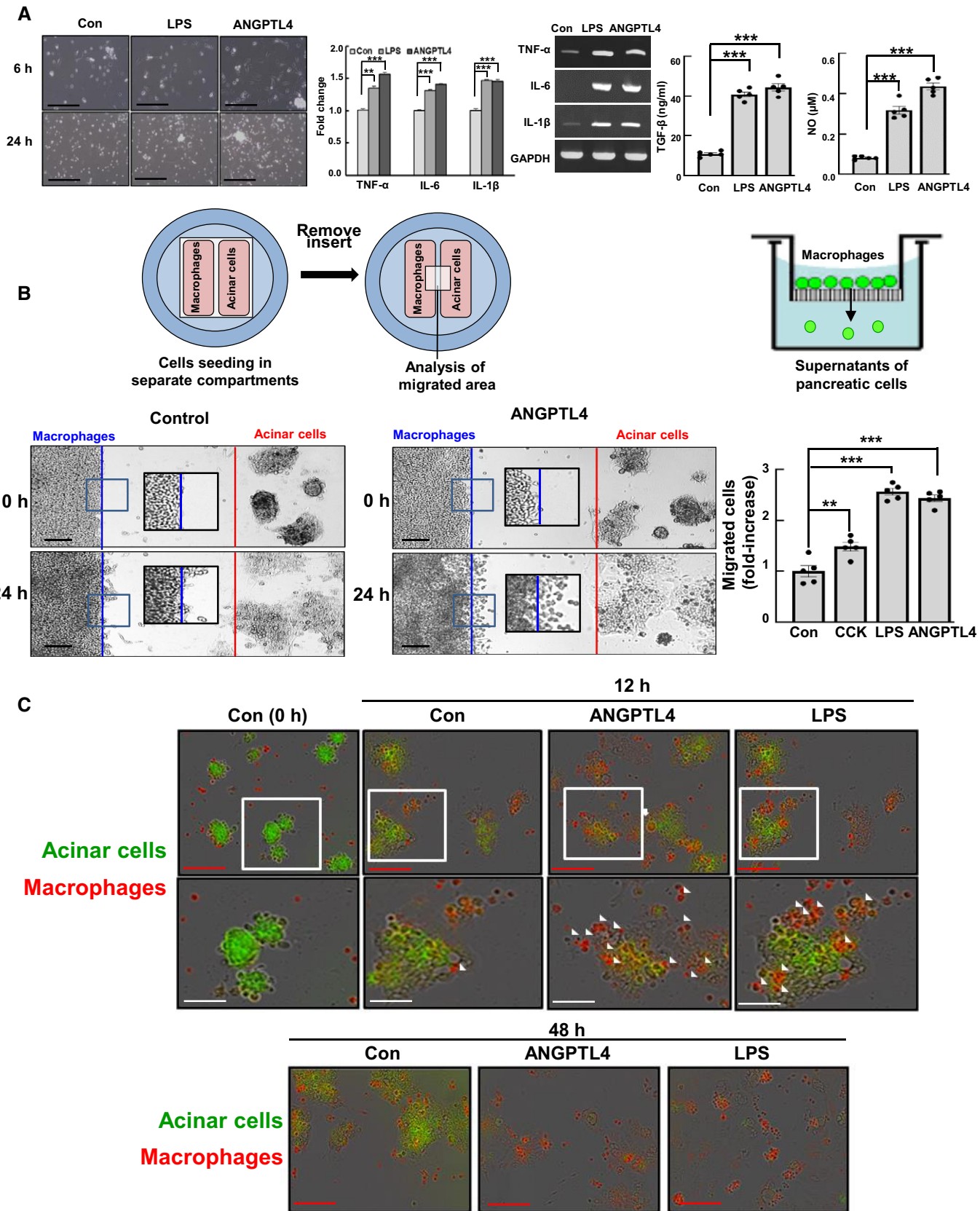

Figure 5.

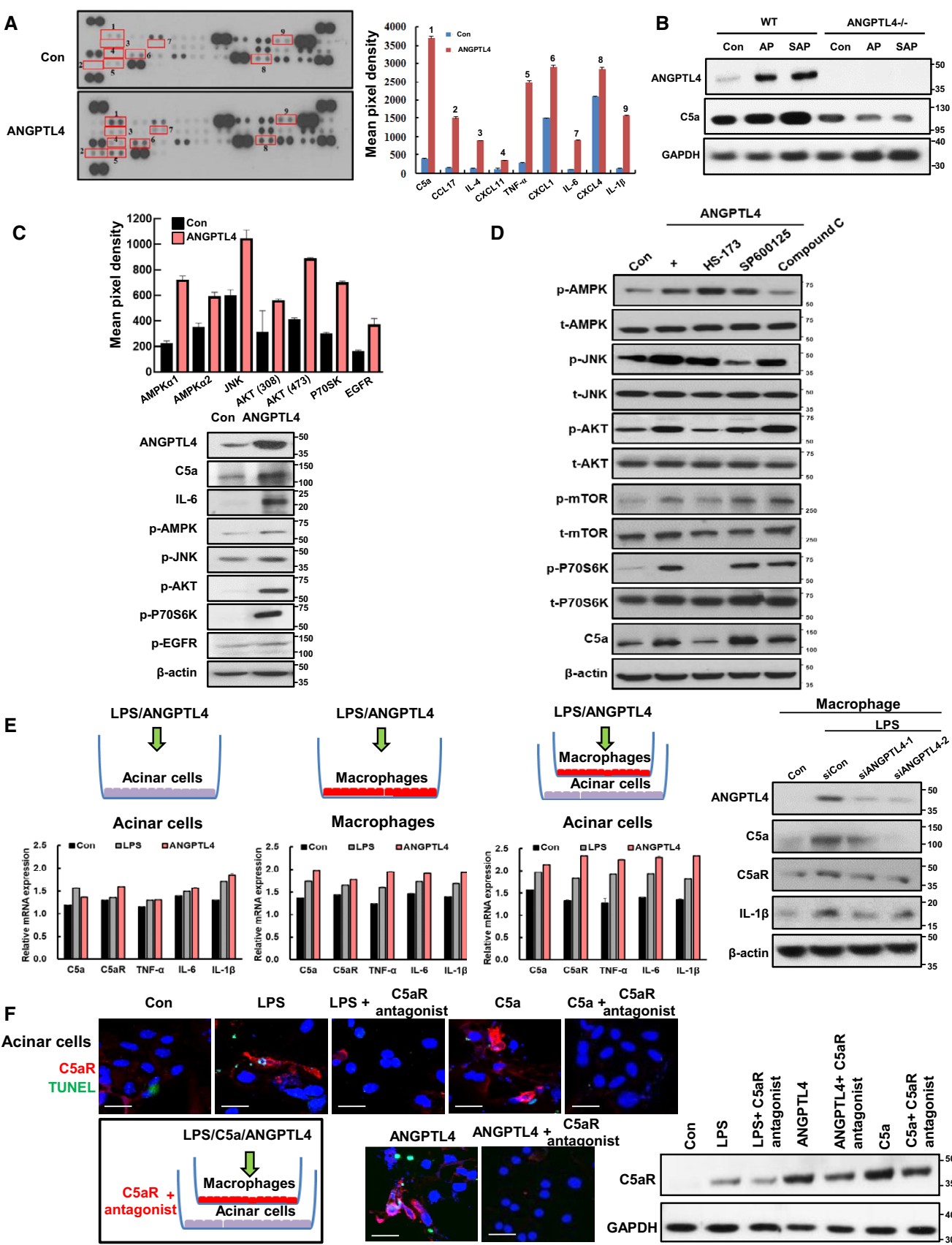

**Figure 6.**

**Figure 6.   Upregulation of C5a-C5aR axis via PI3K/AKT signaling pathway of ANGPTL4 in macrophages.**

A   Macrophage cells were treated with ANGPTL4 (100 ng/ml) for 24 h, and then, cytokine levels were analyzed by a mouse cytokine profiler assay ($n$ = 3). Mean pixel density was quantified using ImageJ software analysis.

B   ANGPTL4 and C5a expression levels were measured in bone marrow-derived macrophages from pancreatitis-induced ANGPTL4$^{-/-}$ and WT mice.

C   Phospho-kinase array was performed to find the mechanism by which ANGPTL4 increased C5a in human THP1-derived macrophage cells ($n$ = 3). Cells were treated with ANGPTL4 (100 ng/ml) for 6 h. After phospho-kinase array analysis, the expression of AMPKα, JNK, AKT, P70S6K, and EGFR was confirmed by Western blotting.

D   To explore whether ANGPTL4 affects JNK, AMPK, and PI3K/AKT signaling, each inhibitor (SP600125, compound C, and HS-173, 1 μM) was treated to in THP1-derived macrophages for 2 h after ANGPTL4 treatment (100 ng/ml) for 6 h.

E   C5a, C5aR, TNF-α, IL-6, and IL-1β mRNA levels were identified by qPCR in mouse macrophages or primary acinar cells after treatment with ANGPTL4 and LPS (100 ng/ml). In the transwell experiment, mouse primary acinar cells were cultured in the lower well and macrophages were incubated in the upper transwell bucket treated with ANGPTL4 (100 ng/ml). Expression of ANGPTL4, C5aR, and C5a was determined in siANGPTL4-treated THP1-derived macrophage after LPS activation.

F   In the transwell experiment, TUNEL-positive cells and C5aR expression were identified in primary acinar cells after C5aR antagonist (W-54011, 10 nM) pretreatment for 12 h prior to treatment with ANGPTL4, LPS, and C5a (100 ng/ml) for 24 h to macrophage in transwell. Scale bar represents 20 μm.

Data information: Data are represented as the mean ± SEM.
Source data are available online for this figure.

induced macrophage activation and migration, and activated the C5a-C5aR axis via PI3K/AKT signaling, resulting in pathological changes in pancreatitis. To the best of our knowledge, this is the first report showing that ANGPTL4 plays major pathological roles in pancreatitis severity.

To date, ANGPTL4 has been known to regulate glucose and lipid metabolism, and inflammatory signaling, and it plays either an anti- or proinflammatory role in a context- or tissue-dependent manner. Indeed, some studies have reported that ANGPTL4 plays an anti-inflammatory role in metabolic diseases (Lichtenstein *et al*, 2010; Galaup *et al*, 2012). However, others have reported that ANGPTL4 perpetuates the state of inflammation (Quintero *et al*, 2012; Qin *et al*, 2019) and functions as a positive regulator of acute-phase responses such as inflammation (Lu *et al*, 2010; Schumacher *et al*, 2015). Moreover, ANGPTL4 has differential expression pattern and roles depending on tissues/diseases, which could be due to different ANGPTL4 isoforms in complex processes such as cleavage (Sukonina *et al*, 2006), glycosylation (Yang *et al*, 2008), and oligomerization (Ge *et al*, 2004; Yau *et al*, 2009). However, the mechanisms whereby ANGPTL4 modulates inflammation in various diseases remain unclear.

In this study, ANGPTL4 was selected as a target gene that has a gradual increase according to the severity of pancreatitis based on microarray analysis of AP and SAP models. Its expression was significantly elevated in serum from patients with acute pancreatitis along with amylase, lipase, and CRP. ANGPTL4 was continuously expressed during acute pancreatitis, in both the AP to SAP models. This trend has been confirmed in other established pancreatitis animal models using arginine and cerulein (Appendix Fig S3). We determined multiorgan ANGPTL4 expression in AP and SAP models, and the expression of ANGPTL4 in the pancreas progressively increased according to severity of pancreatitis (Appendix Fig S1C). Although other organ produced ANGPTL4, it is conceivable that the increased ANGPTL4 secreted by the pancreas contributes to the elevated ANGPTL4 in the blood during AP and SAP.

These data prompted us to investigate whether ANGPTL4 was directly involved in the induction of pancreatitis using ANGPT4$^{-/-}$ mice. The loss of ANGPTL4 in mice reduced inflammatory infiltration and cytokine levels, where pancreatitis pathological phenomena such as acinar cell necrosis and protease activation were very rare, suggesting that ANGPTL4 is a regulator in the progression of pancreatitis severity.

Macrophages are a major source of TNF-α, IL-1β, and IL-6, and high serum levels of these cytokines are consistent with acute pancreatitis disease severity (Liu *et al*, 2003). Moreover, a hallmark of acute pancreatitis is the accumulation of inflammatory cells such as macrophages in the pancreas, and depletion of these cells has been shown to attenuate the severity of pancreatitis (Yu & Kim, 2014). Indeed, it has been reported that macrophage depletion blocks the progression of cerulein-induced pancreatitis (Liou *et al*,

**Figure 7.   Therapeutic effect of the ANGPTL4-neutralizing antibody in AP and SAP.**

A   Concentrations of C5a in the sera from pancreatitis patients, and correlation between ANGPTL4 and C5a in the sera from pancreatitis patients ($n$ = 90). The box plots show a typical display consisting of a median value depicted by the line in the center of the box; an interquartile range (IQR; 25[th] to the 75[th] percentile) depicted by the box; and the maximum (Q3 + 1.5*IQR) and minimum (Q1-1.5*IQR) values depicted by the whisker. Statistical significance of Mann–Whitney *U*-tests is indicated (***$P$ < 0.001). Exact $P$ values are shown in Appendix Table S2. The correlation between C5a and ANGPTL4 is depicted with Pearson correlation coefficient (R). Each dot represents the levels of C5a and ANGPTL4 from an individual patient ($n$ = 90). The immunofluorescent expression of C5a and amylase was observed in pancreatitis and adjacent normal regions in patient tissues. Scale bar represents 50 μm.

B   Immunofluorescence staining of C5a in pancreas tissues from ANGPTL4$^{-/-}$ and WT animals and macrophage depletion/WT models with pancreatitis. Scale bar represents 150 μm. Levels of C5a were measured in serum ($n$ = 7).

C   After macrophages were activated with LPS (100 ng/ml), the cells were treated with either neutralizing ANGPTL4 or C5a antibody (100 ng/ml) or an isotype control (IgG). ANGPTL4, C5a, TNF-α, and IL-6 levels were measured in culture media ($n$ = 5).

D   The neutralizing ANGPTL4 antibody (10 mg/kg) was injected twice (IP) into the AP and SAP models. H&E staining of pancreatic tissues and the concentrations of ANGPTL4, C5a, TNF-α, and amylase in the serum ($n$ = 10, each group). Scale bar represents 50 μm.

E   Summary for how ANGPTL4 induces the severity of pancreatitis in acinar cells and macrophages.

Data information: Each value in (B–D) represents the mean ± SEM. (**$P$ < 0.01 and ***$P$ < 0.001). Values of $P$ were calculated using one-way ANOVA with Tukey's post hoc analysis. Exact $P$ values are shown in Appendix Table S2.

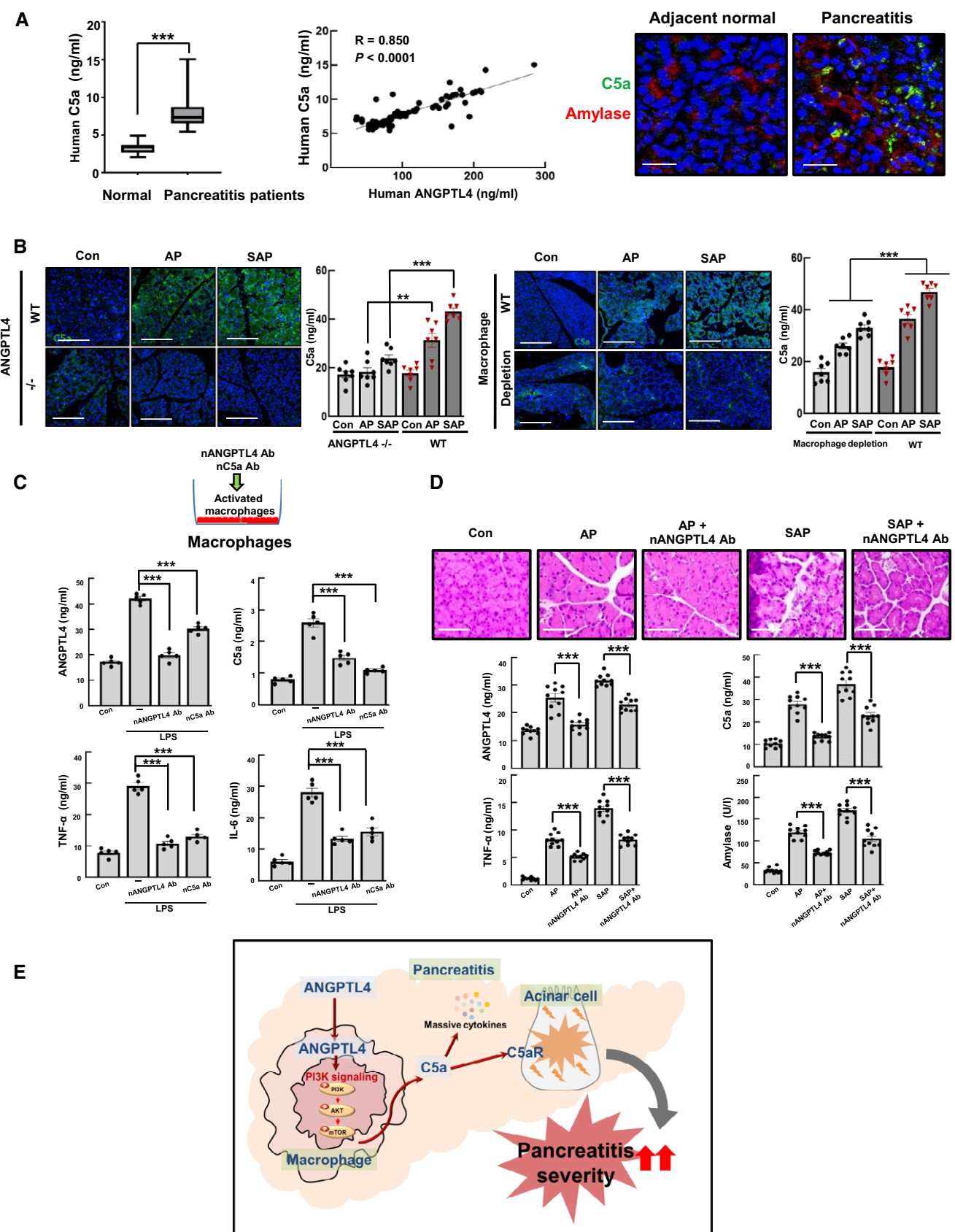

**Figure 7.**

2013). Our study showed that ANGPTL4 increased the proinflammatory cytokines such as TNF-α, IL-1β, and IL-6. In the case of ANGPTL4-induced pancreatitis models, the levels of these cytokines were higher than those of AP and SAP models. On the other hand, pancreatitis conditions induced by ANGPTL4 were prominently reduced in macrophage-depleted animal models. Specifically, ANGPTL4 did not exacerbate pancreatitis when macrophages were depleted, suggesting that ANGPTL4 plays a positive role in macrophages in pancreatitis progression. *In vitro* studies identified that ANGPTL4 activated macrophages by inducing cytokines and promoting the migration of macrophages to acinar cells, followed by acinar cell death. In this regard, our results verified that ANGPTL4 is a pivotal regulator that exacerbates the severity of pancreatitis via macrophage activation and migration. Also, we also speculated that systemic ANGPTL4 increases macrophage infiltration into pancreas, which increase C5a, leading to hypercytokinemia or cytokine storm that accelerated pancreatitis severity. Lending support, pancreatitis did not develop well in macrophage-depleted mice injected with recombinant ANGPTL4. Furthermore, ANGPTL4-overexpressed macrophages showed the increase in cytokines (Appendix Fig S7C), suggesting an autocrine action of ANGPTL4 in macrophage. Similar autocrine action of ANGPTL4 was observed in cancer cells and endothelial cells (Le Jan *et al*,2003; Zhu *et al*, 2011). Considering these results, we postulate that residing and infiltrated macrophages may be a source of increased ANGPTL4 in the pancreas during AP and SAP.

In this study, ANGPTL4 is a possible pathological mediator that induces pancreatitis and increases the inflammatory response by macrophages. To clarify the mechanism of acinar cell injury by macrophages, we examined the chemokines and cytokines that were released from ANGPTL4-activated macrophages and mediated this pathological effect on protease activation and acinar cell death using a cytokine array. C5a showed a pronounced increase in ANGPTL4-treated macrophages.

C5a is a cleavage product of C5, which is generated during the classic and alternative pathways of complement activation (Noris & Remuzzi, 2013). C5a is a potent anaphylatoxin and chemoattractant for macrophages/neutrophils and directly acts on a number of parenchymal cells by binding to C5aR; C5a is also generally believed to serve as a "complete" proinflammatory mediator (Mollnes & Kirschfink, 2006; Ricklin *et al*, 2010). In a recent study, C5a was reported to play a critical role in the progression of chronic pancreatitis (Sendler *et al*, 2015). Additionally, the complement system is activated during pancreatitis, and serum levels of C5a correlate with disease severity (Perez *et al*, 1983; Roxvall *et al*, 1989; Merriam *et al*, 1997). Consistent with these reports, our results demonstrated that C5a levels were highly elevated in the serum of pancreatitis patients. Concurrently, a significant positive correlation between C5a and ANGPTL4 levels was observed in pancreatitis patients. We also observed that the expressions levels of C5a were markedly decreased in ANGPTL4$^{-/-}$ models of AP and SAP.

Furthermore, C5a decreased gradually according to the severity of pancreatitis in macrophages from ANGPTL4$^{-/-}$ mice with AP and SAP, compared with that of WT mice, suggesting that ANGPTL4 increases C5a in macrophages, leading to increased severity of pancreatitis. However, the associations of ANGPTL4 and C5a have not been reported until now. In cellular phospho-kinase arrays in macrophages, ANGPTL4 increased the phosphorylation of AMPKα,

JNK, AKT, and p70S6K. C5a was decreased by treatment with a PI3K inhibitor (HS-173). In some studies, ANGPTL4 has been shown to be regulated by PI3K/AKT pathway in endothelial, cancer cells, and stem cells (Theofilatos *et al*, 2018; Tsai *et al*, 2019). Although the exact mechanism by which ANGPTL4 increases C5a in macrophages was not revealed, we showed that ANGPTL4 partially increases C5a through PI3K/AKT signaling.

C5a interacts with C5aR, which results in a number of effects that are essential to the inflammatory response. Recently, accumulating evidences have shown that the C5a-C5aR axis is involved in the development of diseases pathogenesis (Yan & Gao, 2012; An *et al*, 2014). In present study, corresponding to the increased C5a in macrophages, the expression of C5aR was obviously increased in acinar cells that were co-cultured with ANGPTL4-treated macrophages. However, the C5aR antagonist effectively inhibited C5aR expression and acinar cell death in this condition. These observations may explain why ANGPTL4-activated macrophages induce an increase in C5a, leading to negative effects in pancreatic acinar cells via systemic responses.

Considering the pathological role of ANGPTL4 in pancreatitis, we expected that targeting ANGPTL4 and C5a would reduce pancreatitis. Neutralizing antibodies against ANGPTL4 or C5a decreased edema formation, inflammatory cell infiltration, and necrosis along with reduction in tissue enzymes and cytokines in AP and SAP animal models. Indeed, the effects of these neutralizing antibodies have also been observed in inflammatory diseases, such as influenza pneumonia and sepsis (Hopken *et al*, 1996; Li *et al*, 2015). Our *in vitro* results showed that blocking ANGPTL4 or C5a was sufficient to decrease the inflammatory response in activated macrophages and acinar cells. Interestingly, blocking ANGPTL4 using the neutralizing antibody decreased not only ANGPTL4 but also C5A levels, but C5a neutralizing antibody did not decrease ANGPTL4, indicating that C5a is a downstream molecule that is regulated by ANGPTL4. We also identified that neutralizing antibodies against ANGPTL4 or C5a were more effective in macrophages than in acinar cells, suggesting that the pathological effect of ANGPTL4 was greater in macrophages than in acinar cells; thus, the targeting effects of ANGPTL4-neutralizing antibody might be considerably enhanced in macrophages. Overall, ANGPTL4 and C5a might be therapeutic targets for acute pancreatitis and antibodies that target ANGPTL4 or C5a could be effective in treating this disease.

In conclusion, we showed that ANGPTL4 induced pancreatitis and was correlated with acute pancreatitis severity. ANGPTL4 plays a critical role in the regulation of C5a via PI3K/AKT signaling in macrophages, thereby inducing acinar cell injury together and a massive release of inflammatory cytokines (Fig 7E). Additionally, neutralizing antibodies targeting ANGPTL4 and C5a improved pancreatitis and pathological change *in vitro* and *in vivo*. Our findings suggest that targeting ANGPTL4 could be an effective treatment strategy to directly address the cellular causes of pancreatitis.

# Materials and Methods

### Animals

Male BALB/c mice (6 weeks) were obtained from Orient Bio. Animal Inc. (Gyeonggi-do, Korea). Animal care and experimental

procedures were conducted in accordance with the approval and guidelines of the Inha University-Institutional Animal Care and Use Committee (INHA IACUC, approval ID: 140812-325). All mice were housed in an environmentally controlled room (22 ± 2°C, 40–60% humidity and a 12-h light cycle. ANGPTL4 WT and ANGPTL4$^{-/-}$ (knockout, KO) mice were obtained as previously described (Zhu *et al*, 2011). WT mice and heterozygous mice for ANGPTL4 (C57/B6 background, 6–8 weeks) were obtained from the Mutant Mouse Regional Resource Center (MMRRC, Davis, CA, USA), an NIH-funded strain repository, and were donated to the MMRRC by Genentech. Animals for experiments were randomly allocated in a double-blinded manner. The number of mice used in each experiment is described in each figure legend.

## Human tissue and serum

Sera from human patients with pancreatitis (*n* = 90) and pancreatitis tissues (*n* = 10) were from Korea Biobank Network and Inha University Hospital (Korea). Samples from non-tumor normal region of pancreatic cancer patients were obtained during surgery. We performed a pancreatic cancer mutation analysis to confirm that pancreatic samples obtained from non-tumor normal region were real normal tissue. Sanger sequencing confirmed that there were no representative mutations in *KRAS* and *TP53* genes. Immunostaining for ANGPTL4 was performed using normal region with pancreatitis (*n* = 6) and without pancreatitis (*n* = 4). All investigations have been approved by the Inha University Hospital Human Research Ethics Committee (approval ID: 14-051) and were carried out in accordance with the Declaration of Helsinki, and all study participants provided witnessed written informed consent before entering the study.

## Establishment for pancreatitis mouse model

Male BALB/c mice (6 weeks) were fasted for 12 h with free access to water. AP was induced by 5 hourly intraperitoneal (IP) injections of cerulein (Sigma, St. Louis, MO, USA) at 50 μg/kg body weight. Mice were sacrificed 2 or/and 6 h after the last injection of cerulein. SAP was induced by cerulein and additional injection of lipopolysaccharide (LPS, IP 20 mg/kg). To determine whether SAP was induced by mouse recombinant ANGPTL4 (R&D Systems, Minneapolis, MN), low dose (2 mg/kg) instead of LPS was injected with cerulein. As a control, mice were also injected with ANGPTL4 (4 mg/kg) without cerulein and LPS. The AP+ANGPTL4 group was induced by one injection of ANGPTL4 (IP, 2 mg/kg) 2 h after AP induction. The ANGPTL4 group was induced by twice injection of ANGPTL4 (IP) at a dosage of 2 mg/kg body weight. Another AP model was induced by the administration of two intraperitoneal injections of L-arginine at a concentration of 4 g/kg body weight with 1-h interval between injections. In the supplementary results, a SAP 2 (0.6 mg) model was induced via 12 injections (IP) of cerulein. Likewise, the ANGPTL4 (0.6 mg) group was induced via 12 injections (IP) of ANGPTL4 at a dosage of 50 μg/kg, which is same dose for cerulein for the induction of SAP 2. Mice were sacrificed 12 h after pancreatitis induction. Neutralizing ANGPTL4, C5a, and isotype antibodies (10 mg/kg, IP) were injected once before the induction of AP and SAP. Animals were treated twice with a neutralizing antibody injection after the induction of pancreatitis. Blood

samples were centrifuged at 4°C (250 *g*, 10 min), and serum was stored at −20°C for further studies. The pancreas was immediately removed in a standardized fashion and divided into portions for the following assays.

## Determination of amylase and lipase activities

Amylase activity was assessed with a commercial kit (Bioassay, Hayward, CA, USA) based on the use of cibachron blue–amylose as a chromogenic substrate. The soluble chromogen in 0.1 ml of serum was measured spectrophotometrically at 580 nm. The absorbance was linear compared with the enzyme activity. The plasma lipase activity was also determined using a commercial kit (Bioassay), following the manufacturer's instructions. The titrimetric method is based on the degradation of triolein by lipase and the consequent release of diacetyl glycerol, which leads to the formation of hydrogen peroxide ($H_2O_2$). The latter reacts with a leuco dye, resulting in the formation of a chromophore that can be measured colorimetrically at 412 nm.

## Histopathology

Pancreas samples were fixed in 10% buffered formaldehyde, embedded in paraffin, and sectioned. The 8-μm-thick sections were stained with hematoxylin and eosin (H&E) for routine histology. For H&E staining, sections were stained with hematoxylin for 3 min, washed, and stained with 0.5% eosin for an additional 3 min. After a washing step with water, the slides were dehydrated in 70, 95, and 100% ethanol, and then in xylene. Terminal deoxynucleotidyl transferase–mediated deoxyuridine triphosphate nick-end labeling (TUNEL) staining was performed using the ApopTag peroxidase *in situ* apoptosis detection kit (Millipore, Billerica, MA, USA) according to the manufacturer's protocol.

## Enzyme-linked immunosorbent assay (ELISA)

Mouse ELISA kits (R&D Systems) were used for the analysis of TGF-β, TNF-α, IFN-γ, and C5a in serum and culture media according to manufacturer's recommendations. The plates were coated overnight with 2 or 4 μg/ml anti-TGF-β, anti-TNF-α, anti-IFN-γ, and anti-C5a capture monoclonal antibodies and washed with phosphate-buffered saline (PBS) Tween-20. Then, biotin-labeled (1 or 2 μg/ml) anti-TGF-β, anti-TNF-α, anti-IFN-γ, and anti-C5a detecting antibodies were used and developed using streptavidin–horseradish peroxidase (Vector, Burlingame, CA, USA) and 2, 2-azino-bis substrate (Sigma). ANGPTL4 ELISA kits for human and mouse were purchased from R&D systems and Abcam (Cambridge, UK) and MyBioSource (San Diego, CA, USA), respectively. PPAR-γ ELISA kits and PPAR-γ transcription factor assay kit were from MyBioSource and Abcam, respectively.

## Immunohistochemistry and immunofluorescence

Immunostaining was performed on 8-μm-thick sections after deparaffinization. Microwave antigen retrieval was performed in citrate buffer (pH 6.0) for 10 min before peroxidase quenching with 3% $H_2O_2$ in PBS for 10 min. Then, the sections were washed in water and pre-blocked with normal goat or rabbit serum for 10 min.

For primary antibody reaction, slides were incubated for 1 h at room temperature in a 1:100 dilution of antibody. The sections were then incubated with biotinylated secondary antibodies (1:500) for 1 h. After washing with PBS, streptavidin–horseradish peroxidase was applied. Finally, the sections were developed with a diaminobenzidine tetrahydrochloride substrate (DAB) for 10 min and then counterstained with hematoxylin. For immunofluorescence, 1:50 dilution for primary antibodies was used. After washing twice with PBS, the slides were incubated with fluorescein-labeled secondary antibody (1:100, Dianova) in antibody dilution solution for 1 h at room temperature in the dark. The nuclei were stained with 4,6-diamidino-2-phenylindole (DAPI) in the dark for 30 min at room temperature. The slides were subsequently washed twice with PBS, covered with DABCO (Sigma-Aldrich), and examined by confocal laser scanning microscopy (Olympus, Tokyo, Japan) at 488 and 568 nm. At least five random fields from each section were examined at a magnification of $200 \times$ and analyzed using a computer image analysis system (Media Cybernetics, Silver Spring, MD, USA).

## Western blotting

The pancreas and pancreatic acinar cells were lysed with a radioimmunoprecipitation assay (RIPA) buffer (Biosesang, Gyeonggi-do, Korea) containing protease and phosphatase inhibitor cocktails (GenDepot, Barker, TX). The proteins were resolved by sodium dodecyl sulfate–polyacrylamide gel electrophoresis (SDS-PAGE) and transferred onto nitrocellulose membranes. The blots were immunostained with the appropriate primary antibodies followed by secondary antibodies conjugated to horseradish peroxidase. Antibody binding was detected with an enhanced chemiluminescence reagent (Bio-Rad. Hercules, CA, USA). The primary antibodies against the following factors were used: ANGPTL4 (Thermo Fisher Scientific, Waltham, MA, USA, cat# 40-9800), GAPDH, and β-actin (Santa Cruz Biotechnology, Dallas, TX, USA, cat# sc-47724, and sc-47778). The secondary antibodies were purchased from Santa Cruz Biotechnology.

## qRT-PCR

RNA was incubated in the presence of poly (A) polymerase (PAP; Takara, Kusatsu, Japan), $MnCl_2$, and ATP for 1 h at 37°C. Then, reverse transcription was performed using an oligodT primer harboring a consensus sequence, on total RNA (5 µg) with SuperScript II RT (Invitrogen, Carlsbad, CA, USA). Next, the cDNA was amplified by RT-PCR; SYBR green qRT-PCR was performed on a Step ONE Plus (Applied Biosystems, Forster City, CA, USA). In each run, 1 µl of cDNA was used as template for amplification per reaction. The sample was added to 19 µl of reaction mixture containing 7 µl $H_2O$, 10 µl QuantiTect® SYBR® Green PCR Master Mix (Qiagen, Hilden, Germany) and 1 µl forward and reverse primers (Appendix Table S1). Real-time qRT-PCR amplification of the genes was carried out for 40 cycles of 95°C for 15 s, and 59°C for 1 min after 95°C for 15 min. Three independent experiments were performed.

## Isolation of primary pancreatic acinar cells and macrophages

The pancreas was isolated, washed twice with ice-cold Hank's balanced salt solution (HBSS) media, minced into 1–3 mm³ pieces, digested with collagenase IA solution (HBSS 1× containing 10 mM HEPES, 200 U/ml of collagenase IA, and 0.25 mg/ml of trypsin inhibitor), and incubated for 20–30 min at 37°C. The collagen digestion was stopped by adding an equal volume of ice-cold buffered washing solution (HBSS 1× containing 5% FBS and 10 mM HEPES). The digested pancreatic pieces were washed twice with the washing solution. The supernatant from this cell suspension contained acinar cells. Acinar cells were then pelleted (250 g, for 2 min at 4°C) and resuspended in 12 ml Waymouth complete media (2.5% FBS, 1% penicillin-streptomycin mixture, 0.25 mg/ml of trypsin inhibitor, and 25 ng/ml of epidermal growth factor). The cell mixture was filtered using a 100-µm filter. The isolated acini were seeded in a six-well culture dish (2 ml per well) and cultured at 37°C under a 5% (v/v) $CO_2$ atmosphere. After 24 h, the acini (in suspension) were transferred into a type I collagen-coated 6-well culture dish and cultured at 37°C under 5% (v/v) $CO_2$ atmosphere. The primary murine macrophages were isolated as previously described[58]. Briefly, mice were intraperitoneally injected with 2 ml of 3–5% aged thioglycollate solution. On day 5 after the injection, peritoneal macrophages were collected through a single injection of 10 ml RPMI 1640 containing 10% FBS into the peritoneal cavity. The peritoneal exudate was incubated with RBC lysis buffer on ice for 5 min and subsequently washed with RPMI 1640 media supplemented with 10% FBS and antibiotics, before being seeded onto tissue culture dishes. After 2 h, the cells were washed twice with PBS to remove the non-adherent cells and then provided with medium.

## Transwell migration assay

Pancreatic acinar cells were plated on 6-well dishes and cultured for 24 h. Culture supernatants were collected by centrifugation twice at 500 g for 5 min. Transwell migration assays were performed with 500 µl cell supernatant in the lower chamber and 250 µl macrophages ($2 \times 10^6$ cells/ml) in the upper chamber/insert of a 24-well plate with 3-µM pore-size membrane (Nunc, Rochester, NY, USA). Macrophages were then stimulated with CCK, LPS, and ANGPTL4 (100 ng/ml) for 24 h at 37°C. The cells that migrated into the lower chamber were counted with a hemocytometer.

## GdCl₃ macrophage depletion model

Six-week-old mice (BALB/c, male) were injected with $GdCl_3$ (15 mg/kg, IP) on every 2 days for 1 week. After 2 weeks, the animals were once again treated with $GdCl_3$, every 2 days for 1 week. At the endpoint (week 4), pancreatitis was induced for 24 h.

## Transfection of siRNA

siRNA-control (siCon), and ANGPTL4 siRNA 1 and 2 (siANGPTL4) were purchased from Dharmacon (On-TARGET plus-mouse SMART-pool ANGPTL4, L-56622-00-0010) and Invitrogen (Stealth mouse ANGPTL4 siRNA MSS226640), respectively. Cells were transfected with siRNA using Lipofectamine RNAiMAX transfection reagent (Invitrogen) according to the manufacturer's protocols.

## Generation of ANGPTL4-overexpressing macrophages

RAW 264.7 macrophages were obtained from the Korean Cell Line Bank (Seoul National University, Seoul, Republic of Korea).

**The paper explained**

**Problem**

Acute pancreatitis is an inflammatory disorder of the pancreas that is associated with a high mortality rate. Cytokines and chemokines released from the damaged pancreatic cells attract inflammatory cells; the disease severity is determined by the systemic response of inflammatory cells and cytokines/chemokines. Also, activation of leukocytes such as macrophages can occur before the development of necrosis. However, the etiological factors involved in the initiation and aggravation of acute pancreatitis are still poorly understood, and the treatment of the disease is limited to supportive therapies. Therefore, efforts to understand the pathophysiology of this disorder and develop clinical strategies to attenuate pancreatitis progression are imperative.

**Results**

Our works revealed that ANGPTL4 is highly increased in serum and tissue of patients with pancreatitis and accelerates the process of acute pancreatitis severity. ANGPTL4$^{-/-}$ mice showed decreased pancreatitis-associated pathological phenomena, cytokine levels, and apoptotic cell death in pancreatitis models. We found that ANGPTL4 plays a proinflammatory role by macrophage activation in acute pancreatitis. Also, macrophage activation by ANGPTL4 increases complementary component 5a (C5a) through PI3K/AKT signaling, leading to massive release of inflammatory cytokines, resulting in increased pancreatitis severity by injuring acinar cells via C5areceptor (C5aR). Furthermore, neutralizing antibody against ANGPTL4 alleviated macrophage activation and improved pancreatitis severity.

**Impact**

Our finding that ANGPTL4 exacerbates acute pancreatitis provides a clue to better understanding for pancreatitis severity. Furthermore, ANGPTL4 might represent a therapeutic target for acute pancreatitis.

ANGPTL4-overexpressing RAW 264.7 macrophages were generated by stable transduction of ANGPTL4 retro-viral particles using BMN-I-GFP vector. Briefly, RAW 264.7 macrophages were seeded in 6 well plates and then transfected with ANGPTL4 retro-viral particles in the presence of Polybrene (5 μg/ml). WT retro-viral particle was used as a control. After 3 days, the cells transfected with ANGPTL4 retro-viral particles were selected by GFP using FACS sorter (Bio-Rad).

### AP Induction by ANGPTL4-overexpressed macrophage

To investigate direct effect of ANGPTL4 in macrophages on the progression of AP, ANGPTL4-overexpressing macrophages ($5 \times 10^5$ cells/mouse) were injected into pancreas. The mice were sacrificed on day 1, 3, and 5. The pancreas of AP mice were excised, fixed in neutral buffered formalin, and paraffinized. And then, pancreas tissue slides were stained with C5a, F4/80, amylase, and C5aR.

### Bone marrow-derived macrophage (BMDM)

AP and SAP were induced as described above. Mice were sacrificed by cervical dislocation, then the abdomen and hid legs were sterilized with 70% ethanol and the femurs were dissected using scissors. The bones were placed in 70% ethanol for 1 min, washed in sterile RPMI 1640 medium and then both ends were removed using sterile scissors and forceps. The bones were flushed with a syringe filled with RPMI 1640 to extrude bone marrow into a 15-ml sterile polypropylene tube. A 5-ml plastic syringe and a 25-gauge needle were used to gently resuspend the bone marrow. The single-cell suspension generated thereafter is called fresh bone marrow cells. After that, the cells were resuspended in 10 ml bone marrow differentiation media (R20/30), which is RPMI1640 containing 20% FBS, 30% LCCM, 100 U/ml penicillin, 100 μg/ml streptomycin, and 2 mM L-glutamine. Cells were seeded in Petri dishes (BD Biosciences, San Jose, CA, USA) and incubated at 37°C in a 5% $CO_2$ humidified chamber for 4 days. Next, an extra 10 ml of R20/30 were added to dishes and incubated for an additional 3 days. To establish the BMDM, the culture medium was removed, and the attached cells were rinsed with 10 ml of sterile PBS. The BMDMs were detached by gently pipetting, collected, and centrifuged at $200 \times g$ for 5 min.

Cell pellets were resuspended in 10 ml of BMDM culture medium (R10/5), composed of RPMI 1640, 10% fetal bovine serum, 5% LCCM, and 2 mM L-glutamine. The BMDMs were counted, seeded, and cultured in tissue culture plates 12 h before any further experimental procedure.

### Macrophage differentiation from THP1 monocyte

Human monocytic THP-1 cells were grown in culture in RPMI 1640 (Gibco) culture medium containing 10% of heat-inactivated FBS (Gibco), 10 mM HEPES (Gibco), 100 U/ml penicillin, and 100 μg/ml streptomycin. THP-1 monocytes at $1 \times 10^5$/ml were allowed to differentiate into macrophages by 48 h incubation with 100 nM phorbol 12-myristate 13-acetate (PMA, Sigma).

### ANGPTL4 antibody

ANGPTL4 antibody was prepared as previously described (Goh et al, 2010). Briefly, mice were immunized with adjuvant conjugated-cANGPTL4. The spleen of the mouse was then removed, and a single-cell suspension was prepared. These cells were fused with myeloma cells and cultured in hybridoma selection medium (HAT; Gibco, Waltham, MA, USA). The fused cells were cultured in microtiter plates with peritoneal macrophages for 48-h post-fusion (2–$4 \times 10^6$ cells/ml). mAbs in medium were first screened using ELISA to identify the positive clones. Positive clones were expanded and recloned by a limiting dilution technique to ensure monoclonality. The antibodies were purified by Protein A affinity chromatography as recommended by the manufacturer (GE Healthcare).

### Statistical analysis

Data were expressed as the mean $\pm$ SEM. The data were analyzed using unpaired $t$-test and Mann–Whitney $U$-test for two-group, and ANOVA with Tukey's post hoc tests for multiple comparisons, where $P < 0.05$ was considered statistically significant. Statistical calculations were performed using the SPSS software for Windows (version 10.0; SPSS, Chicago, IL, USA). In animal study, sample size was determined based on previous experiments performed in our laboratory. The evaluator was blinded to the identity of a specific sample as far as the nature of the experiment allowed it.

## Data availability

This study includes no data deposited in external repositories.

**Expanded View** for this article is available online.

## Acknowledgements

This research was supported by National Research Foundation of Korea (2019M3E5D1A02069621, 2018R1A2A1A05077263, 2019R1I1A1A01060699, 2014009392).

## Author contributions

Experiment performance: KHJ; MKS; HHY; ZF; JK; SJK; JHP; JEL; Y-CY; MSS; BSH; SK; ZT; JWKW. Experiment design: KHJ; SSH; NST. Data analysis: KHJ; MKS; HHY; ZF; JK; SJK; JHP; JEL; Y-CY; MSS; BSH; SK; ZT; JWKW; JHL; YJS; D-HL. Manuscript writing: KHJ; S-SH; NST. Manuscript editing: JHL; YJS; D-HL.

## Conflict of interest

The authors declare that they have no conflict of interest.

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
