## [Review Process File · EMBO Molecular Medicine]

ANGPTL4 exacerbates pancreatitis by augmenting acinar cell injury through upregulation of C5a.

Kyung Hee Jung, Mi Kwon Son, Hong Hua Yan, Zhenghuan Fang, Juyoung Kim, Soo Jung Kim, Jung Hee Park, Ji Eun Lee, Young-Chan Yoon, Myeong Seong Seo, Beom Seok Han, Soyeon Ko, Young Ju Suh, Ju Han Lim, Don-Haeng Lee, Ziqiang Teo, Wei Kiat Jonathan Wee, Nguan Soon TAN, and Soon-Sun Hong

DOI: [10.15252/emmm.201911222](https://doi.org/10.15252/emmm.201911222)

Corresponding author(s): *Soon-Sun Hong (hongs@inha.ac.kr)*, *Nguan Soon TAN (NSTAN@ntu.edu.sg)*

Review Timeline:

Submission Date:	29th Jul 19
Editorial Decision:	13th Sep 19
Revision Received:	22nd Jan 20
Editorial Decision:	11th Feb 20
Revision Received:	11th May 20
Editorial Decision:	14th May 20
Revision Received:	20th May 20
Editorial Decision:	26th May 20
Revision Received:	29th May 20
Accepted:	29th May 20

Editor: *Jingyi Hou*

Transaction Report:

13th Sep 2019

Thank you for the submission of your manuscript to EMBO Molecular Medicine. We have now received feedback from 2 out of 3 reviewers whom we asked to evaluate your manuscript. Given that both reviewers provide similar recommendations, we prefer to make a decision now in order to avoid further delay in the process.

As you will see from the reports below, the referees acknowledge the potential interest of the study. However, they also raise substantial concerns about your work, which should be convincingly addressed in a major revision of the present manuscript. In particular, the link between C5a and ANGPTL4 (and the underlying mechanism as highlighted by referee #1) needs to be strengthened. The potential impact of ANGPTL4 on plasma triglycerides (as commented on by referee #2) should be addressed. Further, as pointed out by the referees, the clarity in data/study presentation needs to be improved, more details and information must be provided, along with better controlled experiments, to improve conclusiveness and clarity.

We would welcome the submission of a revised version within three months. Please note that EMBO Molecular Medicine strongly supports a single round of revision and that, as acceptance or rejection of the manuscript will depend on another round of review, your responses should be as complete as possible.

I look forward to receiving your revised manuscript.

Yours sincerely,

Jingyi Hou

Jingyi Hou
Editor
EMBO Molecular Medicine

*** Instructions to submit your revised manuscript ***

** PLEASE NOTE ** As part of the EMBO Publications transparent editorial process initiative (see our Editorial at <https://www.embopress.org/doi/pdf/10.1002/emmm.201000094>), EMBO Molecular Medicine will publish online a Review Process File to accompany accepted manuscripts.

To submit your manuscript, please follow this link:

Link Not Available

- 1) a .doc formatted version of the manuscript text (including Figure legends and tables). Please make sure that the changes are highlighted to be clearly visible to referees and editors alike.
- 2) separate figure files*
- 3) supplemental information as Expanded View and/or Appendix. Please carefully check the authors guidelines for formatting Expanded view and Appendix figures and tables at <https://www.embopress.org/page/journal/17574684/authorguide#expandedview>
- 4) a letter INCLUDING the reviewers' reports and your detailed responses to their comments (as Word file)

Also, and to save some time should your paper be accepted, please read below for additional information regarding some features of our research articles:

- 5) The paper explained: EMBO Molecular Medicine articles are accompanied by a summary of the articles to emphasize the major findings in the paper and their medical implications for the non-specialist reader. Please provide a draft summary of your article highlighting
 - the medical issue you are addressing,
 - the results obtained and
 - their clinical impact.

This may be edited to ensure that readers understand the significance and context of the research.

Please refer to any of our published articles for an example.

6) For more information: There is space at the end of each article to list relevant web links for further consultation by our readers. Could you identify some relevant ones and provide such information as well? Some examples are patient associations, relevant databases, OMIM/proteins/genes links, author's websites, etc...

7) Author contributions: the contribution of every author must be detailed in a separate section (before the acknowledgments).

8) EMBO Molecular Medicine now requires a complete author checklist (<https://www.embopress.org/page/journal/17574684/authorguide>) to be submitted with all revised manuscripts. Please use the checklist as a guideline for the sort of information we need WITHIN the manuscript as well as in the checklist. This is particularly important for animal reporting, antibody dilutions (missing) and exact p-values and n that should be indicated instead of a range.

9) Every published paper now includes a 'Synopsis' to further enhance discoverability. Synopses are displayed on the journal webpage and are freely accessible to all readers. They include a short stand first (maximum of 300 characters, including space) as well as 2-5 one sentence bullet points that summarise the paper. Please write the bullet points to summarise the key NEW findings. They should be designed to be complementary to the abstract - i.e. not repeat the same text. We encourage inclusion of key acronyms and quantitative information (maximum of 30 words / bullet point). Please use the passive voice. Please attach these in a separate file or send them by email, we will incorporate them accordingly.

You are also welcome to suggest a striking image or visual abstract to illustrate your article. If you do please provide a jpeg file 550 px-wide x 400-px high.

10) A Conflict of Interest statement should be provided in the main text

11) Please note that we now mandate that all corresponding authors list an ORCID digital identifier. This takes <90 seconds to complete. We encourage all authors to supply an ORCID identifier, which will be linked to their name for unambiguous name identification.

Currently, our records indicate that the ORCID for your account is 0000-0001-7679-1388.

Please click the link below to modify this ORCID:
Link Not Available

12) The system will prompt you to fill in your funding and payment information. This will allow Wiley to send you a quote for the article processing charge (APC) in case of acceptance. This quote takes into account any reduction or fee waivers that you may be eligible for. Authors do not need to pay any fees before their manuscript is accepted and transferred to our publisher.

Graphs 800-1,200 DPI
Photos 400-800 DPI
Colour (only CMYK) 300-400 DPI"

*Additional important information regarding figures and illustrations can be found at <http://bit.ly/EMBOPressFigurePreparationGuideline>

***** Reviewer's comments *****

Referee #1 (Comments on Novelty/Model System for Author):

None

Referee #1 (Remarks for Author):

The study by Jung and colleagues provides evidence for a novel pathway controlling acinar damage during acute pancreatitis (AP), a dismal disease in great need for new treatment approaches based on a better understanding of the molecular events participating in the pathogenesis of this disease. The data included in this version of the manuscript is well presented and for most part supportive of the main conclusions of the study. However, additional experimentation is needed to fully support them. See below specific comments.

- 1) The description of sample acquisition and institutional approval for these 6 patients is missing. What were the patient conditions that allowed access to these samples?
- 2) The quality of the IFs is quite low specially for human specimens. The images are saturated and it is difficult to evaluate the localization of the signal. These must be improved.
- 3) The RNAi experiments should be repeated using second targeting sequence and the knockdown experiment should be evaluated at the protein level.
- 4) About the mechanism of upregulation of ANGPTL4, is the same one as the other proinflammatory cytokines?
- 5) The role of PPAR γ should be further explored. The data presented does not completely rule out a role for this molecule in the AP processes regulated by ANGPTL4. Knockdown experiments in the presence of an active ANGPTL4 are essential.
- 6) The levels of PPAR γ activity as well as C5a should be determined in the ANGPTL4 null animals.
- 7) The mechanism regulating the increase levels of C5a must be explored. This is crucial to fully define the role of ANGPTL4 in AP as well as design future translational efforts for this disease.

Referee #2 (Comments on Novelty/Model System for Author):

My two major concerns are 1) the complete disregard-experimentally and in writing-of the ability of ANGPTL4 to raise plasma triglycerides, which is a very important risk factor for pancreatitis. 2) the

lack of proper controls in a number of key experiments. For instance, the paper shows that the serum of ANGPTL4^{-/-} mice contains substantial levels of ANGPTL4. These two issues make me quite wary and form the basis of my recommendation to reject the manuscript. The paper is also a very tough read and the evidence is not presented and described in a straightforward and logical fashion. Because the paper has potential after extensive revision, the opportunity to resubmit can be considered.

Referee #2 (Remarks for Author):

The paper describes a novel stimulatory effect of ANGPTL4 on pancreatitis. A very fascinating and novel finding of this paper is that the severity of pancreatitis is markedly reduced in the absence of ANGPTL4 in two mouse models of pancreatitis. In addition, the work performed with ANGPTL4 neutralizing antibody is highly interesting. However, the proposed relationship between ANGPTL4, C5a, and macrophages remains relatively vague. It certainly does not help that the paper is difficult to read and the studies are often not presented in a logical and straightforward manner and order.

Major comments

1) It is very unfortunate that the paper does not mention an important rationale for studying the potential impact of ANGPTL4 on pancreatitis. ANGPTL4 raises plasma triglycerides. Elevated plasma triglycerides has been unequivocally established as key risk factor for pancreatitis. Accordingly, it is reasonable to suggest that ANGPTL4 may promote pancreatitis by raising plasma triglycerides. This line of reason is completely missing from the manuscript and should be addressed experimentally.

2) The evidence presented suggests that ANGPTL4 induces various cytokines in acinar cells and vice versa that various cytokines induce ANGPTL4. However, the model presented in figure 7 only describes the former mechanism. Please elaborate on the chain of events by which ANGPTL4 is suspected to influence pancreatitis. This aspect of the paper is very confusing (Supplemental figures, Figure 5A-C). The paper follows a very convoluted and partly illogical path to arrive at the hypothesis that ANGPTL4 could directly regulate the progression of pancreatitis through macrophages.

3) What is missing from the paper and what would be more useful that the analysis pursued with exogenous ANGPTL4 is a comparative cytokine profiling between WT and Angptl4^{-/-} macrophages. It is important to find out if C5a production is reduced in Angptl4^{-/-} macrophages (peritoneal or bone marrow derived). Currently, the link between ANGPTL4 and C5a is only established in vitro using external ANGPTL4 or overexpression systems.

4) The finding that serum ANGPTL4 levels are elevated in patients with pancreatitis should be confirmed using an established and validated ELISA from R&D Systems.

5) It is highly curious that the serum of ANGPTL4^{-/-} mice contain substantial levels of ANGPTL4. This results strongly indicates that the Elisa is not specific for ANGPTL4.

6) The Western blots in figure 2 and S1C should indicate the molecular weight markers and contain a negative control for ANGPTL4 (serum and tissue from ANGPTL4^{-/-} mice). This is absolutely necessary as the availability of effective and specific anti-ANGPTL4 antibodies has been a huge problem. I thus must insist on the use of proper controls. The Western blot in figure 4B does contain a proper negative control. However, ANGPTL4 is not completely absent in the ANGPTL4^{-/-} SAP group. Please comment.

7) The manuscript contains several graphs based on traditional RT-PCR (Figure 5D, 6C). This is a bit outdated and should be replaced by quantitative real-time PCR.

Minor comments

1) Reference Oster et al. should be Köster et al.

2) Some of the references cited are out of date or improperly cited. Authors are encouraged to use a more recent review than Oike (2005). Kersten (2000) did not identify the PPAR response element in the human and mouse ANGPTL4 gene.

3) Page 4. Please adjust description "Herein, we provide evidence and underlying mechanisms that ANGPTL-4 leads to pancreatitis." Replace "leads to" by "promotes".

4) Page 5. "A 4-fold increase in ANGPTL4 level was observed in pancreatitis patients, with increases in amylase, lipase and CRP (Fig 1A)." . Please specify that ANGPTL4 levels were determined in serum. There is tendency throughout the manuscript to omit this type of relevant information, which renders the manuscript difficult to read.

5) The manuscript needs substantial English language editing.

6) According to the methods section, "In case of human samples, human sample sizes achieved 99.99% power to detect a difference between both groups." How much of a difference? Moreover, the number of 99.99% is hard to believe. Please check and correct.

7) Figure 3B, graph for IL1beta appears to contain a mistake. It is very difficult to imagine that the differences between SAP and ANGPTL4 4mg is statistically significant. Please verify.

8) In the model presented in figure 7E, it is not specified what is the source of the ANGPTL4. Could it be liver derived ANGPTL4? Or ANGPTL4 produced by macrophages?

9) It is unclear how the authors arrived at the dose for ANGPTL4 injection in mice. Did ANGPTL4 injection result in elevation of serum ANGPTL4 levels? This information should be provided.

10) It is unclear why ANGPTL4 was injected in the AP animals but not the SAP animals. Please clarify.

11) The observation that expression of CPT1A, PDK4, and perilipin-2 are not changed in the AP and SAP models cannot be used to argue that the increase in ANGPTL4 in pancreatitis is not mediated by PPAR- γ . I would recommend removing these data.

12) Page 9. "Therefore, we supposed that the effect of ANGPTL4 on cytokines would be rather large." It is unclear how the authors arrived at this statement.

13) The supplementary data should preferably be reserved for supplemental data, not for key data that form an essential part of the narrative of the paper. The authors are encouraged to reconsider the distribution of data across regular and supplemental figures.

14) The observation that expression of C5a and C5aR expression were decreased in the ANGPTL4

-/- AP and SAP models does not point to a cause and effect relationship between ANGPTL4 and C5a but could simply reflect the general decrease in inflammation in the absence of ANGPTL4. In other words, there is no evidence that the relation between ANGPTL4 and C5a is specific. Please comment.

15) Please specify what control was used for the ANGPTL4-neutralizing antibody.

Answers to Reviewers' Comments

Date: JAN 12, 2020 Ms. No.: EMM-2019-11222

Authors: Jung *et al.*

Title: ANGPTL4 functions as a pancreatitis inducer by augmenting acinar cell injury via upregulating C5a”

We thank the reviewers for their thoughtful comments, time, and effort they put in to the review of our manuscript. We strongly believe it will improve the quality of our manuscript a lot. We tried our best efforts to make necessary corrections and additions of experimental data as the reviewers suggested. Our responses to the comments and our explanations for this revised manuscript are described below.

Comment from Reviewers

Reviewer 1

The study by Jung and colleagues provides evidence for a novel pathway controlling acinar damage during acute pancreatitis (AP), a dismal disease in great need for new treatment approaches based on a better understanding of the molecular events participating in the pathogenesis of this disease. The data included in this version of the manuscript is well presented and for most part supportive of the main conclusions of the study. However, additional experimentation is needed to fully support them. See below specific comments.

Point 1

The description of sample acquisition and institutional approval for these 6 patients is missing. What were the patient conditions that allowed access to these samples?

➤ **Our response**

As pointed out, we added information such as acquisition and institutional approval for these 6 patients in material and method section. As you know, it is difficult to obtain pancreas tissue from pancreatitis patients without surgery. For this reason, in some studies (1), immuno-staining together with DNA and RNA analysis for pancreatitis has been performed in non-tumor normal samples with pancreatitis. Likewise, we obtained samples of non-tumor normal region from pancreatic cancer patients during surgery, and then performed immuno-staining for ANGPTL4 in pancreas samples of normal region with pancreatitis (n=6) and normal region without pancreatitis (n=4) that were evaluated pathologically.

New manuscript: page 22, line 4-6

Indeed, we obtained samples of non-tumor normal region from pancreatic cancer patients during surgery, and then performed immuno-staining for ANGPTL4 in pancreas samples of normal region with pancreatitis (n=6) and normal region without pancreatitis (n=4) that were evaluated pathologically. Informed consent was obtained from all patients, and the experimental protocol was approved by the Inha University Hospital Human Research Ethics Committee (approval ID: 14-051).

Reference

1. Shi C, Merchant N, Newsome G et al., Differentiation of pancreatic ductal adenocarcinoma from chronic pancreatitis by PAM4 immunohistochemistry. Arch Pathol Lab Med. 2014;138(2):220-8

Point 2

The quality of the IFs is quite low specially for human specimens. The images are saturated and it is difficult to evaluate the localization of the signal. These must be improved.

➤ Our response

As indicated, we conducted immunofluorescence staining again for human specimens and replaced this result with new data of Figure 1B.

Point 3

The RNAi experiments should be repeated using second targeting sequence and the knockdown experiment should be evaluated at the protein level.

➤ **Our response**

As you pointed out, we performed siRNA experiments repeatedly and confirmed at protein level. We replaced it with this result in Fig 6E.

Point 4

About the mechanism of upregulation of ANGPTL4, is the same one as the other proinflammatory cytokines?

➤ **Our response**

ANGPTL4, a secreted protein, was initially discovered in 2000 by three independent research

groups as a regulator of lipid metabolism-induced by various factors including peroxisome proliferator activated receptor γ (PPAR γ) under fasting conditions (1-3). Recent studies have shown that ANGPTL4 regulates tumorigenesis, angiogenesis, vascular permeability, glucose homeostasis, cell differentiation, wound healing, and inflammation regulation (4-5). Therefore, its dysregulation contributes to the pathogenesis of diseases.

Until now, numerous evidences have indicated that ANGPTL4 is involved in many inflammation-associated diseases. Indeed, ANGPTL4 was induced by cytokines such as IL-1 β , IL-6, and TNF- α in osteoblast cells (6). Also, increased expression of ANGPTL4 in osteoarthritis has been reported to be potential factor in pathogenic cartilage destruction (7,8).

One study identified that ANGPTL4 is upregulated protein during bone marrow inflammatory conditions and plays a vital role during hematopoietic reconstitution after stem cell transplantation (9). Higher ANGPTL4 levels were reported after exposure to interleukins, TNF- α , and IFN- γ in 3T3-L1 adipocytes (10,11).

While, Lichtenstein *et al.* reported that ANGPTL4 can also protect against the severe pro-inflammatory effects of saturated fat by inhibiting fatty acid uptake by mesenteric lymph node macrophages (12). Similarly, ANGPTL4 confers protective effects against the development of atherosclerosis, which has been associated with atherogenesis and macrophage polarization (13). These results suggest that ANGPTL4 may exert both anti and a pro-inflammatory effect in a context-dependent manner. Despite numerous reports of the role of ANGPTL4 in inflammation, the mechanisms whereby ANGPTL4 modulates inflammation in various diseases remain largely unclear. Also, the exact function of ANGPTL4 has not been reported in pancreatitis.

Interestingly, our study showed that the increased ANGPTL4 was positively associated with the high C-reactive protein level in inflammation condition of pancreatitis patients, which are similar to findings of Tjeerdema *et al.* that serum ANGPTL4 was associated with the C-reactive protein level in metabolic disease (14). Also, we observed that ANGPTL4 induced massive release of cytokines by the increase of C5a through macrophage activation in pancreatitis.

Accordingly, it is thought that ANGPTL4 plays a role as a pro-inflammatory cytokine in pancreatitis in our study.

References

1. Kersten S, Mandard S, Tan NS et al., Characterization of the fasting-induced adipose factor FIAF, a novel peroxisome proliferator-activated receptor target gene. *J Biol Chem.* 2000;275(37):28488-93.
2. Kim I, Kim HG, Kim H et al., Hepatic expression, synthesis and secretion of a novel fibrinogen/angiopoietin-related protein that prevents endothelial-cell apoptosis. *Biochem J.*

2000;346 Pt 3:603-10.

3. Yoon JC, Chickering TW, Rosen ED et al., Peroxisome proliferator-activated receptor gamma target gene encoding a novel angiopoietin-related protein associated with adipose differentiation. *Mol Cell Biol.* 2000;20(14):5343-9.
4. Grootaert C, Van de Wiele T, Verstraete W et al., Angiopoietin-like protein 4: health effects, modulating agents and structure-function relationships. *Expert Rev Proteomics.* 2012;9(2):181-99.
5. Yang X, Cheng Y, Su G., A review of the multifunctionality of angiopoietin-like 4 in eye disease. *Biosci Rep.* 2018;13;38(5).
6. Noh JM, Shen C, Kim SJ et al., Interleukin-1 β increases Angptl4 (FIAF) expression via the JNK signaling pathway in osteoblastic MC3T3-E1 cells. *Exp Clin Endocrinol Diabetes.* 2015;123(8):445-60.
7. Wang W, Liu Y, Hao J, Zheng et al., Comparative analysis of gene expression profiles of hip articular cartilage between non-traumatic necrosis and osteoarthritis. *Gene.* 2016;591(1):43-47.
8. Mathieu M¹, Iampietro M, Chuchana P et al., Involvement of angiopoietin-like 4 in matrix remodeling during chondrogenic differentiation of mesenchymal stem cells. *J Biol Chem.* 2014;289(12):8402-12.
9. Schumacher A, Denecke B, Braunschweig T et al., Angptl4 is upregulated under inflammatory conditions in the bone marrow of mice, expands myeloid progenitors, and accelerates reconstitution of platelets after myelosuppressive therapy. *J Hematol Oncol.* 2015;8:64.
10. Lu B, Moser A, Shigenaga JK et al., The acute phase response stimulates the expression of angiopoietin like protein 4. *Biochem Biophys Res Commun.* 2010;391(4):1737-41.
11. Makoveichuk E, Vorrstö E, Olivecrona T et al., TNF- α decreases lipoprotein lipase activity in 3T3-L1 adipocytes by up-regulation of angiopoietin-like protein 4. *Biochim Biophys Acta Mol Cell Biol Lipids.* 2017;1862(5):533-540.
12. Lichtenstein L, Mattijssen F, de Wit NJ et al., Angptl4 protects against severe proinflammatory effects of saturated fat by inhibiting fatty acid uptake into mesenteric lymph node macrophages. *Cell Metab.* 2010;12(6):580-92.
13. Aryal B, Rotllan N, Araldi E et al., ANGPTL4 deficiency in haematopoietic cells promotes monocyte expansion and atherosclerosis progression. *Nat Commun.* 2016;7:12313.

14. Tjeerdema N, Georgiadi A, Jonker JT et al., Inflammation increases plasma angiopoietin-like protein 4 in patients with the metabolic syndrome and type 2 diabetes. *BMJ Open Diabetes Res Care*. 2014;2(1):e000034.

Point 5

The role PPARgamma should be further explored. The data presented does not completely rule out a role for this molecule in the AP processes regulated by ANGPTL4. Knockdown experiments in the presence of an active ANGPTL4 are essential.

➤ **Our response**

PPAR- γ is a ligand-activated nuclear receptor whose γ_2 form is highly expressed in white adipose tissue, where it regulates the expression of a number of genes involved in lipid metabolism, including LPL. The net result of PPAR- γ activation is a remodeling of WAT with a larger number of smaller adipocytes (1, 2). PPAR- γ regulates the expression of pro-inflammatory cytokines as well as gene involved in insulin sensitivity. For this reason, PPAR- γ is a main target of thiazolidinediones (TZDs), a class of drugs used to improve lipid and glucose metabolism in type 2 diabetes (3). Therefore, PPAR- γ activation also efficiently ameliorates insulin resistance and hypertriglyceridemia (4). Some studies have reported that activation of PPAR- γ induces the expression and secretion of ANGPTL4 in adipocyte lipid metabolism and tumor angiogenesis (5,6). However, the association with PPAR- γ and ANGPTL4 in pancreatitis has not been investigated. Therefore, we attempted whether ANGPTL4 could be regulated by PPAR- γ in pancreatitis. First, we evaluated level of PPAR- γ using serum and pancreas nucleic protein of ANGPTL4 WT and KO mice with pancreatitis (PPAR- γ ELISA kit and PPAR- γ transcription factor assay kit). As shown in below Figures, there was little difference in PPAR- γ levels between ANGPTL4 WT and KO mice. We also identified PPAR- γ transcription level in pancreas of ANGPTL4 WT and KO mice with pancreatitis. As a result, although there was a little change in level of PPAR- γ according to the severity of pancreatitis, there was no change between ANGPTL4 WT and KO mice.

Especially, there was little change in level of PPAR- γ between ANGPTL4 WT and KO mice in transcription factor binding assay. From these results, we think that there is little correlation between ANGPTL4 and PPAR- γ in pancreatitis.

However, as reviewer mentioned, to more confirm the role of PPAR- γ in regulation of ANGPTL4 in pancreatitis, we investigated the expression of ANGPTL4 after PPAR- γ knockdown in the presence of an active ANGPTL4 (macrophages). As shown in below Figures, we observed that PPAR- γ was not involved in induction of ANGPTL4 in pancreatitis. Collectively, we concluded that ANGPTL4 was not a target molecule for PPAR- γ and was not be regulated by PPAR- γ in pancreatitis although PPAR-

γ was reported to induce the expression and secretion of ANGPTL4 in adipocyte lipid metabolism.

References

1. Auwerx J, Schoonjans K, Fruchart JC et al., Regulation of triglyceride metabolism by PPARs: fibrates and thiazolidinediones have distinct effects. *J Atheroscler Thromb.* 1996;3(2):81-9.
2. Spiegelman BM. PPAR- γ : adipogenic regulator and thiazolidinedione receptor. *Diabetes.* 1998;47(4):507-14.
3. Bensinger SJ, Tontonoz P. Integration of metabolism and inflammation by lipid-activated nuclear receptors. *Nature.* 2008;454(7203):470-7.
4. Jones SL, Wiseman MJ, Viberti GC, Glomerular hyperfiltration as a risk factor for diabetic nephropathy: five-year report of a prospective study. *Diabetologia.* 1991;34(1):59-60.
5. Okamoto H, Cavino K, Na E et al., Angptl4 does not control hyperglucagonemia or α -cell hyperplasia following glucagon receptor inhibition. *Proc Natl Acad Sci U S A.* 2017;114(10):2747-2752.
6. Tian L, Zhou J, Casimiro MC et al., Activating peroxisome proliferator-activated receptor gamma mutant promotes tumor growth in vivo by enhancing angiogenesis. *Cancer Res.* 2009;69(24):9236-44.

Point 6

The levels of PPAR γ activity as well as C5a should be determined in the ANGPTL4 null animals.

➤ **Our response**

As mentioned above, we determined the levels of PPAR- γ activity in serum and tissue of ANGPTL4 KO and WT animals. Our study showed that there was a little change in level of PPAR- γ

according to the severity of pancreatitis but there was no change between ANGPTL4 WT and KO mice. Also, we identified expression of PPAR- γ in pancreas tissue of ANGPTL4 KO and WT animals. Likewise, we did not observe significant association of PPAR- γ expression between ANGPTL4 KO and WT animals with pancreatitis.

Until now, several studies have reported that ANGPTL4 was induced through regulation of PPAR- γ by fatty acid in fasting, lipid metabolism and metabolic diseases, suggesting that ANGPTL4 is a target factor for PPAR- γ (1,2). However, in present study, we showed that ANGPTL4 was not regulated by PPAR- γ in pancreatitis.

References

1. Alex S, Lange K, Amolo T et al., Short-chain fatty acids stimulate angiopoietin-like 4 synthesis in human colon adenocarcinoma cells by activating peroxisome proliferator-activated receptor γ . *Mol Cell Biol.* 2013;33(7):1303-16.
2. La Paglia L, Listì A, Caruso S et al., Potential Role of ANGPTL4 in the Cross Talk between Metabolism and Cancer through PPAR Signaling Pathway. *PPAR Res.* 2017; 8187235.

Point 7

The mechanism regulating the increased levels of C5a must be explored. This is crucial to fully define the role of ANGPTL4 in AP as well as design future translational efforts for this disease.

➤ **Our response**

ANGPTL4 is an orphan ligand. Until now, its receptor has not been found. ANGPTL4 was reported to

be regulated by PPAR-r and integrin *etc.* in adipocyte and cancer (1), but the exact function and mechanism of ANGPTL4 has not been unveiled.

Also, the complement system plays an important role in the recognition and clearance of pathogens. It consists of 30 soluble and membrane-bound proteins that are rapidly mobilized through a cascade of enzymatic reactions and participate in host defenses through a range of mechanisms, including direct killing of bacteria, facilitation of phagocytosis, the recruitment and activation of immune cells (2,3). Among the many components of the complement system, C5a is a potent chemoattractant for macrophages and neutrophils and directly acts on a number of parenchymal cells *via* binding to the C5a receptor (CD88) (4). Numerous studies suggest the importance of C5a in various inflammatory diseases. Indeed, increased production of C5a has been reported in several pathological states such as pancreatitis and RA (5,6). Similarly, in the present study, we observed that ANGPTL4 increased C5a in macrophage, leading to the increased severity of pancreatitis. However, there has never been reported for the associations of ANGPTL4 and C5a until now.

In order to investigate the mechanism by which ANGPTL4 increased C5a, the mRNA expression of C5, a preform of C5a, was first identified. When macrophage was treated with ANGPTL4, mRNA expression of C5 only showed a little increase in human ANGPTL4-treated THP1-derived macrophage cells. Next, we checked whether ANGPTL4 increased activity of the C5 convertase which cleaves C5 to C5a and C5b. C5 convertase was measured by the value of C5b-9. However, C5 convertase was highly not increased in ANGPTL4-treated THP1-derived macrophage cells.

Accordingly, we thought that ANGPTL4 would not directly regulate complement systems, but would increase C5a through cellular signaling pathways.

To evaluate whether ANGPTL4 increases C5a through regulation of cellular kinase signaling pathway in human macrophages (THP1), we performed a human phospho-kinase arrays analysis. As a result, ANGPTL4 increased the phosphorylation of AMPK α , JNK, AKT, and p70S6K in human THP1-derived macrophages. In same condition, we observed the increased expression of C5 by ANGPTL4 treatment. From these results, we speculated that ANGPTL4 may affect JNK, AMPK, and PI3K/AKT signaling.

To more confirm this result, each inhibitor (SP600125, compound C, and HS-173, respectively) was treated to THP1-derived macrophages after ANGPTL4 treatment. And then, we identified the expression of C5a and its related signaling. Among those, we observed that the PI3K inhibitor decreased the expression of C5a by inhibiting PI3K/AKT signaling. In some studies, ANGPTL4 has been reported to be regulated by PI3K/AKT pathway in endothelial cell, cancer cells, and stem cells (7-9). Although the exact mechanism by which ANGPTL4 increases C5a in macrophage was not revealed, it was concluded that ANGPTL4 partially increases C5a through PI3K/AKT signaling. However, further studies on the mechanism of C5a regulation by ANGPTL4 is needed.

References

1. La Paglia L, Listì A, Caruso S et al., Potential Role of ANGPTL4 in the Cross Talk between Metabolism and Cancer through PPAR Signaling Pathway. *PPAR Res.* 2017; 2017:8187235.
2. Monk PN, Scola AM, Madala P et al., Function, structure and therapeutic potential of complement C5a receptors. *Br J Pharmacol.* 2007;152(4):429-48.
3. Holers VM, Complement and its receptors: new insights into human disease. *Annu Rev Immunol.* 2014;32:433-59.
4. Kemper C, Pangburn MK, Fishelson Z, Complement nomenclature 2014. *Mol Immunol.* 2014;61(2):56-8.
5. Sendler M, Beyer G, Mahajan UM, Complement Component 5 Mediates Development of Fibrosis, via Activation of Stellate Cells, in 2 Mouse Models of Chronic Pancreatitis. *Gastroenterology.* 2015;149(3):765-76.
6. Mehta G, Scheinman RI, Holers VM et al., New Approach for the Treatment of Arthritis in Mice with a Novel Conjugate of an Anti-C5aR1 Antibody and C5 Small Interfering RNA. *J Immunol.* 2015;194(11):5446-54.
7. Tsai Y, Wu AC, Yang WB et al., ANGPTL4 Induces TMZ Resistance of Glioblastoma by Promoting Cancer Stemness Enrichment via the EGFR/AKT/4E-BP1 Cascade. *Int J Mol Sci.* 2019;20(22).
8. Theofilatos D, Fotakis P, Valanti E et al., HDL-apoA-I induces the expression of angiopoietin like 4 (ANGPTL4) in endothelial cells via a PI3K/AKT/FOXO1 signaling pathway.

Metabolism. 2018;87:36-47.

9. Hou M, Cui J, Liu J et al., Angiopoietin-like 4 confers resistance to hypoxia/serum deprivation-induced apoptosis through PI3K/Akt and ERK1/2 signaling pathways in mesenchymal stem cells. PLoS One. 2014;9(1): e85808.

Reviewer 2

My two major concerns are 1) the complete disregard-experimentally and in writing-of the ability of ANGPTL4 to raise plasma triglycerides, which is a very important risk factor for pancreatitis. 2) the lack of proper controls in a number of key experiments. For instance, the paper shows that the serum of ANGPTL4^{-/-} mice contains substantial levels of ANGPTL4. These two issues make me quite wary and form the basis of my recommendation to reject the manuscript. The paper is also a very tough read and the evidence is not presented and described in a straightforward and logical fashion. Because the paper has potential after extensive revision, the opportunity to resubmit can be considered.

The paper describes a novel stimulatory effect of ANGPTL4 on pancreatitis. A very fascinating and novel finding of this paper is that the severity of pancreatitis is markedly reduced in the absence of ANGPTL4 in two mouse models of pancreatitis. In addition, the work performed with ANGPTL4 neutralizing antibody is highly interesting. However, the proposed relationship between ANGPTL4, C5a, and macrophages remains relatively vague. It certainly does not help that the paper is difficult to read and the studies are often not presented in a logical and straightforward manner and order.

Point 1

It is very unfortunate that the paper does not mention an important rationale for studying the potential impact of ANGPTL4 on pancreatitis. ANGPTL4 raises plasma triglycerides. Elevated plasma triglycerides have been unequivocally established as key risk factor for pancreatitis. Accordingly, it is reasonable to suggest that ANGPTL4 may promote pancreatitis by raising plasma triglycerides. This line of reason is completely missing from the manuscript and should be addressed experimentally.

➤ **Our response**

As indicated, ANGPTL4 regulates lipid metabolism by inhibiting lipoprotein lipase activity and stimulating lipolysis in adipose tissue (1). Also, it has been known that ANGPTL4 is associated with the increased plasma triglycerides in various conditions including high fat diet and exercising muscle, and metabolic syndrome (1-3).

In case of pancreatitis, severe hypertriglyceridemia is one of rare risk factors associated with acute pancreatitis. Hypertriglyceridemia may result in acute pancreatitis in up to 7% of cases; however, it rarely occurs except when triglycerides levels are greater than 1,000 mg/dl (4,5). Also, it has been

discussed that acute pancreatitis induced by hypertriglyceridemia is a much rarer cause. Even, it has been shown that non-diabetic, non-alcoholic, non-obese patients with hypertriglyceridemia appear to account for only 15% of acute pancreatitis associated with hypertriglyceridemia (6). To confirm these facts, we evaluated TG level in pancreatitis patients (n=80) compared to normal subjects. As a result, there were no difference between AP patients and normal subjects. TG levels of all two groups were distributed in normal range (40-150 mg/dl). Also, we checked whether there are changes in TG level of ANGPTL4 WT and KO mice. However, we observed that ANGPTL4 did not affect TG regulation in AP and SAP models. Accordingly, in this study, we confirmed that ANGPTL4 did not promote pancreatitis by raising plasma triglycerides in pancreatitis although ANGPTL4 has been reported to raise plasma triglycerides by decreasing LPL activity in metabolic diseases. This was mentioned in discussion section.

New manuscript: page 15-17

ANGPTL4 is known to inactivate LPL, which reduces triglycerides (TG) conversion to free fatty acid, leading to hypertriglyceridemia in adipose tissue (Górecka *et al.*, 2019). Also, it has been reported that ANGPTL4 is associated with the increased plasma TG in various conditions including high fat diet and exercising muscle, and metabolic syndrome (Górecka *et al.*, 2019; Gao J *et al.*, 2019; Barchuk M *et al.*, 2019). Therefore, we expected that elevated plasma TG by ANGPTL4 could act as a key risk factor in pancreatitis. To confirm these, we evaluated TG level in pancreatitis patients (n=80) compared to normal subjects. As shown in Appendix Fig S1D, there was no difference between AP patients and normal subjects. TG levels of two groups were distributed in normal range (40-150 mg/dl). Indeed, hypertriglyceridemia may result in acute pancreatitis in up to 7% of cases; however, it rarely occurs except when triglycerides levels are greater than 1,000 mg/dl (Khan *et al.*, 2015). When we checked TG level in ANGPTL4 -/- and WT mice, there was no difference of TG level between ANGPTL4 -/- and WT mice, suggesting that ANGPTL4 did not affect TG regulation in AP and SAP models.

Also, previous studies have reported that ANGPTL4 is regulated by PPAR- γ in lipid metabolism and PPAR- γ plays a direct role in the inflammation (Rollins *et al.*, 2006; La *et al.*, 2017). In addition, activation of PPAR- γ induced the expression and secretion of ANGPTL4 in adipocyte lipid metabolism and tumor angiogenesis (Tian *et al.*, 2009; La *et al.*, 2017). However, the association with PPAR- γ and ANGPTL4 in pancreatitis has not been investigated. First, we evaluated the level and expression of PPAR- γ using serum and pancreas nucleic protein of ANGPTL4 $-/-$ and WT mice with pancreatitis. As shown in Appendix Fig S1E, there were little differences in serum PPAR- γ levels between ANGPTL4 $-/-$ and WT mice. We also identified PPAR- γ transcription level in pancreas of ANGPTL4 $-/-$ and WT mice with pancreatitis. As a result, although there was a little change in level of PPAR- γ according to the severity of pancreatitis, there was little change between ANGPTL4 $-/-$ and WT mice. To more confirm the role of PPAR- γ in regulation of ANGPTL4 in pancreatitis, we investigated expression of ANGPTL4 after PPAR- γ knockdown in the presence of ANGPTL4 in THP1-derived macrophage. After PPAR- γ knockdown, we observed that ANGPTL4 was highly expressed, showing that PPAR- γ was not involved in induction of ANGPTL4 in pancreatitis (Appendix Fig S1F). These findings suggest that there is little correlation between ANGPTL4 and PPAR- γ in pancreatitis.

References

1. Górecka M, Krzemiński K, Buraczewska M et al., Effect of mountain ultra-marathon running on plasma angiopoietin-like protein 4 and lipid profile in healthy trained men. *Eur J Appl Physiol.* 2019. doi: 10.1007/s00421-019-04256-w.
2. Gao J, Ding G, Li Q et al., Tibet kefir milk decreases fat deposition by regulating the gut microbiota and gene expression of Lpl and Angptl4 in high fat diet-fed rats. *Food Res Int.* 2019; 121:278-287.
3. Barchuk M, Schreier L, López G et al., Glycosylphosphatidylinositol-anchored high density lipoprotein-binding protein 1 and angiopoietin-like protein 4 are associated with the increase of lipoprotein lipase activity in epicardial adipose tissue from diabetic patients. *Atherosclerosis.* 2019;288:51-59.
4. Fortson MR, Freedman SN, Webster PD, Clinical assessment of hyperlipidemic pancreatitis. *Am J Gastroenterol.* 1995;90(12):2134-9.
5. Khan R, Jehangir W, Regeti K et al., Hypertriglyceridemia- Induced Pancreatitis: Choice of Treatment. *Gastroenterology Res.* 2015;8(3-4):234-236

6. Khan AS, Latif SU, Eloubeidi MA, Controversies in the etiologies of acute pancreatitis. JOP. 2010;11(6):545-52.

Point 2

The evidence presented suggests that ANGPTL4 induces various cytokines in acinar cells and vice versa that various cytokines induce ANGPTL4. However, the model presented in figure 7 only describes the former mechanism. Please elaborate on the chain of events by which ANGPTL4 is suspected to influence pancreatitis. This aspect of the paper is very confusing (Supplemental figures, Figure 5A-C). The paper follows a very convoluted and partly illogical path to arrive at the hypothesis that ANGPTL4 could directly regulate the progression of pancreatitis through macrophages.

➤ **Our response**

Compelling evidences have indicated that ANGPTL4 is involved in many inflammation-associated diseases. For instance, ANGPTL4 was induced by IL-1 β in osteoblasts (1) and by IL-1 β , TNF- α , IFN γ , or LPS in adipocytes (2). Treatment of mice with LPS induced ANGPTL4 expression in adipose tissue and muscle that were dependent on TLR4 signaling (3,4). For these reasons, we first investigated whether ANGPTL4 was induced in acinar cells by the cytokines, LPS, and CCK, which are pancreatitis inducers, because pancreatitis was induced by injury of acinar cells (Supplementary Figure 2 and Figure 5). Like previous studies, we observed that ANGPTL4 was induced by various cytokines in acinar cells.

Sometimes, ANGPTL4 is called as a multifunctional cytokine regulating vascular permeability, angiogenesis, and inflammation (5). Also, its dysregulation contributes to the pathogenesis of diseases, since ANGPTL4 is involved in many physiological processes with a variety of effects upon human health and disease. However, few direct effects of ANGPTL4 on inflammation are yet described. At this point, we expected that ANGPTL4 could also induce cytokine production although cytokines have been reported to induce ANGPTL4 in previous studies. In supplementary Figure 5, we found that ANGPTL4 increased the cytokines such as TNF- α , IL-6, and IL-1 β in acinar cells. More importantly, we observed that the effect of ANGPTL4 on the induction of cytokines was stronger than that of cytokine on ANGPTL4 in acinar cells. Additionally, we checked whether ANGPTL4 directly could affect the death of pancreatic acinar cells, leading to pancreatitis (supplementary Figure 5A). Since ANGPTL4 induced cytokine releases in acinar cells, we expected the induction of cell injury due to cytokines released by ANGPTL4. However, cell death by ANGPTL4 was weak compared with LPS and CCK. From this result, we concluded that the direct effect of ANGPTL4 on acinar cells appears to be small, and assumed that ANGPTL4 can cause acinar cell death through other mechanisms (acinar cell injury *via* C5a in macrophage). Namely, we speculated that there would be important mediators to regulate ANGPTL4 on cytokine releases.

In fact, cytokines can be initially released from acinar cells in inflammation condition, but much more is released from inflammatory cells such as macrophages for a long time. Therefore, we studied whether the main mechanism by which ANGPTL4 induces pancreatitis is mainly through the involvement of macrophage although ANGPTL4 partially and locally affects the acinar cells of pancreatitis. However, in this process, as reviewer pointed out, it seems to be the confusion in interpreting the results. Therefore, we corrected these parts in the results and deleted unnecessary data (acinar cell and cytokine, supplementary Figure 5B), which confused the interpretation of the results. Mechanism scheme of Figure 7E were also corrected to help understanding.

New summary Figure 7E

References

1. Noh JM, Shen C, Kim SJ et al., Interleukin-1 β increases Angptl4 (FIAF) expression via the JNK signaling pathway in osteoblastic MC3T3-E1 cells. *Exp Clin Endocrinol Diabetes*. 2015;123(8):445-60.
2. Lu B, Moser A, Shigenaga JK et al., The acute phase response stimulates the expression of angiopoietin like protein 4. *Biochem Biophys Res Commun*. 2010;391(4):1737-41.
3. Lu B, Moser A, Shigenaga JK et al., The acute phase response stimulates the expression of angiopoietin like protein 4. *Biochem Biophys Res Commun*. 2010;391(4):1737-41.
4. Brown R¹, Imran SA, Wilkinson M, Lipopolysaccharide (LPS) stimulates adipokine and socs3 gene expression in mouse brain and pituitary gland in vivo, and in N-1 hypothalamic neurons in vitro. *J Neuroimmunol*. 2009;209(1-2):96-103.
5. Yang X, Cheng Y, Su G, A review of the multifunctionality of angiopoietin-like 4 in eye disease. *Biosci Rep*. 2018;38(5).

Point 3

What is missing from the paper and what would be more useful that the analysis pursued with exogenous ANGPTL4 is a comparative cytokine profiling between WT and Angptl4^{-/-} macrophages. It is important to find out if C5a production is reduced in Angptl4^{-/-} macrophages (peritoneal or bone marrow derived). Currently, the link between ANGPTL4 and C5a is only established in vitro using external ANGPTL4 or overexpression systems.

➤ **Our response**

As reviewer indicated, we checked C5a production in macrophages from ANGPTL4 WT and KO mice. First, we induced the pancreatitis (AP and SAP) in ANGPTL4 WT and KO mice, respectively. Next, we prepared bone marrow-derived macrophages from pancreatitis-induced ANGPTL4 WT and KO mice. We found that C5a production was increased in the macrophages from ANGPTL4 WT mice with pancreatitis (AP and SAP). However, the level of C5a was lower in the macrophages from ANGPTL4 KO mice, compared to ANGPTL4 WT mice. We added the data in Figure 6B.

Point 4

The finding that serum ANGPTL4 levels are elevated in patients with pancreatitis should be confirmed using an established and validated ELISA from R&D Systems.

➤ **Our response**

In our study, we already validated the serum ANGPTL4 levels in patients with pancreatitis using human ELISA of R&D Systems. So, we mentioned it in material and method section.

Point 5

It is highly curious that the serum of ANGPTL4^{-/-} mice contain substantial levels of ANGPTL4. This result strongly indicates that the ELISA is not specific for ANGPTL4.

➤ **Our response**

It has been reported that some of the antibodies that have been used in commercial kits for

measurements of ANGPTL4 could cross-react with other factors (1). Therefore, as pointed out, we think that the ELISA kit used in our study may be not specific for ANGPTL4 (Sunred, Shanghai, China). Indeed, we measured serum ANGPTL4 level in patients with pancreatitis using human ANGPTL4 ELISA kit obtained from R&D system. Unfortunately, ANGPTL4 ELISA kit for mouse was not sold in R&D Systems. So, in order to accurately verify serum ANGPTL4 level in mice, two additional products (ANGPTL4 ELISA kits: ab210577, Abcam; NBS915491, Mybiosource) were purchased and measured. In two additional ELISA kits, the serum of ANGPTL4^{-/-} mice did not contain substantial levels of ANGPTL4. Therefore, data of ANGPTL4 level was replaced with new data in Figure 4B (Abcam data).

New data

References

1. Makoveichuk E, Ruge T, Nilsson S et al., High Concentrations of Angiopoietin-Like Protein 4 Detected in Serum from Patients with Rheumatoid Arthritis Can Be Explained by Non-Specific Antibody Reactivity. PLoS One. 2017;12(1):e0168922.

Point 6

The Western blots in figure 2 and S1C should indicate the molecular weight markers and contain a negative control for ANGPTL4 (serum and tissue from ANGPTL4^{-/-} mice). This is absolutely necessary as the availability of effective and specific anti-ANGPTL4 antibodies has been a huge problem. I thus must insist on the use of proper controls. The Western blot in figure 4B does contain a proper negative control. However, ANGPTL4 is not completely absent in the ANGPTL4^{-/-} SAP group. Please comment.

➤ Our response

As pointed out, the molecular weight in markers was indicated the western blots of Figure 2. In fact, the Western of Figure 2 was simultaneously performed with the sample of KO mice of Figure 4A.

Therefore, it is thought that there is a specificity of ANGPTL4 antibody used in Western blotting analysis of Figure 2. Although ANGPTL4 expression was time-dependently not confirmed in the ANGPTL4 WT and KO mice pancreatitis model, we think that these data are also meaningful (Figure 3B). Therefore, we decided to use it as supplementary data (supplementary Figure 3). Additionally, we identified the expression of ANGPTL4 in ANGPTL4 WT and KO mice with the ANGPTL4 antibody which was used in supplementary Figure 1C (new data: supplementary Figure 1E).

New Figure: supplementary Figure 1E

As reviewer mentioned in Figure 4B, ANGPTL4 is not completely absent in the ANGPTL4^{-/-}-AP and SAP mouse models. This seems to be due to the cross activity and low specificity of antibody used in the ANGPTL4 ELISA kit for mouse. Therefore, we purchased two additional ANGPTL4 ELISA kits for mouse and measured them again. As a result, we found that ANGPTL4 was rarely produced in ANGPTL4 KO AP and SAP mice, compared to WT mice. We replaced it with these data in Figure 3B (Abcam).

New Figures: Figure 3B

Point 7

The manuscript contains several graphs based on traditional RT-PCR (Figure 5D, 6C). This is a bit outdated and should be replaced by quantitative real-time PCR.

➤ **Our response**

As indicated, we performed RT-PCR , qPCR and western blotting again, and the results were replaced by new data in Figure 5A, Figure 6E and supplementary Figure 6C, respectively.

New Figure: Figure 5A

New Figure: Figure 6E

Minor comments

Point 1

Reference Oster et al. should be Köster et al.

➤ **Our response**

As pointed out, we checked and changed it.

Point 2

Some of the references cited are out of date or improperly cited. Authors are encouraged to use a more recent review than Oike (2005). Kersten (2000) did not identify the PPAR response element in the human and mouse ANGPTL4 gene.

➤ **Our response**

As pointed out, we added recent review papers and properly cited them.

Point 3

Page 4. Please adjust description "Herein, we provide evidence and underlying mechanisms that ANGPTL-4 leads to pancreatitis." Replace "leads to" by "promotes".

➤ **Our response**

As pointed out, we changed it.

Point 4

Page 5. "A 4-fold increase in ANGPTL4 level was observed in pancreatitis patients, with increases in amylase, lipase and CRP (Fig 1A)." . Please specify that ANGPTL4 levels were determined in serum. There is tendency throughout the manuscript to omit this type of relevant information, which renders the manuscript difficult to read

➤ **Our response**

We corrected the part you pointed out in the result section of Figure 1.

Point 5

The manuscript needs substantial English language editing.

➤ **Our response**

As your opinion, we corrected our manuscript by professional English language editing site.

Point 6

According to the methods section, "In case of human samples, human sample sizes achieved 99.99% power to detect a difference between both groups." How much of a difference? Moreover, the number of 99.99% is hard to believe. Please check and correct.

➤ **Our response**

We performed PASS power calculation with patient samples by a statistician and obtained the following results.

Two-Sample T-Test Power Analysis

Numeric Results for Mann-Whitney Test (Normal Distribution)

Null Hypothesis: Mean1=Mean2. Alternative Hypothesis: Mean1 <> Mean2

The standard deviations were assumed to be unknown and unequal.

Power	N1	N2	Allocation			Mean1	Mean2	S 1	S2
			Ratio	Alpha	Beta				
1.00000	10	90	9.000	0.05000	0.00000	19.9	103.1	8.1	50.3

References

1. Machin, D., Campbell, M., Fayers, P., and Pinol, A. 1997. Sample Size Tables for Clinical Studies, 2nd Edition. Blackwell Science. Malden, MA.
2. Zar, Jerrold H. 1984. Biostatistical Analysis (Second Edition). Prentice-Hall. Englewood Cliffs, New Jersey.
3. Al-Sunduqchi, Mahdi S. 1990. Determining the Appropriate Sample Size for Inferences Based on the Wilcoxon Statistics. Ph.D. dissertation under the direction of William C. Guenther, Dept. of Statistics, University of Wyoming, Laramie, Wyoming.

Report Definitions

Power is the probability of rejecting a false null hypothesis. Power should be close to one.

N1 and N2 are the number of items sampled from each population. To conserve resources, they should be small.

Alpha is the probability of rejecting a true null hypothesis. It should be small.

Beta is the probability of accepting a false null hypothesis. It should be small.

Mean1 is the mean of populations 1 and 2 under the null hypothesis of equality.

Mean2 is the mean of population 2 under the alternative hypothesis. The mean of population 1 is unchanged.

S1 and S2 are the population standard deviations. They represent the variability in the populations.

Summary Statements

Group sample sizes of 10 and 90 achieve 100% power to detect a difference of -83.2 between the null hypothesis that both group means are 19.9 and the alternative hypothesis that the mean of group 2 is 103.1 with estimated group standard deviations of 8.1 and 50.3 with a significance level (alpha) of 0.05000 using a two-sided Mann-Whitney test assuming that the actual distribution is normal.

Point 7

Figure 3B, graph for IL1beta appears to contain a mistake. It is very difficult to imagine that the differences between SAP and ANGPTL4 4mg is statistically significant. Please verify.

Our response

We performed statistical analysis by one-way ANOVA using SPSS software. The results are shown below. As shown below, significant difference of IL-1β between the SAP and the ANGPTL4 (4 mg/kg) groups was identified.

No	Con	AP	SAP	AP+ANGPTL4	ANGPTL4
1	0.38	0.99	1.93	1.81	1.96
2	0.451	1.33	1.88	1.76	2.45
3	0.48	1.23	1.77	1.72	1.9
4	0.35	0.88	1.83	1.73	2.22
5	0.45	1.22	1.99	1.58	1.97
6	0.48	1.08	2.01	1.63	2.38
7	0.59	1.22	2.22	1.77	2.22
8	0.34	0.99	2.11	1.99	2.08
9	0.41	0.87	1.97	1.55	2.19
10	0.37	1.22	2.09	1.8	2.29

종속 변수: IL1B

		Average difference ^a		Mean error ^a	Significance ^a	95% 신뢰구간	
(I) groups	(J) groups	평균차(I-J)	표준오차	유의확률	하한값	상한값	
Tukey HSD	1.00	2.00	-.6729 [*]	.06365	.000	-.8538	-.4920
		3.00	-1.5499 [*]	.06365	.000	-1.7308	-1.3690
		4.00	-1.3039 [*]	.06365	.000	-1.4848	-1.1230
		5.00	-1.7359 [*]	.06365	.000	-1.9168	-1.5550
	2.00	1.00	.6729 [*]	.06365	.000	.4920	.8538
		3.00	-.8770 [*]	.06365	.000	-1.0579	-.6961
		4.00	-.6310 [*]	.06365	.000	-.8119	-.4501
		5.00	-1.0630 [*]	.06365	.000	-1.2439	-.8821
	3.00	1.00	1.5499 [*]	.06365	.000	1.3690	1.7308
		2.00	.8770 [*]	.06365	.000	.6961	1.0579
		4.00	.2460 [*]	.06365	.003	.0651	.4269
		5.00	-.1860 [*]	.06365	.041	-.3669	-.0051
	4.00	1.00	1.3039 [*]	.06365	.000	1.1230	1.4848
		2.00	.6310 [*]	.06365	.000	.4501	.8119
		3.00	-.2460 [*]	.06365	.003	-.4269	-.0651
		5.00	-.4320 [*]	.06365	.000	-.6129	-.2511
	5.00	1.00	1.7359 [*]	.06365	.000	1.5550	1.9168
		2.00	1.0630 [*]	.06365	.000	.8821	1.2439
		3.00	.1860 [*]	.06365	.041	.0051	.3669
		4.00	.4320 [*]	.06365	.000	.2511	.6129

Point 8

In the model presented in figure 7E, it is not specified what is the source of the ANGPTL4. Could it be liver derived ANGPTL4? Or ANGPTL4 produced by macrophages?

Our response

As mentioned in manuscript, we first identified the level of ANPTL4 in serum of pancreatitis patients. As a result, we found that ANGPTL4 were highly increased in pancreatitis patients compared to normal control.

Since ANGPTL4 expression was found mainly in adipose tissue and liver, this molecule was first classified as an adipokine exclusively involved in lipid metabolism (1). Afterwards, this protein has

been shown to have a highly multifaceted role, since it is involved in several nonmetabolic and metabolic conditions, both physiological and pathological (1). Accordingly, although ANGPTL4 has been known to be mostly secreted from adipose tissue, liver, skeletal muscle, and heart (2), recent studies have reported that the secreted origin of ANGPTL4 could be different depending on pathological condition of various diseases.

For these reasons, we studied how the expression of ANGPTL4 changes in each tissue of pancreatitis (adipose, muscle, heart, liver, kidney, and pancreas tissues). In our supplementary Figure 1C, ANGPTL4 was well expressed in adipose, liver, muscle, and kidney excluding pancreas in normal control, whereas there was little change in ANGPTL4 expression of other organs excluding pancreas, adipose and muscle in AP and SAP condition. Notably, gradual increases of ANGPTL4 in pancreas were observed according to severity of pancreatitis. Although ANGPTL4 is secreted in other organs such as liver, muscle, and adipose, we speculated that the amount of ANGPTL4 secreted from the pancreas increases according to severity of pancreatitis, which may cause an increase in blood.

In this study, we speculated that systemic ANGPTL4 increased macrophage migration to pancreas, which increased C5a, leading to bomb of inflammatory cytokines, resulting in accelerating pancreatitis severity. In addition, ANGPTL4-overexpressed macrophages showed the increase in C5a (supplementary Figure 7C), suggesting an autocrine action of ANGPTL4 in macrophage. Similar autocrine action of ANGPTL4 was observed in many cancers. Based on current work, we also think that macrophage may be a source of ANGPTL4.

<Supplementary Figure 1C>

<Supplementary Figure 6D>

References

1. Zhu P, Goh YY, Chin HF et al., Angiotensin-like 4: a decade of research. *Biosci Rep.* 2012; 32(3):211-9.
2. Aryal B, Singh AK, Zhang X et al., Absence of ANGPTL4 in adipose tissue improves glucose tolerance and attenuates atherosclerosis. *JCI Insight.* 2018 ;22:3(6).

Point 9

It is unclear how the authors arrived at the dose for ANGPTL4 injection in mice. Did ANGPTL4 injection result in elevation of serum ANGPTL4 levels? This information should be provided.

➤ **Our response**

We performed IP injection to determine whether ANGPTL4 caused pancreatitis. As mentioned in the method section (pancreatitis induction), mice were sacrificed on 6 h after injection of ANGPTL4 (Figure 3A and B).

When ANGPTL4 was injected into mice, serum levels of ANGPTL4 were increased from 2 h. Although it was gradually decreased overtime, it showed a tendency to stay in high concentration of ANGPTL4 in ANGPTL4-injected groups, compared to other groups. We added these new data in Figure 3B.

New manuscript: page 6, line 14-17

As shown in Fig 2, the serum level of ANGPTL4 was increased in all groups except control group. Especially, there was a tendency to stay in a little high level in serum in ANGPTL4-injected groups [AP+ANGPTL4 (2 mg/kg) and ANGPTL4 (4 mg/kg)], compared to other groups.

New Figure: Figure 2B

Point 10

It is unclear why ANGPTL4 was injected in the AP animals but not the SAP animals. Please clarify.

➤ **Our response**

In general, the AP model is induced by cerulean, and the SAP model is induced by cerulein and additional injection of LPS (10 or 20 mg / kg). We wondered whether SAP was induced by ANGPTL4 of low dose (2 mg/kg) instead of LPS together with cerulein injection. We also wondered whether pancreatitis is induced by only ANGPT4 injection (4 mg/kg) without

cerulein and LPS. In this study, we found that ANGPTL4 could induce pancreatitis even at low doses. Also, even mice injected only with ANGPTL4 showed the increased severity of pancreatitis compared with the SAP model. This was further explained in the method section for better understanding.

New manuscript: page 22, line 13-17

In general, the AP model is induced by cerulein, and the SAP model is induced by cerulein and additional injection of LPS (10 or 20 mg/kg). At this time, we identified whether SAP was induced by ANGPTL4 of low dose (2 mg/kg) instead of LPS together with cerulein injection. Also, it was investigated whether pancreatitis is induced by only ANGPT4 injection (4 mg/kg) without cerulein and LPS.

Point 11

The observation that expression of CPT1A, PDK4, and perilipin-2 are not changed in the AP and SAP models cannot be used to argue that the increase in ANGPTL4 in pancreatitis is not mediated by PPAR- γ . I would recommend removing these data.

➤ **Our response**

As indicated, we deleted data of expression of CPT1A, PDK4, and perilipin-2 (supplementary Figure 1D).

Point 12

Page 9. "Therefore, we supposed that the effect of ANGPTL4 on cytokines would be rather large." It is unclear how the authors arrived at this statement.

➤ **Our response**

As pointed out, this content seems to be unnecessary in the result part, and was deleted.

Point 13

The supplementary data should preferably be reserved for supplemental data, not for key data that form an essential part of the narrative of the paper. The authors are encouraged to reconsider the distribution of data across regular and supplemental figures.

➤ **Our response**

As pointed out, we have rearranged the distribution of regular and supplemental Figures in this revised version.

Point 14

The observation that expression of C5a and C5aR expression were decreased in the ANGPTL4 -/- AP and SAP models does not point to a cause and effect relationship between ANGPTL4 and C5a but could simply reflect the general decrease in inflammation in the absence of ANGPTL4. In other words, there is no evidence that the relation between ANGPTL4 and C5a is specific. Please comment.

➤ **Our response**

ANGPTL4 is an orphan ligand. Until now, its receptor has not been found. ANGPTL4 was reported to be regulated by PPAR- α and integrin *etc.* in adipocyte and cancer (1), but the exact function and mechanism of ANGPTL4 has not been unveiled.

Also, the complement system plays an important role in the recognition and clearance of pathogens. It consists of 30 soluble and membrane-bound proteins that are rapidly mobilized through a cascade of enzymatic reactions and participate in host defenses through a range of mechanisms, including direct killing of bacteria, facilitation of phagocytosis, the recruitment and activation of immune cells (2,3). Among the many components of the complement system, C5a is a potent chemoattractant for macrophages and neutrophils and directly acts on a number of parenchymal cells *via* binding to the C5a receptor (CD88) (4). Numerous studies suggest the importance of C5a in various inflammatory diseases. Indeed, increased production of C5a has been reported in several pathological states such as pancreatitis and RA (5,6). Similarly, in the present study, we observed that ANGPTL4 increased C5a in macrophage, leading to the increased severity of pancreatitis. However, there has never been reported for the associations of ANGPTL4 and C5a until now.

In order to investigate the mechanism by which ANGPTL4 increased C5a, the mRNA expression of C5, a preform of C5a, was first identified. When macrophage was treated with ANGPTL4, mRNA expression of C5 only showed a little increase in human ANGPTL4-treated THP1-derived macrophage cells. Next, we checked whether ANGPTL4 increased activity of the C5 convertase

which cleaves C5 to C5a and C5b. C5 convertase was measured by the value of C5b-9. However, C5 convertase was highly not increased in ANGPTL4-treated THP1-derived macrophage cells.

Accordingly, we thought that ANGPTL4 would not directly regulate complement systems, but would increase C5a through cellular signaling pathways.

To evaluate whether ANGPTL4 increases C5a through regulation of cellular kinase signaling pathway in human macrophages (THP1), we performed a human phospho-kinase arrays analysis. As a result, ANGPTL4 increased the phosphorylation of AMPK α , JNK, AKT, and p70S6K in human THP1-derived macrophages. In same condition, we observed the increased expression of C5 by ANGPTL4 treatment. From these results, we speculated that ANGPTL4 may affect JNK, AMPK, and PI3K/AKT signaling.

To more confirm this result, each inhibitor (SP600125, compound C, and HS-173, respectively) was treated to THP1-derived macrophages after ANGPTL4 treatment. And then, we identified the expression of C5a and its related signaling. Among those, we observed that the PI3K inhibitor decreased the expression of C5a by inhibiting PI3K/AKT signaling. In some studies, ANGPTL4 has been reported to be regulated by PI3K/AKT pathway in endothelial cell, cancer cells, and stem cells (7-9). Although the exact mechanism by which ANGPTL4 increases C5a in macrophage was not

revealed, it was concluded that ANGPTL4 partially increases C5a through PI3K/AKT signaling. However, further studies on the mechanism of C5a regulation by ANGPTL4 is needed.

References

- La Paglia L, Listì A, Caruso S et al., Potential Role of ANGPTL4 in the Cross Talk between Metabolism and Cancer through PPAR Signaling Pathway. *PPAR Res.* 2017; 2017:8187235.
- Monk PN, Scola AM, Madala P et al., Function, structure and therapeutic potential of complement C5a receptors. *Br J Pharmacol.* 2007;152(4):429-48.
- Holers VM, Complement and its receptors: new insights into human disease. *Annu Rev Immunol.* 2014;32:433-59.
- Kemper C, Pangburn MK, Fishelson Z, Complement nomenclature 2014. *Mol Immunol.* 2014;61(2):56-8.
- Sendler M, Beyer G, Mahajan UM, Complement Component 5 Mediates Development of Fibrosis, via Activation of Stellate Cells, in 2 Mouse Models of Chronic Pancreatitis. *Gastroenterology.* 2015;149(3):765-76.
- Mehta G, Scheinman RI, Holers VM et al., New Approach for the Treatment of Arthritis in Mice with a Novel Conjugate of an Anti-C5aR1 Antibody and C5 Small Interfering RNA. *J Immunol.* 2015;194(11):5446-54.
- Tsai Y, Wu AC, Yang WB et al., ANGPTL4 Induces TMZ Resistance of Glioblastoma by Promoting Cancer Stemness Enrichment via the EGFR/AKT/4E-BP1 Cascade. *Int J Mol Sci.* 2019;20(22).

9. Theofilatos D, Fotakis P, Valanti E et al., HDL-apoA-I induces the expression of angiopoietin like 4 (ANGPTL4) in endothelial cells via a PI3K/AKT/FOXO1 signaling pathway. *Metabolism*. 2018;87:36-47.
10. Hou M, Cui J, Liu J et al., Angiopoietin-like 4 confers resistance to hypoxia/serum deprivation-induced apoptosis through PI3K/Akt and ERK1/2 signaling pathways in mesenchymal stem cells. *PLoS One*. 2014;9(1): e85808.

Point 15

Please specify what control was used for the ANGPTL4-neutralizing antibody.

➤ **Our response**

We used IgG antibody as a control. This information was added in results section.

Sincerely yours,

Soon-Sun Hong, Ph. D.

11th Feb 2020

Dear Prof. Hong,

Thank you for the submission of your revised manuscript to EMBO Molecular Medicine. We have now received the enclosed report from the two referees who were asked to re-assess it. As you will see below both referees still raise a couple of concerns on your work, which need to be addressed in a revision of the present manuscript.

In particular, we think that it is important to address referee #1's concerns with regard to the validation of key knockdown experiments and the PDAC mutation assessment of the pancreatitis tissue derived from tumor resections (the sequencing experiment suggested by this referee is not a mandatory for publication), and to discuss referee #2's concerns with regard to the ANGPTL4 Western blot data.

All other concerns raised by the referees need to be satisfactorily addressed as well.

Revised manuscripts should be submitted within three months of a request for revision; they will otherwise be treated as new submissions, except under exceptional circumstances in which a short extension is obtained from the editor.

I look forward to seeing a revised form of your manuscript as soon as possible.

Yours sincerely,

Jingyi Hou

Jingyi Hou
Editor
EMBO Molecular Medicine

*** Instructions to submit your revised manuscript ***

** PLEASE NOTE ** As part of the EMBO Publications transparent editorial process initiative (see

our Editorial at <https://www.embopress.org/doi/pdf/10.1002/emmm.201000094>), EMBO Molecular Medicine will publish online a Review Process File to accompany accepted manuscripts.

To submit your manuscript, please follow this link:

Link Not Available

- 1) a .doc formatted version of the manuscript text (including Figure legends and tables). Please make sure that the changes are highlighted to be clearly visible to referees and editors alike.
- 2) separate figure files*
- 3) supplemental information as Expanded View and/or Appendix. Please carefully check the authors guidelines for formatting Expanded view and Appendix figures and tables at <https://www.embopress.org/page/journal/17574684/authorguide#expandedview>
- 4) a letter INCLUDING the reviewers' reports and your detailed responses to their comments (as Word file)

Also, and to save some time should your paper be accepted, please read below for additional information regarding some features of our research articles:

- 5) The paper explained: EMBO Molecular Medicine articles are accompanied by a summary of the articles to emphasize the major findings in the paper and their medical implications for the non-specialist reader. Please provide a draft summary of your article highlighting
 - the medical issue you are addressing,
 - the results obtained and
 - their clinical impact.

- 6) For more information: There is space at the end of each article to list relevant web links for further consultation by our readers. Could you identify some relevant ones and provide such information as well? Some examples are patient associations, relevant databases, OMIM/proteins/genes links, author's websites, etc...

- 7) Author contributions: the contribution of every author must be detailed in a separate section (before the acknowledgments).

8) EMBO Molecular Medicine now requires a complete author checklist (<https://www.embopress.org/page/journal/17574684/authorguide>) to be submitted with all revised manuscripts. Please use the checklist as a guideline for the sort of information we need WITHIN the manuscript as well as in the checklist. This is particularly important for animal reporting, antibody dilutions (missing) and exact p-values and n that should be indicated instead of a range.

9) Every published paper now includes a 'Synopsis' to further enhance discoverability. Synopses are displayed on the journal webpage and are freely accessible to all readers. They include a short stand first (maximum of 300 characters, including space) as well as 2-5 one sentence bullet points that summarise the paper. Please write the bullet points to summarise the key NEW findings. They should be designed to be complementary to the abstract - i.e. not repeat the same text. We encourage inclusion of key acronyms and quantitative information (maximum of 30 words / bullet point). Please use the passive voice. Please attach these in a separate file or send them by email, we will incorporate them accordingly.

You are also welcome to suggest a striking image or visual abstract to illustrate your article. If you do please provide a jpeg file 550 px-wide x 400-px high.

10) A Conflict of Interest statement should be provided in the main text

11) Please note that we now mandate that all corresponding authors list an ORCID digital identifier. This takes <90 seconds to complete. We encourage all authors to supply an ORCID identifier, which will be linked to their name for unambiguous name identification.

Currently, our records indicate that the ORCID for your account is 0000-0001-7679-1388.

Please click the link below to modify this ORCID:
Link Not Available

12) The system will prompt you to fill in your funding and payment information. This will allow Wiley to send you a quote for the article processing charge (APC) in case of acceptance. This quote takes into account any reduction or fee waivers that you may be eligible for. Authors do not need to pay any fees before their manuscript is accepted and transferred to our publisher.

Photos 400-800 DPI

*Additional important information regarding figures and illustrations can be found at <http://bit.ly/EMBOPressFigurePreparationGuideline>

***** Reviewer's comments *****

Referee #1 (Comments on Novelty/Model System for Author):

N/A

Referee #1 (Remarks for Author):

The reviewers have been responsive to reviewers critiques, however, there some issues remaining needed to be addressed. Key knockdown experiments should be validated using a second targeting sequence and pancreatitis tissue derived from tumor resections should be sequenced. At least the the major PDAC mutations should be evaluated to rule-out that pancreatitis tissue lacks tumor.

Referee #2 (Comments on Novelty/Model System for Author):

Manuscript conveys a very interesting message. There are still many open ends but this is normal for this kind of paper. There are two issues that still bother me:

1) relatively poor writing, which unfortunately distracts from the message and overall quality of the paper.

2) Based on my own experience and the experience of my colleagues worldwide with antibodies against ANGPTL4, the Western blot data presented raise suspicion. It should be mentioned, though, that proper controls are included in the paper. Why this discrepancy between the nice ANGPTL4 Western blots presented in this paper and the lack of success experienced by others?

Referee #2 (Remarks for Author):

I am sorry to say but the rebuttal is often incomprehensible. The authors often don't manage to get their point across.

The answer about the power calculation is unsatisfactory. A power calculation should be done BEFORE performing the study and cannot be done retrospectively based on the values obtained in the study. The bottom line is that a proper power calculation was not performed. Any statement on having performed a power calculation should be removed.

In the PDF version available for review, many of the histological images (immunofluorescence, immunohistochemistry, H&E) have poor resolution and are not publication quality. This is especially true for figure 1 and 2. Authors should provide (much) higher quality images. My impression is that the error is in the conversion. Also, many images are much too small.

The paper remains difficult to read and contains numerous syntax and grammar mistakes. This distracts from an otherwise very interesting paper. The paper needs extensive editorial work after/before acceptance.

Page 4. "...which stimulate ANGPTL4 expression via a PPARresponse element in mouse and human

ANGPTL4 gene (La et al, 2017)". Please refer to the primary literature, not review articles. Proper reference here should be PMID: 15190076

A speculation on the source of the increase in ANGPTL4 during pancreatitis should be added to the discussion. If pancreas tissue is the source (as indicated by the initial microarray data showing a marked increase in ANGPTL4 in pancreatic tissue in two mouse pancreatitis model), which cells in the pancreas contribute to ANGPTL4 production and what is the trigger that leads to elevated ANGPTL4 production?

Authors should describe the source of recombinant ANGPTL4 that was injected into the mice and used for in vitro experiments.

Page 15. "ANGPTL4 is known to inactivate LPL, which reduces triglycerides (TG) conversion to free fattyacid, leading to hypertriglyceridemia in adipose tissue (Górecka et al., 2019)". This sentence is incorrect as such. Please remove "in adipose tissue". In this section, authors are encouraged to refer to the primary literature and not to review articles.

The data showing a lack of change in plasma TG between WT and Angptl4^{-/-} mice should be incorporated into the results section. In the present version, they are only described in the discussion section.

The same is true for the data on the relation between PPAR γ and Angptl4. Please move description of the results to the results section.

The experiment in S1F does not make any sense. Why test the effect of PPAR γ silencing on ANGPTL4 protein levels in macrophages treated with ANGPTL4? Please remove.

Answers to Reviewers' Comments

Date: May 11, 2020 Ms. No.: EMM-2019-11222

Authors: Jung *et al.*

Title: ANGPTL4 exacerbates pancreatitis by augmenting acinar cell injury through upregulation of C5a.”

Comment from Editor

We think that it is important to address referee #1's concerns with regard to the validation of key knockdown experiments and the PDAC mutation assessment of the pancreatitis tissue derived from tumor resections (the sequencing experiment suggested by this referee is not a mandatory for publication), and to discuss referee #2's concerns with regard to the ANGPTL4 Western blot data.

➤ Our response

We thank the reviewers for their thoughtful comments, time, and effort they put in to the review of our manuscript. We strongly believe it will improve the quality of our manuscript a lot. We tried our best efforts to make necessary corrections and additions of experimental data as the reviewers suggested. Our responses to the comments and our explanations for this revised manuscript are described below.

Comment from Reviewers

Reviewer 1

Point 1

The reviewers have been responsive to reviewer's critiques, however, there some issues remaining needed to be addressed. Key knockdown experiments should be validated using a second targeting sequence and pancreatitis tissue derived from tumor resections should be sequenced. At least the the major PDAC mutations should be evaluated to rule-out that pancreatitis tissue lacks tumor.

➤ Our response

As suggested by this reviewer, we performed new siANGPTL4 experiments using a second targeting sequence and confirmed the knockdown efficiency by western blot. Also, we provided information about the siRNA ANGPTL4-1 and siRNA ANGPTL4-2 in the materials and method section. These new data are presented in Fig. 6E

- 1) siANGPTL4-1: On-TARGETplus-mouse SMART-pool ANGPTL4 siRNA (Dharmacon, Cat. No: L-56622-00-0010). Patented modifications reduce off-targets for guaranteed gene silencing; pre-

designed ON-TARGETplus siRNA were used in SMART-pool format.

2) siANGPTL4-2 : Stealth mouse ANGPTL4 siRNA MSS226640

Revised manuscript: page 29

Transfection of siRNA

siRNA-control (siCon), ANGPTL4 siRNA 1 and 2 (siANGPTL4) were purchased from Dharmacon (On-TARGET plus-mouse SMART-pool ANGPTL4, L-56622-00-0010) and Invitrogen (Stealth mouse ANGPTL4 siRNA MSS226640), respectively. Cells were transfected with siRNA using Lipofectamine RNAiMAX transfection reagent (Invitrogen) according to the manufacturer's protocols.

Gene	Species	Transcripts	Transcript Type	Product Type	Avai
Angptl4	Mouse	1 RefSeq (NM)	Coding	Stealth siRNA	Mad

Gene Transcripts

Gene Symbol: Angptl4
Entrez Gene ID: 57875
Gene Name: angiotensin-like 4
Gene Aliases: Arp4, Bk89, Fiaf, Hfarp, Ng27, Pgar, Pparg, Pp1158
Chromosome Location: Chr. 17: 33774900 - 33781575 on Build GRCm38
UniGene ID: Mm.196189
Species: Mus musculus

Interrogated Sequence	Translated Protein	Targeted Exon(s)	siRNA Location
RefSeq: NM_020581.2	NP_065606.2	1	455

Revised Fig. 6E

Reviewer mentioned that at least the major PDAC mutations should be valued to reveal that pancreatitis tissue lacks tumor.

KRAS is mutated in more than 90% of human pancreatic cancer. Mutations of the codons G12, G13, or Q61 are usually associated with constitutively active KRAS, and recurrent mutations in K117 and A146 seem to be additional hotspots. Ninety-five percent of pancreatic cancers carry activating mutations in KRAS, and modifications in G12 account for 99% of all mutations (G12D—50%) (1). Clinical studies have shown that mutations of KRAS could be used as a significant prognostic biomarker, as well as a tool for therapy prediction.

In the case of *TP53* gene, most early investigators have analyzed *TP53* chiefly in exons 5–8, which is highly conserved through evolution, with 95% of the reported mutations. Additionally, among 560 mutations in studies in which the entire coding region of *p53* was sequenced, 87% were in exons 5–8, and most of the others were in exons 4 (8%) and 10 (4%) (2).

Guided by previous studies, we analyzed whether there were major pancreatic cancer mutations of *KRAS* and *TP53* in 5 non-tumor normal samples using DNA sanger sequencing.

KRAS was analyzed for G12, G13, and Q1 mutations (6 mutations) in exon 2 and exon 3, and *TP53* was analyzed for well-known mutations (7 mutations) in exon 5-8 (See below for detailed mutations). As shown in below figures, we confirmed that there were no mutations in *KRAS* and *TP53* genes in 5 non-tumor normal samples. We added it in material and methods section.

Revised manuscript: page 22

Samples from non-tumor normal region of pancreatic cancer patients were obtained during surgery. We performed a pancreatic cancer mutation analysis to confirm that pancreatic samples obtained from non-tumor normal region were real normal tissue. Sanger sequencing confirmed that there were no representative mutations in *KRAS* and *TP53* genes.

Reference

1. Bamford S, Dawson E, Forbes S *et al* The COSMIC (Catalogue of Somatic Mutations in Cancer) database and website. *Br J Cancer*. 2004; 91(2):355-8..
2. Yasushi Y, Hiroyuki W, Songür Y *et al* Detection of Mutations of *p53* Tumor Suppressor Gene in Pancreatic Juice and Its Application to Diagnosis of Patients with Pancreatic Cancer: Comparison with *K-ras* Mutation. *Clin Cancer Res*. 1999; 5:1147-53.

KRAS

Amplicon Variant Table (6 mutations of KRAS)

Variant ID	Reviewed	Variant Result	Variant Source	Base Position	Confidence Score	VAF %	Type	Effect	AA change	Amplicon ID	Layer ID	ROI ID	ROI Position	SNP ID	Category	Variant Comments
1 34G>N	no	accept	Known	34	0	0.0(N)	sub			KRAS_Exon2	NP_004976.2	NP_004976.2_region_1	225	G12C, G12S, G12R		Exon2
2 35G>N	no	accept	Known	35	59	0.0(N)	sub			KRAS_Exon2	NP_004976.2	NP_004976.2_region_1	226	G12D, G12V, G12A		Exon2
3 38G>R	no	accept	Known	38	0	0.0	sub			KRAS_Exon2	NP_004976.2	NP_004976.2_region_1	229	G13D		Exon2
4 181C>S	no	accept	Known	181	0	0.0	sub			KRAS_Exon3	NP_004976.2	NP_004976.2_region_1	372	Q61K, Q61E		Exon3
5 182A>H	no	accept	Known	182	0	0.0(T), 0.0(C)	sub			KRAS_Exon3	NP_004976.2	NP_004976.2_region_1	373	Q61P, Q61R, Q61L		Exon3
6 183A>H	no	accept	Known	183	0	0.0(T), 0.0(C)	sub			KRAS_Exon3	NP_004976.2	NP_004976.2_region_1	374	Q61H		Exon3

Specimen Variant Table

	Specimen ID	Genotype	Base Position	Genotype Result	User Edit	QV	Base Position Coverage	Amplicon ID	Layer ID	ROI ID	ROI Position	Genotype Comments
1 mutation	32011305_N_F	[=]+[=] Sample 1	34	-	no	65	2X	KRAS_Exon2	NP_004976.2	NP_004976.2_region_1	225	
	32063253_N_F	[=]+[=] " 2	34	-	no	65	2X	KRAS_Exon2	NP_004976.2	NP_004976.2_region_1	225	
	32147477_N_F	[=]+[=] " 3	34	-	no	65	2X	KRAS_Exon2	NP_004976.2	NP_004976.2_region_1	225	
	32159994_N_F	[=]+[=] " 4	34	-	no	65	2X	KRAS_Exon2	NP_004976.2	NP_004976.2_region_1	225	
	32355782_N_F	[=]+[=] " 5	34	-	no	65	2X	KRAS_Exon2	NP_004976.2	NP_004976.2_region_1	225	
2 mutation	32011305_N_F	[=]+[=] Sample 1	35	-	no	65	2X	KRAS_Exon2	NP_004976.2	NP_004976.2_region_1	226	
	32063253_N_F	[=]+[=] " 2	35	-	no	65	2X	KRAS_Exon2	NP_004976.2	NP_004976.2_region_1	226	
	32147477_N_F	[=]+[=] " 3	35	-	no	65	2X	KRAS_Exon2	NP_004976.2	NP_004976.2_region_1	226	
	32159994_N_F	[=]+[=] " 4	35	-	no	65	2X	KRAS_Exon2	NP_004976.2	NP_004976.2_region_1	226	
	32355782_N_F	[=]+[=] " 5	35	-	no	65	2X	KRAS_Exon2	NP_004976.2	NP_004976.2_region_1	226	
3 mutation	32011305_N_F	[=]+[=] Sample 1	38	-	no	65	2X	KRAS_Exon2	NP_004976.2	NP_004976.2_region_1	229	
	32063253_N_F	[=]+[=] " 2	38	-	no	65	2X	KRAS_Exon2	NP_004976.2	NP_004976.2_region_1	229	
	32147477_N_F	[=]+[=] " 3	38	-	no	65	2X	KRAS_Exon2	NP_004976.2	NP_004976.2_region_1	229	
	32159994_N_F	[=]+[=] " 4	38	-	no	65	2X	KRAS_Exon2	NP_004976.2	NP_004976.2_region_1	229	
	32355782_N_F	[=]+[=] " 5	38	-	no	65	2X	KRAS_Exon2	NP_004976.2	NP_004976.2_region_1	229	

	Specimen ID	Genotype	Base Position	Genotype Result	User Edit	QV	Base Position Coverage	Amplicon ID	Layer ID	ROI ID	ROI Position	Genotype Comments
4 mutation	32011305_N_F	[=][+]=	181	-	no	65	2X	KRAS_Exon3	NP_004976.2	NP_004976.2_region_1	372	
	32063253_N_F	[=][+]=	181	-	no	65	2X	KRAS_Exon3	NP_004976.2	NP_004976.2_region_1	372	
	32147477_N_F	[=][+]=	181	-	no	65	2X	KRAS_Exon3	NP_004976.2	NP_004976.2_region_1	372	
	32159994_N_F	[=][+]=	181	-	no	65	2X	KRAS_Exon3	NP_004976.2	NP_004976.2_region_1	372	
	32355782_N_F	[=][+]=	181	-	no	65	2X	KRAS_Exon3	NP_004976.2	NP_004976.2_region_1	372	
5 mutation	32011305_N_F	[=][+]=	182	-	no	65	2X	KRAS_Exon3	NP_004976.2	NP_004976.2_region_1	373	
	32063253_N_F	[=][+]=	182	-	no	65	2X	KRAS_Exon3	NP_004976.2	NP_004976.2_region_1	373	
	32147477_N_F	[=][+]=	182	-	no	65	2X	KRAS_Exon3	NP_004976.2	NP_004976.2_region_1	373	
	32159994_N_F	[=][+]=	182	-	no	65	2X	KRAS_Exon3	NP_004976.2	NP_004976.2_region_1	373	
	32355782_N_F	[=][+]=	182	-	no	65	2X	KRAS_Exon3	NP_004976.2	NP_004976.2_region_1	373	
6 mutation	32011305_N_F	[=][+]=	183	-	no	65	2X	KRAS_Exon3	NP_004976.2	NP_004976.2_region_1	374	
	32063253_N_F	[=][+]=	183	-	no	65	2X	KRAS_Exon3	NP_004976.2	NP_004976.2_region_1	374	
	32147477_N_F	[=][+]=	183	-	no	65	2X	KRAS_Exon3	NP_004976.2	NP_004976.2_region_1	374	
	32159994_N_F	[=][+]=	183	-	no	65	2X	KRAS_Exon3	NP_004976.2	NP_004976.2_region_1	374	
	32355782_N_F	[=][+]=	183	-	no	65	2X	KRAS_Exon3	NP_004976.2	NP_004976.2_region_1	374	

Snippets for Amplicon KRAS_Exon2

Snippets for Amplicon KRAS_Exon3

TP53

Amplicon Variant Table (7 mutations of TP53)

Variant ID	Reviewed	Variant Result	Variant Source	Base Position	Confidence Score	VAF %	Type	Effect	AA change	Amplicon ID	Layer ID	ROI ID	ROI Position	SNP ID	Category	Variant Comments
1	524G>N	no	accept	Known	524	0	0.0(N)	sub		TP53_Exon5	NP_000537.3	NP_000537.3_region_1	725	R175H, R175P, R175L	Exon5	
2	659A>V	no	accept	Known	659	0	0.0(G), 0.0(C)	sub		TP53_Exon6	NP_000537.3	NP_000537.3_region_1	860	Y220S, Y220C	Exon6	
3	733G>N	no	accept	Known	733	0	0.0(N)	sub		TP53_Exon7	NP_000537.3	NP_000537.3_region_1	934	G245S, G245R, G245C	Exon7	
4	743G>N	no	accept	Known	743	0	0.0(N)	sub		TP53_Exon7	NP_000537.3	NP_000537.3_region_1	944	R248Q, R248P, R248L	Exon7	
5	747G>N	no	accept	Known	747	0	0.0(N)	sub		TP53_Exon7	NP_000537.3	NP_000537.3_region_1	948	R249=, R249S, R249S	Exon7	
6	818G>N	no	accept	Known	818	0	0.0(N)	sub		TP53_Exon8	NP_000537.3	NP_000537.3_region_1	1019	R273H, R273P, R273L	Exon8	
7	844C>N	no	accept	Known	844	0	0.0(N)	sub		TP53_Exon8	NP_000537.3	NP_000537.3_region_1	1045	R282=, R282G, R282W	Exon8	

	Specimen ID	Genotype	Base Position	Genotype Result	User Edit	QV	Base Position Coverage	Amplicon ID	Layer ID	ROI ID	ROI Position	Genotype Comments
1 mutation	32011305_N_F	[=]+[=] Sample 1	524	-	no	65	2X	TP53_Exon5	NP_000537.3	NP_000537.3_region_1	725	
	32063253_N_F	[=]+[=] " 2	524	-	no	65	2X	TP53_Exon5	NP_000537.3	NP_000537.3_region_1	725	
	32147477_N_F	[=]+[=] " 3	524	-	no	65	2X	TP53_Exon5	NP_000537.3	NP_000537.3_region_1	725	
	32159994_N_F	[=]+[=] " 4	524	-	no	65	2X	TP53_Exon5	NP_000537.3	NP_000537.3_region_1	725	
	32355782_N_F	[=]+[=] " 5	524	-	no	65	2X	TP53_Exon5	NP_000537.3	NP_000537.3_region_1	725	
2 mutation	32011305_N_F	[=]+[=] Sample 1	659	-	no	65	2X	TP53_Exon6	NP_000537.3	NP_000537.3_region_1	860	
	32063253_N_F	[=]+[=] " 2	659	-	no	65	2X	TP53_Exon6	NP_000537.3	NP_000537.3_region_1	860	
	32147477_N_F	[=]+[=] " 3	659	-	no	65	2X	TP53_Exon6	NP_000537.3	NP_000537.3_region_1	860	
	32159994_N_F	[=]+[=] " 4	659	-	no	65	2X	TP53_Exon6	NP_000537.3	NP_000537.3_region_1	860	
	32355782_N_F	[=]+[=] " 5	659	-	no	65	2X	TP53_Exon6	NP_000537.3	NP_000537.3_region_1	860	
3 mutation	32011305_N_F	[=]+[=] Sample 1	733	-	no	65	2X	TP53_Exon7	NP_000537.3	NP_000537.3_region_1	934	
	32063253_N_F	[=]+[=] " 2	733	-	no	65	2X	TP53_Exon7	NP_000537.3	NP_000537.3_region_1	934	
	32147477_N_F	[=]+[=] " 3	733	-	no	65	2X	TP53_Exon7	NP_000537.3	NP_000537.3_region_1	934	
	32159994_N_F	[=]+[=] " 4	733	-	no	65	2X	TP53_Exon7	NP_000537.3	NP_000537.3_region_1	934	
	32355782_N_F	[=]+[=] " 5	733	-	no	65	2X	TP53_Exon7	NP_000537.3	NP_000537.3_region_1	934	

	Specimen ID	Genotype	Base Position	Genotype Result	User Edit	QV	Base Position Coverage	Amplicon ID	Layer ID	ROI ID	ROI Position	Genotype Comments
4 mutation	32011305_N_F	[=]+[=] Sample 1	743	-	no	65	2X	TP53_Exon7	NP_000537.3	NP_000537.3_region_1	944	
	32063253_N_F	[=]+[=] " 2	743	-	no	65	2X	TP53_Exon7	NP_000537.3	NP_000537.3_region_1	944	
	32147477_N_F	[=]+[=] " 3	743	-	no	65	2X	TP53_Exon7	NP_000537.3	NP_000537.3_region_1	944	
	32159994_N_F	[=]+[=] " 4	743	-	no	65	2X	TP53_Exon7	NP_000537.3	NP_000537.3_region_1	944	
	32355782_N_F	[=]+[=] " 5	743	-	no	65	2X	TP53_Exon7	NP_000537.3	NP_000537.3_region_1	944	
5 mutation	32011305_N_F	[=]+[=] Sample 1	747	-	no	65	2X	TP53_Exon7	NP_000537.3	NP_000537.3_region_1	948	
	32063253_N_F	[=]+[=] " 2	747	-	no	65	2X	TP53_Exon7	NP_000537.3	NP_000537.3_region_1	948	
	32147477_N_F	[=]+[=] " 3	747	-	no	65	2X	TP53_Exon7	NP_000537.3	NP_000537.3_region_1	948	
	32159994_N_F	[=]+[=] " 4	747	-	no	65	2X	TP53_Exon7	NP_000537.3	NP_000537.3_region_1	948	
	32355782_N_F	[=]+[=] " 5	747	-	no	65	2X	TP53_Exon7	NP_000537.3	NP_000537.3_region_1	948	
6 mutation	32011305_N_F	[=]+[=] Sample 1	818	-	no	65	2X	TP53_Exon8	NP_000537.3	NP_000537.3_region_1	1019	
	32063253_N_F	[=]+[=] " 2	818	-	no	65	2X	TP53_Exon8	NP_000537.3	NP_000537.3_region_1	1019	
	32147477_N_F	[=]+[=] " 3	818	-	no	65	2X	TP53_Exon8	NP_000537.3	NP_000537.3_region_1	1019	
	32159994_N_F	[=]+[=] " 4	818	-	no	65	2X	TP53_Exon8	NP_000537.3	NP_000537.3_region_1	1019	
	32355782_N_F	[=]+[=] " 5	818	-	no	65	2X	TP53_Exon8	NP_000537.3	NP_000537.3_region_1	1019	
7 mutation	32011305_N_F	[=]+[=] Sample 1	844	-	no	65	2X	TP53_Exon8	NP_000537.3	NP_000537.3_region_1	1045	
	32063253_N_F	[=]+[=] " 2	844	-	no	65	2X	TP53_Exon8	NP_000537.3	NP_000537.3_region_1	1045	
	32147477_N_F	[=]+[=] " 3	844	-	no	65	2X	TP53_Exon8	NP_000537.3	NP_000537.3_region_1	1045	
	32159994_N_F	[=]+[=] " 4	844	-	no	65	2X	TP53_Exon8	NP_000537.3	NP_000537.3_region_1	1045	
	32355782_N_F	[=]+[=] " 5	844	-	no	65	2X	TP53_Exon8	NP_000537.3	NP_000537.3_region_1	1045	

Reviewer 2

Manuscript conveys a very interesting message. There are still many open ends but this is normal for this kind of paper. There are two issues that still bother me:

Point 2

Relatively poor writing, which unfortunately distracts from the message and overall quality of the paper.

➤ Our response

We apologize for the poor writing. The revised manuscript has been thoroughly edited for clarity and readability, and changes are marked in blue color.

Point 3

Based on my own experience and the experience of my colleagues worldwide with antibodies against ANGPTL4, the Western blot data presented raise suspicion. It should be mentioned, though, that proper controls are included in the paper. Why this discrepancy between the nice ANGPTL4 Western blots presented in this paper and the lack of success experienced by others?

➤ Our response

First, we have tested 3 ANGPTL4 antibodies from different companies [Thermo (40-9800), Novos (NBP2-19016), abnova (h00051129-B01P)] which have been used in previous studies. Among them, we could detect the cleanest band with Thermo ANGPTL4 antibody when compared with other antibodies. Therefore, we used this antibody in our all experiment and was able to detect a clean band. This antibody was also used in following studies (see below).

Point 4

I am sorry to say but the rebuttal is often incomprehensible. The authors often don't manage to get their point across. The answer about the power calculation is unsatisfactory. A power calculation should be done BEFORE performing the study and cannot be done retrospectively based on the values obtained in the study. The bottom line is that a proper power calculation was not performed. Any statement on having performed a power calculation should be removed.

➤ **Our response**

We apologize that we misunderstood your point. As suggested, we have removed power calculation from in the revised manuscript.

Point 5

In the PDF version available for review, many of the histological images (immunofluorescence, immunohistochemistry, H&E) have poor resolution and are not publication quality. This is especially true for figure 1 and 2. Authors should provide (much) higher quality images. My impression is that the error is in the conversion. Also, many images are much too small.

➤ **Our response**

The original resolution of each TIFF image was 400-500 dpi. As this reviewer has correctly pointed out, the conversion to PDF file at journal site reduced the resolution of the images. We apologize for small image size, as each figure contained many subfigures, To the end, we have enlarged the size of each image and provided higher quality images in Figures 1 and 2.

Point 6

The paper remains difficult to read and contains numerous syntax and grammar mistakes. This distracts from an otherwise very interesting paper. The paper needs extensive editorial work after/before acceptance. Page 4. "...which stimulate ANGPTL4 expression via a PPAR response element in mouse and human ANGPTL4 gene (La *et al.*, 2017)". Please refer to the primary literature, not review articles. Proper reference here should be PMID: 15190076

➤ **Our response**

We apologize for our mistakes. The revised manuscript has been thoroughly edited for clarity and readability, and changes are marked in blue color. Also, we cited the primary reference.

New reference: page 38

Mandard S, Zandbergen F, Tan NS, Escher P, Patsouris D, Koenig W, Kleemann R, Bakker A, Veenman F, Wahli W Z *et al* (2004) The direct peroxisome proliferator-activated receptor target fasting-induced adipose factor (FIAF/PGAR/ANGPTL4) is present in blood plasma as a truncated protein that is increased by fenofibrate treatment. *J Biol Chem.* 279: 34411-234420.

Point 7

A speculation on the source of the increase in ANGPTL4 during pancreatitis should be added to the discussion. If pancreas tissue is the source (as indicated by the initial microarray data showing a marked increase in ANGPTL4 in pancreatic tissue in two mouse pancreatitis model), which cells in the pancreas contribute to ANGPTL4 production and what is the trigger that leads to elevated ANGPTL4 production?

➤ **Our response**

As mentioned in manuscript, the marked increase of ANGPTL4 in pancreatic tissue was observed by initial microarray data. Consistent with the data, we found that ANPTL4 was highly increased in serum and tissues of pancreatitis patients and animal models compared to cognate normal control.

ANGPTL4 was well expressed in normal adipose, liver, muscle, and kidney tissues, which showed little change in AP and SAP conditions (Appendix Fig S1C). In contrast, the expression of ANGPTL4 was undetectable in the normal pancreas. Notably, we observed that the amount of ANGPTL4 secreted from the pancreas increases according to severity of pancreatitis. Although other organs produce ANGPTL4, it is conceivable that the increased ANGPTL4 secreted by the pancreas contributes to the elevated ANGPTL in the blood during AP and SAP. We also speculated that systemic ANGPTL4 increased macrophage infiltration into the pancreas, which increased C5a, leading to hypercytokinemia or cytokine storm that accelerated pancreatitis severity. Lending supports, pancreatitis did not develop well in macrophage-depleted mice injected with recombinant ANGPTL4. Furthermore, ANGPTL4-overexpressed macrophages showed the increase in C5a (Appendix Fig S7C), suggesting an autocrine action of ANGPTL4 in macrophage. Similar autocrine action of ANGPTL4 was observed in cancer cells and endothelial cells. Considering these results, we postulate that residing and infiltrated macrophages may be a source of increased ANGPTL4 in the pancreas during AP and SAP. We have included this possible source of ANGPTL4 during pancreatitis in the discussion.

Revised manuscript: page 17

We determined multiorgan ANGPTL4 expression in AP and SAP models, and the expression of ANGPTL4 in the pancreas progressively increased according to severity of pancreatitis (Appendix Fig S1C). Although other organ produced ANGPTL4, it is conceivable that the increased ANGPTL4 secreted by the pancreas contributes to the elevated ANGPTL4 in the blood during AP and SAP.

Revised manuscript: page 18

In this regard, our results verified that ANGPTL4 is a pivotal regulator that exacerbates the severity of

pancreatitis *via* macrophage activation and migration. We also speculate that systemic ANGPTL4 increased macrophage infiltration into the pancreas, which increased C5a, leading to hypercytokinemia or cytokine storm that accelerated pancreatitis severity. Lending support, pancreatitis did not develop well in macrophage-depleted mice injected with recombinant ANGPTL4. Furthermore, ANGPTL4-overexpressed macrophages showed the increase in cytokines (Appendix Fig S7C), suggesting an autocrine action of ANGPTL4 in macrophage. Similar autocrine action of ANGPTL4 was observed in cancer cells and endothelial cells (Zhu *et al*, 2011; Le Jan *et al*,2003). Considering these results, we postulate that residing and infiltrated macrophages may be a source of increased ANGPTL4 in the pancreas during AP and SAP.

New reference

Le Jan S, Amy C, Cazes A, Monnot C, Lamandé N, Favier J, Philippe J, Sibony M, Gasc JM, Corvol P *et al* (2003) Angiopoietin-like 4 is a proangiogenic factor produced during ischemia and in conventional renal cell carcinoma. *Am J Pathol* 162(5):1521-1528.

Point 8

Authors should describe the source of recombinant ANGPTL4 that was injected into the mice and used for *in vitro* experiments.

➤ Our response

We apologize for the missing information. Commercial source of recombinant ANGPTL4 (mouse recombinant protein, R&D, Cat. NO: 4880-AN) was injected into the mice and used for *in vitro* experiments. This is now in the revised material and method.

Point 9

Page 15. "ANGPTL4 is known to inactivate LPL, which reduces triglycerides (TG) conversion to free fatty acid, leading to hypertriglyceridemia in adipose tissue (GÓrecka *et al.*, 2019)". This sentence is incorrect as such. Please remove "in adipose tissue". In this section, authors are encouraged to refer to the primary literature and not to review articles.

➤ Our response

As suggested, we have removed "in adipose tissue". We have also cited the primary literature and a relevant review on the role of ANGPTL4 in hypertriglyceridemia.

New reference

Mattijssen F, Kersten S. (2012) Regulation of triglyceride metabolism by Angiotensin-like proteins. *Biochim Biophys Acta*. 1821:782-789.

Point 10

The data showing a lack of change in plasma TG between WT and Angptl4^{-/-} mice should be incorporated into the results section. In the present version, they are only described in the discussion section. The same is true for the data on the relation between PPAR γ and Angptl4. Please move description of the results to the results section.

➤ **Our response**

As suggested, we included the data showing no change in plasma TG between WT and ANGPTL4^{-/-} mice and the relation between PPAR- γ and ANGPTL4 in the results section.

Revised manuscript: page 8-9

Association of ANGPTL4 with TG or PPAR- γ

ANGPTL4 is known to inactivate LPL, which reduces triglycerides (TG) conversion to free fatty acids, leading to hypertriglyceridemia (Sukonina *et al*, 2012; Mattijssen *et al*, 2019). Additionally, ANGPTL4 is associated with increased plasma TG (Gao J *et al*, 2019; Barchuk M *et al*, 2019). To identify whether elevated plasma TG by ANGPTL4 is a key risk factor in pancreatitis, we evaluated TG levels in pancreatitis patients (n=80) compared to those of normal subjects. As shown in Appendix Fig S1D, there was no difference between AP patients and normal subjects. The TG levels of two groups were distributed in normal range (40-150 mg/dl). Indeed, hypertriglyceridemia results in acute pancreatitis in up to 7% of cases; however, it rarely occurs except when triglycerides levels are greater than 1,000 mg/dl (Khan *et al.*, 2015). Also, we found that there was no difference in TG levels between ANGPTL4^{-/-} and WT mice, showing that ANGPTL4 does not affect TG regulation in AP and SAP models.

Previous studies have reported that ANGPTL4 is regulated by PPAR- γ in lipid metabolism and PPAR- γ plays a direct role in the inflammatory cascade during early events of AP (Rollins *et al*, 2006; Kennedy *et al*, 2008; La *et al*, 2017). In addition, activation of PPAR- γ induced the expression and secretion of ANGPTL4 in adipocyte lipid metabolism and tumor angiogenesis (Tian *et al*, 2009; La *et al*, 2017). However, the association of PPAR- γ and ANGPTL4 in pancreatitis has not been investigated. We identified PPAR- γ transcription levels in the pancreas of ANGPTL4^{-/-} and WT mice

with pancreatitis. There was little change in the level of PPAR- γ according to the severity of pancreatitis as well as between ANGPTL4 -/- and WT mice. These findings suggest that there is little correlation between ANGPTL4 and PPAR- γ in pancreatitis.

Point 11

The experiment in S1F does not make any sense. Why test the effect of PPAR γ silencing on ANGPTL4 protein levels in macrophages treated with ANGPTL4? Please remove.

➤ **Our response**

We performed the experiment on the request of another reviewer in our 1st revision. However, we found that ANGPTL4 was not regulated by PPAR- γ in pancreatitis. As you pointed out, we think that this result did not significantly affect the flow of this study (previous appendix Fig S1F). Therefore, we removed Fig. S1F from this revised manuscript.

Sincerely yours,

Soon-Sun Hong, Ph. D.

14th May 2020

Dear Prof. Hong,

Thank you for the submission of your revised manuscript to EMBO Molecular Medicine. We have now received the enclosed report from the two referees who were asked to re-assess it. As you will see below both referees still raise a couple of concerns on your work, which need to be addressed in a revision of the present manuscript.

Referee #2 pointed out that the abstract would benefit from language and text editing, and we would strongly recommend that you have the manuscript edited by a native English speaker. Referee #1 is still concerned about the antibody specificity, please address this as suggested by this referee.

On a more editorial level, please address these issues:

1. In the main manuscript file, please do the following:

- remove the blue color font.
- Author Contribution: Is JL Ju Han Lim or Ji Eun Lee? Please clarify.
- Indicate in legends exact n= and exact p= values, not a range, along with the statistical test used. Some people found that to keep the figures clear, providing an Appendix table Sx with all exact p-values was preferable. You are welcome to do this if you want to.
- Please check the figure callouts in the main article and make sure that all figures are called for. Currently Fig 7E is not called out.
- Please move The Paper Explained to the manuscript file.
- in Material and Methods, for animal work, gender, age and genetic background must be indicated, along with housing conditions.
- in Material and Methods and in the checklist, include a statement that informed consent was obtained from all subjects and that the experiments conformed to the principles set out in the WMA Declaration of Helsinki and the Department of Health and Human Services Belmont Report.
- in Material and Methods, for animal studies, include a statement about randomization even if no randomization was used.

2. Figures: Please add scale bars in all microscope images.

3. Source data: The source data needs to be provided as one file/ figure. Please combine Fig 6B,C,D,E,F into one Fig 6.

4. Checklist:

- Manuscript number and both correspondence authors' names should be on the checklist

5. We have replaced Supplementary Information with Expanded View (EV) Figures and Tables that are collapsible/expandable online (<https://www.embopress.org/page/journal/17574684/authorguide#expandedview>). A maximum of 5 EV Figures can be typeset. EV Figures should be cited as 'Figure EV1, Figure EV2" etc... in the text and their respective legends should be included in the main text after the legends of regular figures. Please remove the legends from individually uploaded EV figures.

For the figures that you do NOT wish to display as Expanded View figures, they should be bundled together with their legends in a single PDF file called *Appendix*, which should start with a short Table of Content. Appendix figures should be referred to in the main text as: "Appendix Figure S1, Appendix Figure S2" etc.

Please remember to update the figure callouts in the main text accordingly.

6. Data availability: Since this study does not provide large-scale primary datasets, please only add the following sentence in this section- "This study includes no data deposited in external repositories".

7. I noticed that you have provided a synopsis text. However, it appears incomplete. Synopsis text should include a short stand first (maximum of 300 characters, including space) as well as 2-5 one sentence bullet points that summarise the paper. Please write the bullet points to summarise the key NEW findings. They should be designed to be complementary to the abstract - i.e. not repeat the same text. Please use the passive voice. Please attach these in a separate file or send them by email, we will incorporate them accordingly.

Here are some examples:

<https://www.embopress.org/doi/10.15252/emmm.201911571>

<https://www.embopress.org/doi/10.15252/emmm.201910270>

<https://www.embopress.org/doi/10.15252/emmm.201911419>

7. As part of the EMBO Publications transparent editorial process initiative (see our Editorial at <http://embomolmed.embopress.org/content/2/9/329>), EMBO Molecular Medicine will publish online a Review Process File (RPF) to accompany accepted manuscripts.

In the event of acceptance, this file will be published in conjunction with your paper and will include the anonymous referee reports, your point-by-point response and all pertinent correspondence relating to the manuscript. Please let me know if you agree with this.

I look forward to seeing a revised version of your manuscript as soon as possible.

Yours sincerely,
Jingyi Hou

Jingyi Hou
Editor
EMBO Molecular Medicine

I look forward to seeing a revised form of your manuscript as soon as possible.

Yours sincerely,

Jingyi Hou

Jingyi Hou
Editor
EMBO Molecular Medicine

*** Instructions to submit your revised manuscript ***

** PLEASE NOTE ** As part of the EMBO Publications transparent editorial process initiative (see our Editorial at <https://www.embopress.org/doi/pdf/10.1002/emmm.201000094>), EMBO Molecular Medicine will publish online a Review Process File to accompany accepted manuscripts.

To submit your manuscript, please follow this link:

<https://embomolmed.msubmit.net/cgi-bin/main.plex>

- 1) a .docx formatted version of the manuscript text (including Figure legends and tables). Please make sure that the changes are highlighted to be clearly visible to referees and editors alike.
- 2) separate figure files*
- 3) supplemental information as Expanded View and/or Appendix. Please carefully check the authors guidelines for formatting Expanded view and Appendix figures and tables at <https://www.embopress.org/page/journal/17574684/authorguide#expandedview>
- 4) a letter INCLUDING the reviewers' reports and your detailed responses to their comments (as Word file)

Also, and to save some time should your paper be accepted, please read below for additional information regarding some features of our research articles:

- 5) The paper explained: EMBO Molecular Medicine articles are accompanied by a summary of the articles to emphasize the major findings in the paper and their medical implications for the non-specialist reader. Please provide a draft summary of your article highlighting
 - the medical issue you are addressing,
 - the results obtained and
 - their clinical impact.

6) For more information: There is space at the end of each article to list relevant web links for further consultation by our readers. Could you identify some relevant ones and provide such information as well? Some examples are patient associations, relevant databases, OMIM/proteins/genes links, author's websites, etc...

7) Author contributions: the contribution of every author must be detailed in a separate section (before the acknowledgments).

8) EMBO Molecular Medicine now requires a complete author checklist (<https://www.embopress.org/page/journal/17574684/authorguide>) to be submitted with all revised manuscripts. Please use the checklist as a guideline for the sort of information we need WITHIN the manuscript as well as in the checklist. This is particularly important for animal reporting, antibody dilutions (missing) and exact p-values and n that should be indicated instead of a range.

9) Every published paper now includes a 'Synopsis' to further enhance discoverability. Synopses are displayed on the journal webpage and are freely accessible to all readers. They include a short stand first (maximum of 300 characters, including space) as well as 2-5 one sentence bullet points that summarise the paper. Please write the bullet points to summarise the key NEW findings. They should be designed to be complementary to the abstract - i.e. not repeat the same text. We encourage inclusion of key acronyms and quantitative information (maximum of 30 words / bullet point). Please use the passive voice. Please attach these in a separate file or send them by email, we will incorporate them accordingly.

You are also welcome to suggest a striking image or visual abstract to illustrate your article. If you do please provide a jpeg file 550 px-wide x 400-px high.

10) A Conflict of Interest statement should be provided in the main text

11) Please note that we now mandate that all corresponding authors list an ORCID digital identifier. This takes <90 seconds to complete. We encourage all authors to supply an ORCID identifier, which will be linked to their name for unambiguous name identification.

Currently, our records indicate that the ORCID for your account is 0000-0001-7679-1388.

Link Not Available

12) The system will prompt you to fill in your funding and payment information. This will allow Wiley to send you a quote for the article processing charge (APC) in case of acceptance. This quote takes into account any reduction or fee waivers that you may be eligible for. Authors do not need to pay any fees before their manuscript is accepted and transferred to our publisher.

Photos 400-800 DPI

*Additional important information regarding figures and illustrations can be found at <http://bit.ly/EMBOPressFigurePreparationGuideline>

***** Reviewer's comments *****

Referee #1 (Comments on Novelty/Model System for Author):

None.

Referee #1 (Remarks for Author):

The authors have been responsive to reviewers critiques and have addressed most of the their concerns. However, the response to quality of the antibody is superficially addressing the antibody specificity. Controls such knockdown and overexpression studies should be used demonstrate the specificity of the antibody and all of the source files of the western blots included in the manuscript should be provided to check accuracy of the blotting assessment.

Referee #2 (Comments on Novelty/Model System for Author):

I would suggest that the journal editors help to improve the abstract. It has a poor flow.

Referee #2 (Remarks for Author):

No more comments

Answers to Reviewers' Comments

Date: May 20, 2020

Ms. No.: EMM-2019-11222

Authors: Jung *et al.*

Title: ANGPTL4 exacerbates pancreatitis by augmenting acinar cell injury through upregulation of C5a.”

Comment from Editor

We have now received the enclosed report from the two referees who were asked to re-assess it. As you will see below both referees still raise a couple of concerns on your work, which need to be addressed in a revision of the present manuscript.

Referee #2 pointed out that the abstract would benefit from language and text editing, and we would strongly recommend that you have the manuscript edited by a native English speaker. Referee #1 is still concerned about the antibody specificity, please address this as suggested by this referee.

On a more editorial level, please address these following issues.

➤ Our response

We thank the reviewers for their comments, time, and effort in the review of our manuscript. We tried our best efforts to make necessary corrections, as the reviewers suggested. Our responses to their comments are described below.

We have also addressed all the editorial issues.

Comment from Reviewers

Reviewer 1

Point 1

The authors have been responsive to reviewer's critiques and have addressed most of their concerns. However, the response to quality of the antibody is superficially addressing the antibody specificity. Controls such knockdown and overexpression studies should be used demonstrate the specificity of the antibody and all of the source files of the western blots included in the manuscript should be provided to check accuracy of the blotting assessment.

➤ Our response

As suggested by this reviewer, to confirm the specificity of the ANGPTL4 antibody, we repeatedly examined the expression of ANGPTL4 after the knockdown by two siANGPTL4 in ANGPTL4-overexpressed macrophages. As shown in figure below, the expression of ANGPTL4 was strongly increased in ANGPTL4-overexpressed cells (siCon) compared to normal macrophages (Con). ANGPTL4 expression was decreased after the knockdown (siANGPTL4-1 and siANGPTL4-2). We have included these in Appendix S1F and provided the necessary source files.

Original x-ray film

Reviewer 2

Point 1

I would suggest that the journal editors help to improve the abstract. It has a poor flow.

➤ Our response

As suggested by this reviewer, we have edited the abstract to improve on the flow. The revised abstract is below.

Revised Abstract

Pancreatitis is the inflammation of the pancreas. However, little is known about the genes associated with pancreatitis severity. Our microarray analysis of pancreatic tissues from mild and severe acute

pancreatitis mice models identified angiopoietin-like 4 (ANGPTL4) as one of the most significantly up-regulated genes. Clinically, ANGPTL4 expression was also increased in the serum and pancreatic tissues of pancreatitis patients. The deficiency in ANGPTL4 in mice, either by gene deletion or neutralizing antibody, mitigated pancreatitis-associated pathological outcomes. Conversely, exogenous ANGPTL4 exacerbated pancreatic injury with elevated cytokine levels and apoptotic cell death. High ANGPTL4 enhanced macrophage activation and infiltration into the pancreas, which increased complement component 5a (C5a) level through PI3K/AKT signaling. The activation of the C5a receptor led to hypercytokinemia that accelerated acinar cell damage and furthered pancreatitis. Indeed, C5a neutralizing antibody decreased inflammatory response in LPS-activated macrophages and alleviated pancreatitis severity. In agreement, there was a significant positive correlation between C5a and ANGPTL4 levels in pancreatitis patients. Taken together, our study suggests that targeting ANGPTL4 is a potential strategy for the treatment of pancreatitis. (174 words)

Sincerely yours,

Soon-Sun Hong, Ph. D.

26th May 2020

Dear Prof. Hong,

Thank you for the submission of your revised manuscript to EMBO Molecular Medicine. We have now received the enclosed report from the referee who was asked to re-assess it. As you will see the referee is now supportive and I am pleased to inform you that we will be able to accept your manuscript pending the following amendments:

1. Tables 1&2 needs to be renamed into "Appendix Table S1 &S2", and please also update the callouts in the main text. Also, the Appendix Table S2 is currently not called out, please fix.
2. In appendix Fig S3 A - the bottom scale bar needs to be aligned correctly. Currently spaced over two boxes.
3. Indicate in legends exact p= values, not a range, along with the statistical test used. Some people found that to keep the figures clear, providing an Appendix table Sx with all exact p-values was preferable. You are welcome to do this if you want to.
4. Our data editor has made a couple of suggestions on your manuscript(see attached), please fix.
5. I have slightly modified the synopsis text. Could you please let me know if you are fine with it of if you would like to introduce further modifications?

Synopsis text:

This study identifies ANGPTL4 as a potential pathological mediator that induces pancreatitis. Targeting ANGPTL4 by a neutralizing antibody is shown to be an effective strategy for the treatment of pancreatitis in mice.

- ANGPTL4 induces pancreatitis in mice and its expression is correlated with pancreatitis severity in acute pancreatitis patients.
- ANGPTL4 expression induces acinar cell injury as well as a massive release of inflammatory cytokines.
- ANGPTL4 plays a critical role in the regulation of C5a via PI3K/AKT signaling in macrophages.
- A neutralizing antibody against ANGPTL4 alleviates macrophage activation and improves pancreatitis severity in mice.

I look forward to reading a new revised version of your manuscript as soon as possible.

Yours sincerely,
Jingyi Hou

Jingyi Hou
Editor
EMBO Molecular Medicine

*** Instructions to submit your revised manuscript ***

To submit your manuscript, please follow this link:

<https://embomolmed.msubmit.net/cgi-bin/main.plex>

- 1) a .docx formatted version of the manuscript text (including Figure legends and tables)
- 2) Separate figure files*
- 3) supplemental information as Expanded View and/or Appendix. Please carefully check the authors guidelines for formatting Expanded view and Appendix figures and tables at <https://www.embopress.org/page/journal/17574684/authorguide#expandedview>
- 4) a letter INCLUDING the reviewer's reports and your detailed responses to their comments (as Word file).
- 5) The paper explained: EMBO Molecular Medicine articles are accompanied by a summary of the articles to emphasize the major findings in the paper and their medical implications for the non-specialist reader. Please provide a draft summary of your article highlighting
 - the medical issue you are addressing,
 - the results obtained and
 - their clinical impact.This may be edited to ensure that readers understand the significance and context of the research. Please refer to any of our published articles for an example.
- 6) For more information: There is space at the end of each article to list relevant web links for further consultation by our readers. Could you identify some relevant ones and provide such information as well? Some examples are patient associations, relevant databases, OMIM/proteins/genes links, author's websites, etc...
- 7) Author contributions: the contribution of every author must be detailed in a separate section.
- 8) EMBO Molecular Medicine now requires a complete author checklist (<https://www.embopress.org/page/journal/17574684/authorguide>) to be submitted with all revised manuscripts. Please use the checklist as guideline for the sort of information we need WITHIN the manuscript. The checklist should only be filled with page numbers where the information can be found. This is particularly important for animal reporting, antibody dilutions (missing) and exact

values and n that should be indicated instead of a range.

9) Every published paper now includes a 'Synopsis' to further enhance discoverability. Synopses are displayed on the journal webpage and are freely accessible to all readers. They include a short stand first (maximum of 300 characters, including space) as well as 2-5 one sentence bullet points that summarise the paper. Please write the bullet points to summarise the key NEW findings. They should be designed to be complementary to the abstract - i.e. not repeat the same text. We encourage inclusion of key acronyms and quantitative information (maximum of 30 words / bullet point). Please use the passive voice. Please attach these in a separate file or send them by email, we will incorporate them accordingly.

You are also welcome to suggest a striking image or visual abstract to illustrate your article. If you do please provide a jpeg file 550 px-wide x 400-px high.

10) A Conflict of Interest statement should be provided in the main text

11) Please note that we now mandate that all corresponding authors list an ORCID digital identifier. This takes <90 seconds to complete. We encourage all authors to supply an ORCID identifier, which will be linked to their name for unambiguous name identification.

Currently, our records indicate that the ORCID for your account is 0000-0001-7679-1388.

Link Not Available

12) The system will prompt you to fill in your funding and payment information. This will allow Wiley to send you a quote for the article processing charge (APC) in case of acceptance. This quote takes into account any reduction or fee waivers that you may be eligible for. Authors do not need to pay any fees before their manuscript is accepted and transferred to our publisher.

Photos 400-800 DPI

*Additional important information regarding figures and illustrations can be found at <http://bit.ly/EMBOPressFigurePreparationGuideline>

The system will prompt you to fill in your funding and payment information. This will allow Wiley to send you a quote for the article processing charge (APC) in case of acceptance. This quote takes into account any reduction or fee waivers that you may be eligible for. Authors do not need to pay

any fees before their manuscript is accepted and transferred to our publisher.

***** Reviewer's comments *****

Referee #1 (Comments on Novelty/Model System for Author):

N/A

Referee #1 (Remarks for Author):

The authors have fully addressed the reviewers critiques. This reviewer has no further comments. Congratulations on the great work!

YOU MUST COMPLETE ALL CELLS WITH A PINK BACKGROUND ↓
PLEASE NOTE THAT THIS CHECKLIST WILL BE PUBLISHED ALONGSIDE YOUR PAPER

Corresponding Author Name: Soon Sun Hong, Nguan Soon Tan
Journal Submitted to: EMBO Molecular Medicine
Manuscript Number: 2019-11222